# Million-year solar wind irradiation recorded in chang'E-5 and chang'E-6 samples

Renrui Liu [1,2,7], Xiaoping Zhang [1,2 ✉], Sizhe Zhao[1,2,7], Yi Xu [1,2 ✉], Pengwei Luo [1,2,7], Yang Li [3 ✉], Xiaojia Zeng [1,3], Chenkun Sun[1,2], Ronghua Pang [3], Chen Li[3,4], Xiongyao Li [3], Lianghai Xie[5], Zhiguo Meng [1,6], Qiugang Zong [1,2] & Chi Wang[5]

The long-term effects of Earth's magnetosphere on solar wind (SW) irradiation asymmetry between the lunar nearside and farside, and their implications for space weathering processes, remain poorly characterized. Here, we measure exposure ages and SW-induced amorphous rim thicknesses of individual grains from the Chang'E-5 (CE-5) and Chang'E-6 (CE-6) lunar soils to derive rim growth rates. Comparative analysis of SW irradiation records from CE-5, CE-6, and Apollo (11, 16, 17) samples reveals that CE-6 grains from the southern mid-latitude farside exhibit higher rim growth rates than those from the low-latitude nearside Apollo sites. This trend aligns with simulated lunar surface SW fluxes and is consistent with the hypothesis that reduced SW exposure on the nearside, due to Earth's magnetospheric shielding, may contribute to a persistent hemispheric asymmetry in SW irradiation. However, CE-5 samples from the northern mid-latitude nearside yield unexpectedly high rim growth rates, suggesting the potential involvement of additional local factors. The exact reasons for this anomaly remain unclear and warrant further investigation.

Solar wind (SW) ions, consisting mainly of low-energy H and He ions with typical energies around 1 keV/amu[1–4], can induce structural damage in lunar mineral grains, leading to the formation of SW-damaged amorphous rims[5–8]. This kind of rims record cumulative SW-induced radiation damage and serve as reliable proxies for long-term SW exposure[7,9]. The growth rates of such rims provide critical constraints for evaluating the contribution of SW to the overall space weathering processes on the lunar surface[7,10]. In contrast, solar energetic particles (SEPs)[11–14]—particularly high-energy heavier ions like Fe—can penetrate more deeply into mineral grains and produce damage tracks[7,15]. These tracks serve as a chronological marker that allows the exposure ages of individual grains to be estimated[7,16,17]. Although SEPs are not the main focus of this study, the grain exposure ages they help

determine are essential for quantifying the long-term growth rates of SW-damaged rims.

While individual SEP events can exhibit strong anisotropy depending on heliospheric magnetic connectivity, the cumulative SEP flux is generally considered near-isotropic across the lunar surface over geological timescale[18–20]. In comparison, the flux of lower-energy SW ions can be significantly modulated by various shielding mechanisms, including Earth's magnetosphere (especially the magnetotail), the bow shock at the SW–magnetosphere interface, direct occlusion by the Moon itself, and localized magnetic anomalies[21–28]. In a typical 28-day lunar orbit, the Moon spends approximately 25% of its time (~7 days) within Earth's magnetosphere, including up to ~4 days in the magnetotail, where the incident SW flux is largely suppressed due to

[1]State Key Laboratory of Lunar and Planetary Sciences, Macau University of Science and Technology, Macau, China. [2]CNSA Macau Center for Space Exploration and Science, Macau, China. [3]Center for Lunar and Planetary Sciences, Institute of Geochemistry, Chinese Academy of Sciences, Guiyang, China. [4]School of Engineering, Yunnan University, Kunming, China. [5]State Key Laboratory of Space Weather, National Space Science Center, Chinese Academy of Sciences, Beijing, China. [6]College of Geoexploration Science and Technology, Jilin University, Changchun, China. [7]These authors contributed equally: Renrui Liu, Sizhe Zhao, Pengwei Luo. ✉e-mail: xpzhangnju@gmail.com; yixu@must.edu.mo; liyang@mail.gyig.ac.cn

magnetic shielding[24,29]. During nearside daytime, the central region may spend roughly half its daytime within the magnetotail, experiencing partial shielding from incoming SW ions. In contrast, the sunlit farside remains continuously exposed to direct SW irradiation, without the geomagnetic protection[30]. Earth's magnetosphere can efficiently shield solar energetic protons with energies up to ~4 MeV when the Moon is within it during SEP events. Outside this protective environment, under direct SW exposure, the lunar body provides only limited occlusion of particles up to ~100 keV. For particles exceeding ~150 keV, the shadowing effect becomes negligible, as fluxes within the lunar wake approximate those in the ambient SW[30]. Moreover, although Earth's bow shock offers some protection, it can also inject additional low-energy particles (below a few hundred keV) into lunar orbit, exerting a significantly greater influence on the nearside relative to the farside[26,30]. Additionally, local magnetic anomalies on the Moon may form mini-magnetospheres that partially shield the surface by deflecting incoming SW particles, thereby reducing their flux and mitigating space weathering in certain regions[21–23,25]. This spatially variable shielding landscape gives rise to potential long-term asymmetries in SW irradiation between the lunar nearside and farside, which may in turn be recorded as distinct space weathering signatures in the surface regolith.

Lunar space weathering is primarily driven by the irradiation of charged particles—including the SW, SEPs, galactic cosmic rays (GCRs)[31–35]—as well as micrometeorite impacts[36–40] and episodic contributions from Earth's magnetospheric particles (often referred to as "Earth wind"), particularly when the Moon traverses the geomagnetic tail[20,41–47]. Over geological timescales, these processes collectively modify the physical and chemical properties of the lunar regolith, leading to progressive surface maturation[39,44]. Micrometeorite impacts continuously "garden" the regolith, cyclically excavating and burying grains[48]. Lunar surface grains retain distinct structural and chemical signatures of irradiation by SW and SEPs. Low-energy (~1 keV/amu) SW H/He ions induce radiation damage primarily within the outermost ~100 nm of grains, forming a characteristic SW-damaged rim[7]. In contrast, high-energy SEPs (e.g., Fe ions ranging from <1 MeV/nucleon to ~100 MeV/nucleon) penetrate to depths of a few millimeters, generating lattice damage and leaving latent particle tracks throughout the regolith[7,43,49–51]. Surface grains exposed for extended periods accumulate substantial radiation damage[43], reflecting persistent regional variations in SW irradiation[42,43,52]. These complementary records enable investigations into the spatial variability of long-term SW irradiation across the lunar surface.

To date, lunar soil samples returned by the Apollo missions have been employed to investigate the solar particle radiation differences across various regions[41–43]. Keller et al.[7] investigated the relationship between SW-damaged rims and exposure ages in lunar soils from various regions, finding no significant differences in the growth rates of SW-damaged rims among samples. These findings suggest that seemingly only minor variations in SW radiation exist across different regions[43,53]. However, these studies are limited to the low latitudes on the nearside of the Moon, leaving the conditions on the farside and higher latitudes largely unknown. To enhance our understanding of the long-term changes in the SW irradiation environment across different regions of the Moon and its effects on lunar soil evolution over a million-year timescale, an analysis of the SW-damaged rims and SEP track records in lunar soils from the lunar farside and at high-latitude regions is essential for gaining further insights. On December 17, 2020 (Beijing time), the Chang'E-5 (CE-5) mission successfully returned 1.731 kg of lunar soil from a mid-latitude region on the Moon's nearside, the northeastern Oceanus Procellarum (51.916°W, 43.058°N)[54]. On June 25, 2024 (Beijing time), the Chang'E-6 (CE-6) mission successfully returned the first samples from the South Pole-Aitken (SPA) basin (41.625°S, 153.978°W) at the lunar farside[55–57], providing a unique opportunity to investigate the long-term solar particle irradiation environment at the lunar farside and its differences from the nearside.

In this work, we examine the SW-damaged rims and SEP tracks within the CE-5 and CE-6 samples, explore the long-term differences in SW irradiation environment between the nearside and the farside of the Moon over millions of years, and study their impacts on the evolution of lunar soil grains.

## Results and discussion

To allow comparison with previous Apollo findings and minimize the influence of mineral-type variations, we restricted our analysis to plagioclase grains and employed the same experimental methods as in prior studies[7,58]. In the CE-5 (CE5C1000YJFM006, CE5C0400YJFM00505, CE5C0600YJFM00304, CE5C0300YJFM00401) and CE-6 (CE6C0300YJFM001, CE6C0300YJFM00401) shoveled samples, we selected grains exhibiting surface exposure features such as micrometeorite impact craters or melt splashes, indicating prior exposure on the lunar surface and thus recording SW and SEP irradiation damage (Fig. 1a)[59,60]. In total, seven grains from CE-5 (CE5-01 to CE5-07) and eight from CE-6 (CE6-01 to CE6-08) were selected for the focused ion beam (FIB) cross-section, respectively. As shown in Fig. 1b–d, the scanning transmission electron microscopy (STEM) and high-resolution transmission electron microscopy (HRTEM) images of the FIB cross-section indicate that the uppermost surface of the sample consists of a space weathering-induced amorphous rim, while the underlying matrix region contains solar particle tracks. The formation of rims and solar particle tracks is closely related to the exposure of samples on the lunar surface. The types of rims are primarily classified as SW-damaged rims and vapor-deposited rims. In all samples, SW-damaged rims were observed. In the CE-5 samples, their thickness ranged from approximately 40–64 nm, while in the CE-6 samples, their thickness varied from around 19 to 109 nm (Figs. S1–S15). Quantitative transmission electron microscopy-energy dispersive X-ray spectroscopy (TEM-EDS) composition maps indicate that SW-damaged rim composition is consistent with

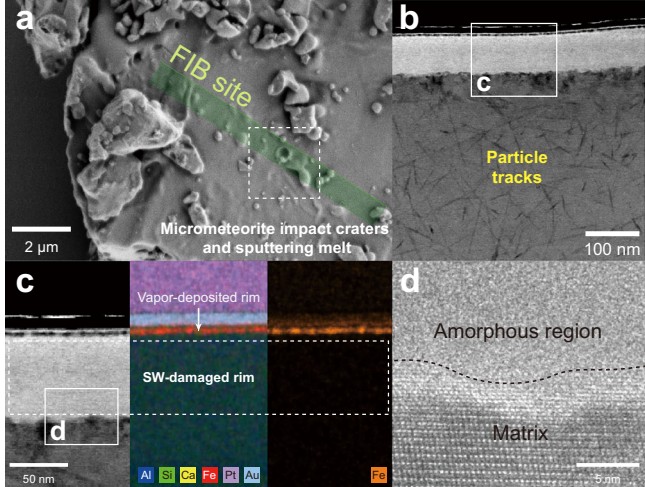

**Fig. 1 | The microscopic characteristics of representative studied grains.**
**a** Secondary electron (SE) image of the studied CE6-PL1 grain. The green rectangle indicates the FIB cross-section site. **b**, **d** STEM and HRTEM image of the studied grain surface, black dashed line represents the boundary between the amorphous rim and the matrix (**c**) STEM, quantitative TEM-EDS maps, and corresponding TEM-EDS profiles of space weathering rim. The white arrow and white dashed box indicate the vapor-deposited rim and solar wind (SW)-damaged rim, respectively. Note that FIB, STEM, HRTEM, and TEM-EDS are abbreviations for Focused Ion Beam, Scanning Transmission Electron Microscopy, High-Resolution Transmission Electron Microscopy, and Transmission Electron Microscopy-Energy Dispersive X-ray Spectroscopy, respectively.

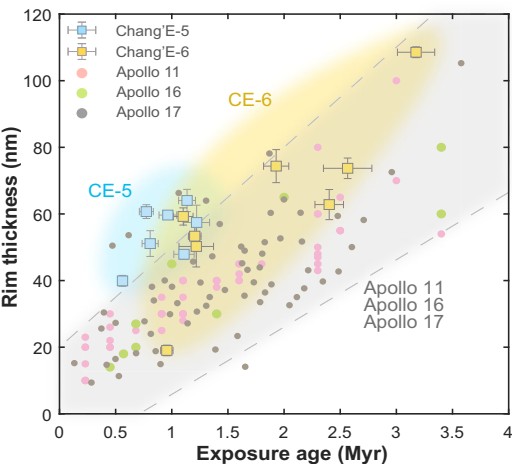

**Fig. 2 | Comparisons of solar wind (SW)-damaged rim thickness and exposure age in lunar grains from Chang'E-5, Chang'E-6, Apollo 11[7], Apollo 16[7], and Apollo 17[61] missions.** Symbols distinguish grains from different missions: light blue squares (Chang'E-5), yellow squares (Chang'E-6), pink circles (Apollo 11), green circles (Apollo 16), and gray circles (Apollo 17). Light blue and yellow shaded regions indicate typical data ranges for Chang'E-5 ("CE-5") and Chang'E-6 ("CE-6"), respectively. A light gray shaded region, bounded by two dashed lines and the figure edges, marks the range spanned by most Apollo mission data (Apollo 11, 16, and 17).

the matrix (Fig. 1c). However, the impact events are episodic, and we did not find vapor-deposited rims in all samples. In the six particles containing CE5-PL1, CE5-PL3, CE5-PL7, CE6-PL1, CE6-PL7, and CE6-PL8, iron-rich vapor-deposited rims were observed (Figs. S1, S3, S7, S8, S14, S15). Particle tracks within the matrix were observed in all grains. Grain CE6-PL1 is shown in Fig. 1, with the remaining grains presented in Figs. S1 to S15 of the *Supplementary Information*.

Prolonged exposure induces both continuous thickening of SW-damaged rims and increased density of particle tracks in grains. We counted the track density of each grain and measured the thickness of the surface SW-damaged rim (Table S1), and further calculated the exposure age of these grains through the track density analysis based on the track production rate[7]. For further details on the track density analysis, please refer to the "*Methods*" section[7,61]. The relationship between the exposure age of the grain and the thickness of SW-damaged rim can reflect the SW irradiation in the region, as shown in Fig. 2 for CE-5, CE-6, Apollo 11, Apollo 16, and Apollo 17[62].

## Solar wind-damaged rim growth rates

The distribution of SW-damaged rim thickness against exposure age in lunar grains from different sampling sites is shown in Fig. 2, where the two variables exhibit a positive correlation overall. In the interval with exposure ages less than 4 Myr, this relationship can be approximated by a linear model. The data exhibit a degree of scatter. Notably, the CE-5 data points are relatively clustered, probably because the samples were collected from a single basalt unit with a relatively young formation age (~2.0 Gyr)[63]. In contrast, the CE-6 (~2.8 Gyr)[64] samples and those from the Apollo missions[65]—including Apollo 11 (3.6–3.9 Gyr), Apollo 16 (3.4–3.8 Gyr), and Apollo 17 (3.7–3.8 Gyr)—originate from regions with older formation ages, more complex geological contexts (e.g., mare-highland boundary), and a higher likelihood of incorporating exotic components. Moreover, due to the Apollo missions' samples being collected through extravehicular activities (EVAs) and considering the

significant distances between different stations, this could also lead to the collection of samples with varying exposure histories. These combined factors likely account for the broader dispersion observed in the data. At the same exposure age around 1.20 Myr, CE-5 samples exhibit the greatest SW-damaged rim thickness, followed by CE-6, with Apollo samples showing the smallest values (Fig. 2). To quantify and compare SW irradiation levels across these sampling sites, the SW-damaged rim growth rate was defined. To extract the growth rate from different sites, a linear model $D = k \cdot t$ was used to fit SW-damaged rim thickness data against exposure age for these five sampling sites. Here, $k$ represents the growth rate of the SW-damaged rim (in nm/Myr), $D$ is the rim thickness (in nm), $t$ is the exposure age (in Myr), and the fitting error uses the standard deviation ($\sigma$). Table 1 shows that growth rates are highest in CE-5 samples, intermediate in CE-6, and lowest in Apollo samples. This variability can be attributed to local SW irradiation conditions. Note that both CE-5 and Apollo sampling sites are located on the lunar near side: the CE-5 site lies at mid-northern latitude, while the Apollo sites are near the equator. In contrast, the CE-6 sampling site is situated at mid-southern latitude on the lunar far side. Detailed geographic coordinates for these sites are provided in Table 1.

## Solar wind flux across sampling sites

Lunar surface SW flux is strongly influenced by latitude, Earth's magnetosphere, and lunar magnetic anomalies. Xie et al.[22] employed Hall MHD simulations to investigate SW irradiation conditions across the entire Moon, accounting for the abovementioned factors. Using data from the ARTEMIS spacecraft[66] and the Chang'E-4 (CE-4) rover during 01:00–08:00 UT on December 31, 2019, the model successfully reproduced the interaction between the SW and strong lunar magnetic anomalies. Key features observed in situ were accurately captured, including distinct boundary structures and wake dynamics consistent with plasma and magnetic field profiles along the ARTEMIS P2 trajectory. The model also reproduced shock-associated density enhancements and magnetic field rotations in close agreement with ARTEMIS measurements. Furthermore, it simulated the formation of a downstream plasma cavity, consistent with observed plasma depletion linked to magnetotail dynamics. Notably, the simulations replicated the observed reduction in SW kinetic energy and flux, with deceleration rates and shielding efficiencies consistent with CE-4's energetic neutral atom (ENA) flux data, confirming partial surface shielding by the magnetic anomaly. These results validate the model's capability to resolve the complex dynamics of SW-lunar magnetic anomaly interactions[22,23]. The lunar surface maps depicting the distributions of average normalized SW number flux ($J_{avg}$) and energy flux ($E_{f,avg}$) are presented in Fig. 3a, c. Note that the normalized SW energy flux is defined as $E = NV_r V^2/(N_{sw}V_{sw}^3)$ in this work, where $N$, $V$, and $V_r$ denote the simulated SW number density, total velocity, and radial velocity component along the lunar radius, respectively, while $N_{sw}$ and $V_{sw}$ represent parameters of the undisturbed upstream SW. This definition contrasts with that employed in Xie et al.[22]. Please refer to Table 1 for more details. Since the model is primarily constructed using recent short-term observations and does not incorporate the long-term evolution of solar activity, magnetic field intensity, etc., its applicability to million-year-scale SW irradiation on the lunar surface may be limited.

Results show that the mid-to-high-latitude regions on the lunar farside exhibit lower $J_{avg}$ and $E_{f,avg}$ due to the solar zenith angle effect. Additionally, $E_{f,avg}$ is more strongly influenced by magnetic anomaly regions than $J_{avg}$, as lunar magnetic anomalies only partially shielded SW protons, particularly in weakly magnetized areas[22,25]. This partial shielding results in limited reductions in the number flux within crustal magnetic fields due to SW penetration. However, deceleration during penetration leads to a decrease in energy flux, making energy flux more sensitive to the presence of magnetic

**Table 1 | Growth rates of SW-damaged rims, together with average normalized solar wind (SW) number flux $J_{avg}$ and energy flux $E_{f,avg}$ at selected lunar sampling sites**

| Sampling site | Chang'E-5 | Chang'E-6 | Apollo 11 | Apollo 16 | Apollo 17 | Farside subsolar point |
|---|---|---|---|---|---|---|
| Location coordinates | 43.06°N 51.92°W | 41.63°S 153.98°W | 0.67°N 23.47°E | 8.97°S 15.50°E | 20.19°N 30.77°E | 0° 180° |
| SW-damaged rim growth rate $k$ (nm/Myr) | 55.96 ± 10.82 | 33.21 ± 6.16 | 25.10 ± 1.76 | 23.52 ± 5.36 | 25.40 ± 2.32 | – |
| Average normalized SW number flux $J_{avg}$ | 0.1929 | 0.2359 | 0.2321 | 0.2171 | 0.2233 | 0.3101 |
| Average normalized SW energy flux $E_{f,avg}$ | 0.1785 | 0.2309 | 0.2036 | 0.1116 | 0.1961 | 0.3058 |

The distribution of SW number flux and energy flux across the entire lunar surface was simulated for various positions of the Moon in its orbit around the Earth, with both fluxes being normalized. The normalized SW number flux and energy flux are defined as $J = NV_r/(N_{sw}V_{sw})$ and $E = NV_rV^2/(N_{sw}V_{sw}^3)$, where $N$ and $V_r$ represent the simulated SW number density and its radial velocity component along the lunar radius. The constants $N_{sw}$ and $V_{sw}$ are set to be 5 $cm^{-3}$ and 400 km/s, which are parameters of the undisturbed upstream solar wind, used for normalization such that $N_{sw}V_{sw}$ and $N_{sw}V_{sw}^3$ correspond to unit number flux and energy flux. The average normalized SW number flux ($J_{avg}$) and energy flux ($E_{f,avg}$) across the entire lunar surface during the Moon's orbital period were then calculated. Note that, apart from the definition of SW energy flux, all other aspects of the model calculations in this work align with those described in Xie et al.[22].

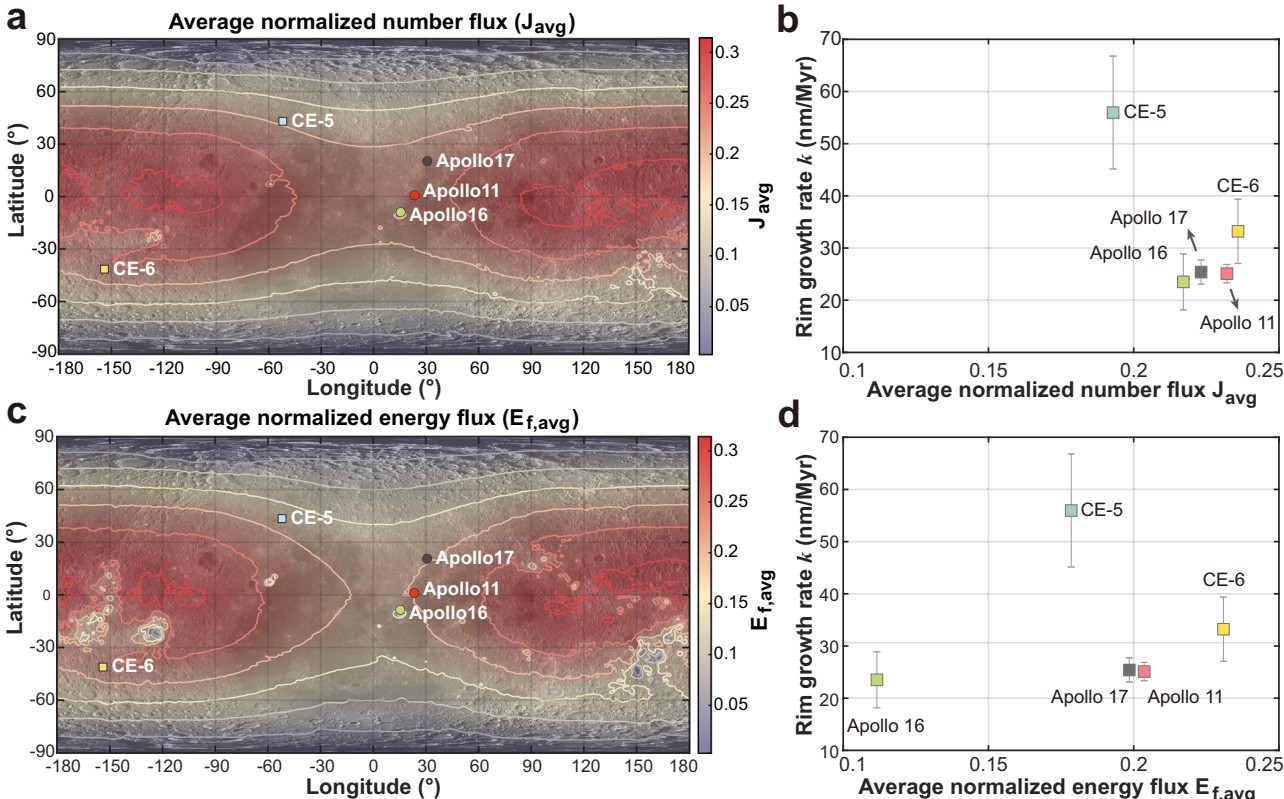

**Fig. 3 | Maps of the lunar surface showing average normalized solar wind (SW) number flux ($J_{avg}$) and energy flux ($E_{f,avg}$), alongside profiles of SW-damaged rim growth rate ($k$) versus $J_{avg}$ and $E_{f,avg}$. a, c** Lunar surface maps of $J_{avg}$ and $E_{f,avg}$, derived from Hall MHD simulations that incorporate latitude, Earth's magnetosphere, and lunar magnetic anomalies. **b, d** Profiles of SW-damaged rim growth rate $k$ versus $J_{avg}$ and $E_{f,avg}$ at different sampling sites, including Chang'E-5 (CE-5),

Chang'E-6 (CE-6), Apollo 11, Apollo 16, and Apollo 17. Further details on $J_{avg}$ and $E_{f,avg}$ can be found in the caption to Table 1. Note that the SW flux data in panel (**a**) are from Xie et al.[22], whereas those in panel (**c**) are calculated using the normalized SW energy flux definition $E = NV_rV^2/(N_{sw}V_{sw}^3)$, where $N$ and $V_r$ denote the simulated SW number density and its radial velocity component along the lunar radius, respectively; $N_{sw}$ and $V_{sw}$ represent parameters of the undisturbed upstream SW.

anomalies. Here, based on the simulation results, we extracted the $J_{avg}$ and $E_{f,avg}$ on the CE-5, CE-6, Apollo 11, Apollo 16, and Apollo 17 sampling sites, as well as at the lunar farside subsolar point, as presented in Table 1.

The CE-5 sampling site, located at mid-latitudes (~43°N) on the lunar near side, exhibits a relatively lower $J_{avg}$ than other compared sampling sites. While the CE-6 sampling site on the lunar farside shows the highest $J_{avg}$ and $E_{f,avg}$, these values are about 24% lower than those at the farside subsolar point. This discrepancy arises mainly from latitudinal differences and can be corrected by applying the factor

$\cos\theta$, with $\theta$ defined as the sampling site's latitude. Specifically, the CE-6 sampling site at 41.63°S yields a correction factor of $\cos(41.63°) = 0.747$, reducing $J_{avg}$ and $E_{f,avg}$ by 25.3% compared to corresponding values at the lunar farside subsolar point. This reduction is in good agreement with the above-mentioned 24% model-derived values. The Apollo 11, Apollo 16, and Apollo 17 sampling sites near the nearside equator lie at relatively low latitudes. To account for latitudinal effects, their flux values were corrected by dividing by the factor $\cos\theta$. The latitudinally corrected average SW number and energy fluxes, $J_{avg\_cor}$ and $E_{f,avg\_cor}$, for these three sites are (0.2321,

0.2198, 0.2379) and (0.2036, 0.1130, 0.2089), respectively. $J_{avg\_cor}$ are close across the three Apollo sites, particularly for Apollo 11 and Apollo 17 (both around 0.23). However, Apollo 16 exhibits a slightly lower $J_{avg\_cor}$ than the other two, which can be attributed to its location in the Descartes Highlands near the Descartes Magnetic Anomaly—an area where magnetic field anomalies reduce $J_{avg\_cor}$[25,67]. Additionally, $E_{f,avg\_cor}$ at Apollo 16 is significantly lower than that at Apollo 11 and Apollo 17, attributed to $E_{f,avg}$ being more sensitive to magnetic anomalies than $J_{avg}$, as previously mentioned. The latitudinally corrected average SW fluxes on the lunar near side are overall significantly lower than those on the farside, with this asymmetry primarily attributed to the shielding effect of Earth's magnetosphere on SW particles incident on the near side. This is also evident from comparisons of $J_{avg\_cor}$ and $E_{f,avg\_cor}$ values at the CE-5 and CE-6 sampling sites. The ($J_{avg\_cor}$, $E_{f,avg\_cor}$) values for CE-5 and CE-6 are (0.2640, 0.2443) and (0.3157, 0.3090), respectively. The average latitudinally corrected SW fluxes at the CE-6 sampling site are greater than those at CE-5.

### Influence of solar wind irradiation on SW-damaged rim growth rates

Based on the measured SW-damaged rim growth rates ($k$) and the simulations, we constructed correlation plots between average SW fluxes (the average normalized SW number flux $J_{avg}$ and energy flux $E_{f,avg}$) and rim growth rate $k$ across these sampling sites (Fig. 3b, d). Excluding the CE-5 site, the remaining four sites appear to exhibit a positive correlation between $k$ and both $J_{avg}$ and $E_{f,avg}$ with Pearson's correlation coefficients of 0.7758 and 0.7260 ($n = 4$), respectively. However, the corresponding $P$-values (0.2242 and 0.2740) indicate the correlations do not reach statistical significance at the conventional threshold ($p < 0.05$). This is primarily attributed to two factors: limited sample size due to the small number of sampling sites, and high data-point clustering caused by the geographic proximity of Apollo sampling sites, which leads to similar rim growth rates. Notably, the CE-6 sample from the mid-latitude lunar farside exhibits a relatively higher $k$ than the low-latitude nearside Apollo samples. This trend aligns with the hypothesis that shielding by Earth's magnetosphere reduces SW exposure on the lunar nearside, potentially giving rise to a persistent hemispheric asymmetry in SW irradiation. Nonetheless, this interpretation remains provisional, as alternative environmental and structural factors—such as surface temperature variations (which may affect annealing efficiency), as well as local topography and regolith porosity (which may modulate SW irradiation or implantation conditions)—may also contribute to the growth rate of SW-damaged amorphous rims and cannot be definitively ruled out.

Elevated proton-dominated SW irradiation enhances particle bombardment on mineral surfaces. Upon penetration, low-energy protons ( ~ 1 keV) primarily lose energy through electronic stopping—namely, ionization and excitation of target electrons[68]. The deposited energy can break chemical bonds and generate localized electron-hole pairs. Non-radiative recombination of these excitations or energy transfer via localized electronic excitations may induce atomic displacements, particularly in low-binding-energy lattice environments. Although such displacements are typically isolated, continuous irradiation leads to the accumulation of point defects such as vacancies and interstitials. These defects may further agglomerate into clusters or form interconnected networks, especially when the rate of damage accumulation exceeds the material's intrinsic annealing capacity, which is governed by defect mobility and recombination efficiency under ambient conditions. In parallel, a smaller but non-negligible fraction of the proton's energy is transferred directly to lattice atoms through nuclear stopping. These direct collisions initiate displacement cascades, producing vacancy–interstitial pairs that further contribute to lattice damage. While each cascade event is localized, their cumulative effect

enhances the density and spatial connectivity of defects. With continued irradiation, the synergistic interplay between electronically and collision-induced defects drives the progressive disordering of the crystalline lattice, eventually facilitating cooperative atomic rearrangements and the formation of an amorphous phase in the near-surface region. Our Stopping and Range of Ions in Matter (SRIM)[69] simulations show that higher-energy ions penetrate more deeply into the target material and exhibit elevated vacancy generation rate (Table. S3). When SW irradiation intensifies–particularly through simultaneous increase in both particle number and energy–the combined effects typically accelerate the formation rate of amorphous layers on mineral surfaces and extend the amorphization depth.

Additionally, the anomalously high growth rate of the SW-damaged rim observed in CE-5 regolith grains, if confirmed, may suggest that this region has experienced stronger SW irradiation than previously expected. Xu et al.[70]. reported a significantly higher abundance of SW-derived water in CE-5 samples compared to Apollo samples, attributing this enrichment to the lower surface temperatures at the CE-5 site, which suppress SW-hydrogen outgassing. While this thermal effect likely plays a role, evidence from multiple studies suggests that the amorphous layer in minerals acts as the primary reservoir for SW-derived water[70-73]. Bradley et al.[71] used transmission electron microscopy (TEM) and valence electron energy-loss spectroscopy (VEELS) to examine the amorphous rims of interstellar dust particles (IDPs), revealing that SW protons interact with oxygen in silicate minerals to form hydroxyl (−OH) and/or molecular water ($H_2O$), which are subsequently stored within vesicular structures in the amorphous rims. Their findings further demonstrated that SW irradiation induces structural disorder, potentially facilitating both the formation and retention of water. In a separate study, Zeng et al.[72] conducted ion implantation experiments using deuterium ions as analogs for SW protons to irradiate olivine, pyroxene, and plagioclase. They found that over 73% of the implanted deuterium was retained within a fully amorphous rim ~70 nm thick, whereas only ~25% extended to depths of ~190 nm, where amorphization was incomplete. Similarly, Zhou et al.[73] investigated the depth distribution of SW-derived water in CE-5 mineral samples and found that it is predominantly hosted within amorphous rims, with only a minor fraction preserved in the underlying crystalline matrix. As discussed above, enhanced SW irradiation can facilitate the formation of amorphous layers that act as primary reservoirs for implanted water. Therefore, the enhanced growth rate of the SW-damaged rim at the CE-5 sampling site, relative to Apollo sites, may represent an additional contributing factor to the higher SW-derived water content observed in CE-5 materials.

We note that an enhanced growth rate of the SW-damaged rim at the CE-5 sampling site does not necessarily imply an increase in the incident SW flux in space, as local factors may contribute to enhanced effective SW irradiation at the sampling site. One plausible factor involves regolith porosity, which likely modulates SW–grain interactions at the microscale. Simulations by Szabo et al.[74] demonstrated that highly porous regolith enhances ion scattering and increases the spatial distribution of ion–grain interactions. Such structures may allow SW ions to penetrate deeper into the near-surface region and interact with a larger number of mineral grains, effectively increasing the local exposure of grain surfaces to incident particles. This elevated interaction density enhances the accumulation of irradiation-induced damage at grain boundaries, thereby promoting faster formation and thickening of SW-damaged amorphous rims over time. These simulation results are in line with the findings of Hu et al.[75], who reported that the in-site porosity at the CE-5 site is about 10% higher than that at CE-6, despite sample-based analyzes suggesting otherwise—a discrepancy that likely results from mechanical disturbance during sample return. This elevated porosity may have contributed to the

enhanced development of SW-damaged rims at the CE-5 site. It could also help explain the observed differences between CE-5 and Apollo sites.

Another contributing factor to the enhanced effective SW irradiation at the CE-5 site could be the local surface slope. Sim et al.[76] showed that SW−induced space weathering varies with crater wall orientations, consistent with slope-related effects. A steeper slope can effectively reduce the solar zenith angle and enlarge the projected area facing the incoming SW, thereby increasing the effective flux of incident particles. While terrain-related enhancement does not alter the incident SW flux itself, it can increase the effective flux received by sloped surfaces, thereby elevating energy deposition onto individual grains and accelerating the formation of amorphous rims. This mechanism may partly account for both the increased growth rate of SW-damaged rims and the higher SW-derived water content observed in CE-5 samples relative to those from the Apollo missions. It may also contribute to the higher rim growth rate at CE-5 compared to CE-6. Notably, due to severe erosion of the lunar surface at the CE-5 and CE-6 sampling sites by the landers' engine plumes[77–81], precise local topographic slope data prior to plume-induced erosion are currently unavailable.

In addition, local thermal environments may influence the efficiency of amorphous rim formation by modulating the accumulation of SW irradiation-induced lattice damage. Although CE-5 and CE-6 are located at similar latitudes and share nearly identical surface temperature conditions−average (-226 K), maximum (-361 K), and minimum (-89 K)−Apollo sites tend to experience slightly higher maximum (-384–396 K) and minimum (-93–97 K) temperatures[82]. While the thermal sensitivity of amorphous rim formation remains poorly constrained, elevated temperatures could partially anneal irradiation-induced defects before they accumulate to the threshold needed for amorphization. Experimental studies demonstrate the critical role of temperature in mediating the balance between irradiation-induced lattice damage and thermal annealing[83]. At keV ion energies, temperature significantly affects defect dynamics. In Ge crystals irradiated with 25–600 keV ions, amorphization proceeds more readily at cryogenic than at room temperature, underscoring the role of thermally activated defect recombination in mitigating lattice disorder[84]. At higher ion energies (MeV scale), similar thermally driven recovery processes are observed. For example, in $Fa_{12}$ olivine, the threshold for amorphization under heavy-ion irradiation increases markedly near -500 K, indicative of the onset of dynamic annealing processes[85]. The surface temperatures at the CE-5 and CE-6 sampling sites are generally lower than those at the Apollo sites, which may be more conducive to the growth of SW−damaged amorphous rims. However, when considered alongside the influence of regolith porosity, the combined effect of temperature and porosity remains poorly constrained. In addition, the potential long-term impact of thermal cycling on the lunar surface is not well understood. Both aspects warrant further investigation.

In summary, while regolith porosity appears to be one plausible driver of the anomalously rapid growth of SW-damaged amorphous rims at the CE-5 site, potential contributions from the local surface slope and local thermal environment cannot be entirely ruled out. Other contributing factors may also exist, but yet to be identified. Despite these efforts, current interpretations remain qualitative and largely inferential. Quantitative investigations are required to disentangle these factors and to clarify the physical drivers underlying the observed enhancement.

## Methods

### Samples
As mentioned earlier, although samples of both CE-5 and CE-6 missions originate from mare regions, their characteristics exhibit certain differences. The average true density of CE-5 samples is measured as 3.1952 g/cm³ [54], higher than that of CE-6 samples (3.035 g/cm³)[57]. In terms of chemical composition, compared to CE-5 samples, CE-6 samples exhibit a decrease in FeO (CE-5: 22.5 wt.%, CE-6: 17.2 wt.%), $TiO_2$ (CE-5: 5.0 wt.%, CE-6: 2.7 wt.%), but an increase in $Al_2O_3$ content (CE-5: 10.8 wt.%, CE-6: 14.3 wt.%). This is attributed to the fact that CE-6 samples have extremely low olivine content (CE-5: 5.7 wt.%, CE-6: 0.5 vol%), less ilmenite content (CE-5: 4.5 wt.%, CE-6: 1.6 vol%), and higher plagioclase content (CE-5: 30.1 wt.%, CE-6: 32.6 vol%)[54,57]. The nomenclature for CE-5 and CE-6 lunar samples adheres to the established protocols, which are publicly accessible at: https://moon.bao.ac.cn/moonSampleMode/index.html.

In the samples from CE-5 and CE-6, EDS results (Table S2) indicate that the feldspar is predominantly composed of the two calcium-rich varieties: bytownite ($Na_{0.3}Ca_{0.7}Al_{1.3}Si_{2.7}O_8$) and anorthite ($CaAl_2Si_2O_8$). The differences in feldspar composition within the CE-5 and CE-6 samples are minimal, primarily reflected in the mineral proportions (CE-5: bytownite 90%, anorthite 10%; CE-6: bytownite 50%, anorthite 50%) and average compositions (CE-5: An range 76.1–97.6, bytownite average composition An84.5Ab14.6Or0.9, anorthite average composition An92.5Ab7.3Or0.2; CE6: An range 82.9–99.7, bytownite average composition An86.6Ab12.9Or0.6, anorthite average composition An94.3Ab5.4Or0.3). In contrast, the Apollo samples exhibit calcium enrichment, with An ranging from 80.5 to 95.7. Additionally, SRIM[69] simulations of H and Fe ion implantation in anorthite and bytownite (Table S4) show that both ions produce nearly identical damage depths in the two minerals. Overall, the feldspar in the CE-5 and CE-6 samples is highly similar to that in the Apollo samples, demonstrating significant comparability.

### Preparation of FIB cross-sections
The FIB cross-sections of CE-5 and CE-6 soil grains were prepared by the FEI Scios Dual-Beam Focused Ion Beam/Scanning Electron Microscope (FIB/SEM) at the Institute of Geochemistry, Chinese Academy of Sciences. Initially, platinum was applied to coat the regions of interest on the lunar soil grains. Next, a gallium ion beam at 30 kV/3–5 nA was used for digging. Finally, the preliminary sections were polished using a gallium ion beam set to 5 kV/48 pA and 2 kV/43 pA. All FIB sections were -100 nm thick.

### TEM and STEM observations
The prepared FIB sections of CE-5 and CE-6 lunar soil grains were analyzed at the Suzhou Institute of Nano-Tech and Nano-Bionics (SINANO), Chinese Academy of Sciences, by using High-Resolution Transmission Electron Microscopy (HRTEM) and Scanning Transmission Electron Microscopy (STEM) imaging with FEI Talos F200X. Images of the FIB cross-section were captured within a field of view limit of 1 μm. The image scale is 1024 × 1024 px, corresponding to 1 μm².

### Calculation of exposure age
Using the calibration by Keller et al.[7], the track production rate at a depth of 0.1 μm is given as $R_{0.1} \pm \Delta R_{0.1} = (4.4 \pm 0.4) \times 10^4 \, tracks/(cm^2 \cdot yr)$. The exposure age of the soil grain ($t_{expo}$, in years) was calculated from the corresponding track density ($\rho_{0.1}$, in $tracks/cm^2$) using the following equation:

$$t_{\text{expo}} = \frac{\rho_{0.1}}{R_{0.1}} \qquad (1)$$

The uncertainty in the exposure age ($\Delta t_{expo}$), expressed in years, can be estimated as:

$$\Delta t_{\text{expo}} = \sqrt{\left(\frac{\Delta \rho_{0.1}}{R_{0.1}}\right)^2 + \left(\frac{\rho_{0.1}}{R_{0.1}^2} \Delta R_{0.1}\right)^2}, \qquad (2)$$

where $\Delta\rho_{0.1}$ denotes the error in track density at a depth of $0.1\,\mu m$, and $\Delta R_{0.1}$ represents the uncertainty in the track production rate, taken as $4.0\times10^3\,tracks/(cm^2\cdot yr)$.

## Data availability

The source data generated in this study have been deposited in the Mendeley Data under accession code "https://data.mendeley.com/datasets/xn73vjk95x/1".

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

## Acknowledgments

We express our gratitude to all the scientists and engineers who made significant contributions to the Chang'E-5 and Chang'E-6 missions. Additionally, we extend our sincere appreciation to the China National Space Administration for providing us with the allocated Chang'E-5 and Chang'E-6 samples. We are also grateful to the Institute of Geochemistry, Chinese Academy of Sciences, for providing the FIB testing equipment, and the Suzhou Institute of Nano-Tech and Nano-Bionics (SINANO) for providing the STEM testing equipment. This research was supported by the National Natural Science Foundation of China (grant No. 12461160265 [Y.X.]) and the Science and Technology Development Fund (FDCT) of Macau (grant Nos. 0014/2022/A1 [X.Z.], 0034/2024/AMJ [X.Z.], 0021/2024/RIA1 [Y.X.], 0158/2024/AFJ [Y.X.], 0008/2024/AKP [Q.Z.], 0052/2024/RIA1 [X.Z.], 002/2024/SKL [Q.Z.]).

## Author contributions

X.Z. conceptualized this study. R.L., S.Z., and X.Z. contributed to the conception and design of the overall project. R.L. wrote the original draft. X.Z., P.L., Y.X., and C.S. were involved in writing, reviewing, and editing. The investigation was conducted by R.P., C.L., Y.L., X.L., L.X., Z.M., Q.Z., and C.W.

## Competing interests

The authors declare no competing interests.
