## [Transparent Peer Review file · Nature Communications]

Million-year Solar Wind Irradiation Recorded in Chang'E-5 and Chang'E-6 Samples

Corresponding Author: Professor Xiaoping Zhang

Version 0:

Reviewer comments:

Reviewer #1

(Remarks to the Author)

This manuscript investigates the asymmetries in solar energetic particles and solar wind radiation of the lunar farside and nearside using the Chang'E-5 and Chang'E 6 returned samples. They find that the Earth's magnetotail can protect the Moon from the influence of solar wind particles at low energy to some extent, but cannot shield solar energetic particles. This study will be beneficial to gain a deeper understanding of the plasma environment of the Moon and may reveal the evolution of the Earth under the influence of solar activity. This topic is very important and the results look very interesting. I highly recommend this article be published in this journal.

However, I have a major concern about some of the main results and statements presented in the manuscript. In my mind the following issues should be addressed before publication.

In Introduction, the authors have provided a brief, but relatively thorough, review of the past observations of solar particle radiation. However, lunar surface space weathering processes also includes other particle radiation such as GCR, Earth wind, and micrometeorite impact. Could the author introduce more about these processes?

Mesick, K. E., Feldman, W. C., Coupland, D. D. S., and Stonehill, L. C. (2018). Benchmarking Geant4 for Simulating Galactic Cosmic Ray Interactions Within Planetary Bodies. *Earth Sp. Sci.*, 5(7), 324–338. <https://doi.org/10.1029/2018EA000400>

Wang, H. Z., Zhang, J., Shi, Q. Q., Saito, Y., Degeling, A. W., Rae, I. J., et al. (2021). Earth wind as a possible exogenous source of lunar surface hydration. *The Astrophysical Journal*, 907(2), L32. <https://doi.org/10.3847/2041-8213/abd559>

Line 105-108: "In the CE-5 (CE5C1000YJFM006, CE5C0400YJFM00505 and CE5C0600YJFM00304) and CE-6 samples, we selected grains exhibiting surface exposure features such as micro-meteorite impact craters or melt splashes, indicating prior exposure on the lunar surface and thus recording SW and SEP irradiation damage."

Micro-meteorite impact can also affect the lunar soil grain. How do the authors distinguish the SW irradiation damage and the micro-meteorite damage of lunar soil grain in FIB section and STEM images?

Line 112: The research utilizes FIB sectioning and STEM analysis for examining solar wind damage and SEP tracks. Could the authors provide more details on the limitations or potential sources of error associated with these techniques, particularly concerning the accuracy of track density measurements and their impact on the calculated exposure ages?"

Line 117: Figure 2: Could the authors interpret why the CE5-03 sample shows significant abnormalities compared to other returned samples? And Figure 2b doesn't seem very necessary.

"primarily" in sentence at Line 170-172: "According to simulations based ... ~3 MeV/nucleon." seems confusing. Only when the composition of the lunar soil and the type of incident ions fixed, the range of ions in the lunar soil has a clear relationship with the ion energy. In addition to SEP, the lunar soil is also bombarded by GCR and its secondary particles with high energy, which may produce tracks in the lunar soil as well. How about the contribution of these particles? I suggest more words are needed to explain why the tracks in the lunar soil are mainly contributed by Fe ions.

Table 1 started from Line 209: The grain CE5-03 may have experienced a heating event (Line 192), resulting a much higher SW-damaged growth rate (40.55) at CE5 site than other sites in Table 1. It should be excluded as outlier (Line 202), and calculate SW-damaged growth rate of CE5 site using CE1-01 and CE5-02 grains so as to show that the growth rates of SW-damaged rims at all four sampling sites are consistent (Line 206).

The paragraph started from Line 248:

1) As mentioned in Line 226, the SW number flux in Table 1 takes account of the influences of latitude and Earth's magnetosphere. Therefore, in order to study the shielding effect of Earth's magnetotail on solar particle radiation recorded by lunar samples, the two sites for comparison should be chosen at the same latitude, one from nearside and the other from farside; it seems that the CE5 (43.06 N) and CE6 (41.64 S) sites are more suitable than Apollo 11 (0.67 N)/CE6. However, as shown in Table 1, the SW number flux of CE5 site at nearside is lower than that of CE6 site at farside due to shielding effect of Earth's magnetotail, which suggests that the damage rate to CE5 samples by SW should be lower than CE6 as well. This is contradictory to the statement in Line 222, "the damage rate to the lunar soil by SW on the nearside and farside of the Moon seems to be consistent". Could the authors clarify this?

2) As mentioned in Line 45, "in a typical 28-day lunar orbit, the Moon spends approximately 25% of its time (~7 days) within Earth's magnetotail", in which shields solar wind from the lunar nearside. Therefore, the average SW number flux of the same latitude (e.g., equator) on the nearside is ~25% lower than that on the farside (See Apollo 11 and Farside subsolar point in Table 1); Meanwhile, on the farside, the SW number flux ratio between CE6 and Farside subsolar point is $\cos(41.64)/\cos(0)=0.747$, indicating that SW number flux at CE6 site is also ~25% lower than that at Farside subsolar point. So that's why the average SW number fluxes at Apollo 11 and CE6 sites are almost the same value (See Table 1), it is caused by the influences of both latitude and Earth's magnetosphere (Line 226).

3) Micrometeorite impact is another process that can produce amorphous rims on lunar soil grains, although this process cannot be affected by shielding of Earth's magnetotail. This might produce consistent damage rate to the lunar soil on the lunar nearside and farside. This should be discussed.

Line 286: Consistent SEP spectra are recorded in samples, is there any other evidence? Since that SEP events are inherently random and their flux can vary significantly, how do the authors justify the conclusion of consistent SEP spectra over million-year timescales? Could there be unaccounted factors or biases in the data that might affect the long-term average of SEP radiation doses?"

Lines 275-281: Could the authors clarify the relationship between the possible trend of weakening solar activity and the Earth's Ocean temperature?

Reviewer #2

(Remarks to the Author)

The manuscript by Liu et al. presents the results of the measurements of the SW and SEP particle tracks in lunar samples from the far side of the Moon, collected by the Chang E-5/6 missions, followed by a discussion. While the measurements are very interesting, the discussion is partly unfocused, highly speculative and ungrounded. This reviewer recommends a major revision of the manuscript in the discussion part in line with a more detailed comment below.

The discussion includes two different phenomena: solar wind (SW) and solar energetic particles (SEPs).

The SW-related analysis and discussion are based on a comparison of the experimental data with model simulations and convincingly imply the important role of the SW particle flux vs. energy flux as well as a notable effect of the Earth's magnetosphere on the lunar surface irradiation by SW. This part looks good, is sufficient to warrant the publication, and requires only a minor revision as suggested below in greater detail.

On the contrary, the discussion on the SEP flux is very speculative, lacking any quantitative or modelling ground and leads to controversial results. This part needs to be either greatly improved in providing quantitative modelling estimates or be totally removed.

More detailed comments are listed here.

1. The title is unclear and imprecise: It is unclear from the title what the main result is; SW should be explicitly mentioned; the results are based not only on E6 but also on E5 Chang missions.

2. In the discussion about SEPs, the authors may want to mention that there is another method to reconstruct the SEP spectrum from lunar samples on the mega-year timescale, based on the cosmogenic isotopes (see, e.g., Reedy and Arnold, 1972, doi: 10.1029/JA077i004p00537; Poluianov et al., 2018, doi: 10.1051/0004-6361/201833561). It would be useful to compare the results based on these two methods. Usoskin et al. (2023, doi: 10.1051/0004-6361/202245810) have shown that the SEP average flux reconstructed from lunar isotopes is nearly doubled compared to the present-day one and supports the existence of rare but extremely strong solar eruptive events.

3. The nomenclature of the samples needs to be specified and explained, probably in the Methods section. For example, notations in line 105 are unclear for non-experts. Also, full information on the samples should be given (or referenced to) including the chemical composition and density profile.

4. More details on the grain structure need to be given. For example, is erosion important and if yes, how is it accounted for? Can the SEP-related tracks be distinguished from the GCR and ACR ones?

5. Fig.3 requires a more detailed discussion. What causes the large difference in track density between different samples? Even arguing that CE5-03 is an outlier, the factor 2—3 difference still remains between samples. It should be explained. As discussed below, the explanation for the different slopes looks unconvincing.

6. In lines ~160, the authors discuss that the different slopes indicate harder or softer SEP spectra. This must be quantified

via a model. What do harder or softer spectra mean – what energy spectrum? How are they related to the present-day SEP spectrum? What slope would have corresponded to the present-day SEP spectrum? Could other interpretations be proposed – e.g., that the higher-slope samples correspond to an inclined (not horizontal) or partly obstructed surface? Are the densities of all samples the same? Different densities can lead to different slopes per μm .

7. In lines 177+, the authors say that their result proves that the effect of the Earth's magnetotail on SEP is negligible. While I agree with this statement (in particular because SEPs are nearly isotropic in space), I don't see how it is proven here.

8. The statement in lines 192+ is very speculative. While the effect of annealing sounds plausible, it is not supported by any data or modelling. It would be fairer to say that the sample CE5-03 is an outlier for an unknown reason.

9. Line 214: shouldn't the energy flux be proportional to $N \cdot V_r \cdot V^2$ rather than $N \cdot V_r^3$?

10. The entire discussion in the section entitled "Ancient solar activity and its impact on Earth" is only an ungrounded speculation and must be removed or fully elaborated. It is all based on the fact that two samples have a higher slope than three others. Despite the doubtfulness of this result (see item 6 above), the discussion is not supported by the data here. In lines 264+, the authors speculate that something weird had happened to solar activity about 2 Myr ago which had led to the change of the slope. However, as seen in Fig.2, the exposure ages of higher-slope samples 64455 (2 Myr) and CE5-01 (1.5 Myr) are not systematically different from smaller-slope samples (1–2.7 Myr). Thus, this speculation is not supported by the data. The authors propose a declining trend of solar activity, but this is not supported by any data or models or speculations. The authors may confuse it with the ageing of the Sun which was indeed much more active in its younger age, but this refers to Gyr scales, not Myr.

11. The discussion about climate (lines 275-281) should be removed. It is irrelevant to this work and grossly incorrect. Reference 39 is known to be incorrect, references 40–42 do not apply here.

Reviewer #3

(Remarks to the Author)

In the present manuscript titled "Long-term Solar Particle Radiation Records on the Lunar Farside from Chang'E-6 Samples", the authors Liu et al. present the first measurements of solar particle radiation damage in samples returned from the lunar farside. In particular, they use established transmission electron microscopy methods to characterize amorphized rims from continuous solar wind exposure and tracks created by heavy solar energetic particles (SEPs) that have been used to date the exposure age of lunar samples. The authors analyze three samples from a previous Chang'E-5 sample return and two grains from the new Chang'E-6 mission to compare rim thickness and track densities to previous analysis of Apollo samples. The authors declare a consistent ratio of rim growth and track accumulation for the different samples, from which they extrapolate an estimate of 25% solar wind shielding of the lunar nearside due to different sampling locations. Regarding the track accumulation, they find two distinct sample groups of different slopes in track density over depth. From this relation, the authors speculate a more energetic SEP spectrum for SEP exposure older than 2 million years ago, which they state could contribute to declining ocean temperatures.

Analysis of lunar regolith grains that were retrieved from the far side is highly interesting to provide new insights into the interaction of the lunar surface with its space environment. The present study is the first that attempts to perform such an analysis, due to the recent sample return from the Chang'E-6 mission. In this context, the present study by Liu et al. uses well-suited and established methods for studying the damaging of lunar regolith grains by solar radiation. The evaluation of the authors' TEM measurements to extract rim thicknesses and track densities is appropriately performed and experimental observations such as the different track-over-depth slopes are novel and interesting. However, in their discussion the authors rely on incomprehensible reasoning, inconsistent statistical arguments, and extensive speculation to arrive at the conclusions outlined in their abstract. Ultimately, the limited number of studied samples and missing analysis of sample properties key to the interpretation do not support several of the authors' claims (see detailed comments below). Due to these fundamental issues, I am afraid that I must recommend a rejection of the paper. However, I strongly encourage the authors either to extend their sample analysis or to rework the interpretation of their results as some of their experimental findings would certainly be interesting to the community.

My main point of criticism is that the authors' conclusions regarding rim growth rate versus damage rate do not appear warranted. The slope of SW-damaged growth of the two CE5 samples (in Fig. 2a) is around 40 nm/Myr compared to 23.5 – 26.3 nm/Myr for the other sample groups. The standard deviations for both CE-5 and CE-6 sample groups are around 6 nm/Myr. Even though the CE-5 slope is almost twice as high as the CE-6 slope, the authors declare the difference statistically insignificant because it falls below two sigma (The difference between the two, 14.25, is actually slightly higher than two standard deviations of the uncertainties of the respective slope values for CE-5, 5.82, and CE-6, 5.74). Thus, the authors state a consistent rim growth rate for all samples and base several other arguments on this relation - however, without applying the same scrutiny of needing at least a two-sigma significance (see below).

The conclusion of equivalent solar wind damage rates has several other problems. CE6 samples lie within the data spread of Apollo samples, but CE5 appear as a higher group (Figure 2a). Due to the very low number of samples (2 or 3 depending on whether CE5-03 is excluded), it is difficult to judge whether there is a systematic difference in slope in Fig. 2a for the CE-5 group. The authors argue for a consistent relation across all sample groups by excluding CE5-03 from their analysis as an outlier. The omission of CE5-03 is based on the authors speculating that a heating event happened, which erased tracks due to annealing but left the solar-wind damaged rim intact. However, neither are any experimental signs of annealed tracks reported, nor do the authors present any references supporting that a heating event would erase only tracks but not the rim. The number of other samples in the CE-5 group is also too low to really support CE5-03 fully being a statistical outlier. Inclusion or exclusion of CE5-03 has a big effect on the finally derived value of the slope k of the rim growth rate. An additional issue is that the exclusion of CE5-03 leaves the authors with only two samples for CE-5 and CE-6, respectively. This represents a very small number of samples considering the scattering of the rim thickness vs. track density

data in Figure 2a. Error bars are given for the CE-5 and CE-6 samples, but the general scattering of past measurements is not discussed. It is not clear if new analysis is improved over the past measurements of Apollo grains or if the scattering of single grains is a general expected feature because of different grain-specific solar wind exposure history. In the latter case, a reliable identification of the rim growth rate from just two samples is not possible. Furthermore, the track growth rate fits for different sample groups (CE-5, CE-6) clearly have different, non-zero intercepts, which is not discussed in the paper. The authors argue CE-6 and CE-5 samples to show the same behavior, but at a similar exposure age, CE5-02 and CE6-02 have a rim thickness difference by about a factor of 2 (or several standard deviations of the single measurements).

In their abstract, the authors describe shielding of the lunar near side of 25%. This number aligns with known lower plasma exposure of the lunar surface in the earth's magnetotail, as for example in detail characterized by the THEMIS-ARTEMIS mission. The number of 25% is also not an experimental finding, but an estimate from simulations from previous work by Xie et al. based on the consistent damage-growth rate k between farside samples from CE-6 and other sample groups. As discussed above, there are significant issues with the authors' argument of consistent rim growth rates. However, the authors are also not consistent within their own analysis: Given the stated k -value for Chang'E-6 (derived from only two samples) of 26.30 nm/Myr with a standard deviation of 5.74 nm/Myr (21.8%), it seems questionable that differences on the order of 25% would be detectable if present – especially when considering the above-mentioned argument of two-sigma significance.

The different slopes of track density over depth presented in Figure 3 are quite interesting and to my knowledge, this has not been reported before. The authors connect these slope differences to a different hardness of the SEP spectrum, potentially related to higher solar activity longer in the past than 2 Myr ago. An assumption on a different SEP exposure is not unreasonable, but the authors are not able to provide any evidence for the grains from Group 1 being exposed to SEPs at different times than those from Group 2. Conclusions on different solar activity at different times (and especially when a solar activity decrease would have occurred) cannot be made with the results presented in this study. Any connection to Earth's ocean temperatures in this context is completely speculative.

Minor comments:

- Lines 51 & 177: The authors should cite recent reports from the ARTEMIS spacecraft that have shown SEP access to the Moon even deep in the magnetotail (<https://doi.org/10.1029/2023GL103990>, <https://doi.org/10.1029/2024GL110228>).
- Line 105: In comparison to the description of the Chang'e-6 samples, context on Chang'E-5 mission and sampling location are missing. It should furthermore be described in more detail that the Chang'E-5 samples also represent newly analyzed samples specifically for this paper.
- Lines 224-225: The wording needs to be changed to explain more clearly what the authors want to say.
- Lines 231-234: The authors should reference studies from the Chandrayaan-1 spacecraft using H reflected from the lunar surface, which provide evidence for the extent of shielding at some magnetic anomalies (<https://doi.org/10.1029/2012JA017553>).
- Lines 235-237: Some context on the simulation work by Xie et al. is missing here. Which aspects of the model have been verified with measurements? Is there any way to estimate an uncertainty for the used number fluxes and energy fluxes? Especially regarding the previously mentioned issues with the uncertainties of the measurement results and the related conclusions, this would be helpful.
- Lines 240-247: As mentioned before, the ratio of rim growth slope to J_{avg} is almost twice as high for the CE5 samples. J_{avg} is actually the lowest for the CE5 sampling site.

Version 1:

Reviewer comments:

Reviewer #1

(Remarks to the Author)

The revised manuscript has addressed most of my concerns satisfactorily. The author has provided detailed and clear responses to all inquiries, carefully considered all suggestions, and made comprehensive revisions to the manuscript. By conducting additional experiments, the author has significantly strengthened the reliability and robustness of the results. The revised version now presents the findings and conclusions in a more logical and convincing manner. Therefore, I recommend that this article be accepted for publication.

Reviewer #2

(Remarks to the Author)

The paper has been improved by the authors in the process of revision and is now closer to acceptable quality. However, a further revision is still needed as detailed below.

Several ungrounded speculations caused by over-interpretation of the results should be removed:

- * The paragraph about the correlation (lines 203+) should be removed as misleading. First, the CE5 point is identified as an

outlier (l. 210) and should not be used in the correlation analysis. Second, the remaining 4-point correlation is, while yielding a high formal value of the Pearson's linear correlation coefficient, insignificant, viz. the p-values for the points shown in Fig.3b and 3d are 0.09 and 0.08 (formally insignificant). Moreover, these correlations are mostly based on the single CE-6 point, which lies higher than the Apollo-based points and has large error bars. If the CE-6 point is excluded, the remaining Apollo points don't exhibit any significant trend. Thus, the claimed relation is not well-established.

* The last paragraph (l. 229+) also needs to be removed. The CE-5 point is identified as an outlier, and thus, no conclusions can be drawn based on its difference from other points. The conclusion that the present solar wind models are incorrect in that the model computations correctly predict all other data points but the CE-5 one, which is an outlier. Accordingly, the last two sentences of the Abstract should be removed, too.

The authors overuse self-citations. For example, the first 5 references should be related to the process of solar wind energetic particle acceleration, but instead cite publications about measurements at the lunar surface. General references to solar energetic particles and solar wind are missing. The following ones are recommended: Desai & Giacalone (2016, doi: 10.1007/s41116-016-0002-5) and Vidotto (2018, doi: 10.1007/s41116-021-00029-w).

Other comments are minor textual corrections.

* Line 114-115: "flare tracks"  "particle tracks".

* Line 140+: Solar activity is not expected to vary on the mega-year time scale. It varies on the synoptic through secular timescales (solar variability) and also on the giga-year scale (solar evolution), but not on mega-year scale. There are numerous pieces of evidence that solar activity is roughly stable over millions of years, including lunar samples. This needs to be re-formulated.

* Line 159: "the absolute difference ... within 10%". If it is given in %, it must be the relative difference.

Reviewer #3

(Remarks to the Author)

In their revised manuscript, Liu et al. present a largely reworked paper on electron microscopy analysis of Chang'e 5 and Chang'e 6 lunar samples. Compared to the previous draft, detailed studies of the solar energetic particle (SEP) spectra are removed as well as any connection to Earth climate studies. Instead, the paper now focuses solely on the formation of weathering features in the lunar regolith across different sampling sites. In order to improve the significance of their findings, the authors have significantly increased the number of studied samples for both Chang'e 5 and Chang'e 6, which is highly commendable. Weathering features are characterized by comparing SEP track densities with the formation of amorphized rims from solar wind (SW) exposure. When compared to Apollo samples, the Chang'e 6 farside samples show a similar weathering behavior, while the Chang'e 5 samples tend to have faster rim formation. This appears to be different with the behavior in other samples, especially as simulations predict lower SW exposure than at other some of the other locations. Compared to the previous version of the manuscript, the methodological aspects of the study are improved. As similarly stated in my previous review, the selection of sample analysis methods is highly appropriate and through the increased number of samples the findings are more robust. The observation that Chang'e 5 rims have a tendency of being formed more quickly is interesting to report to the community and this finding appears robust. Both for amorphous rim and SEP track formation, previous work has presented models that explain these processes qualitatively but not fully quantitative yet. As a result, we do not fully understand the details of the formation processes and why some samples form larger rims than others. The submitted manuscript is also not able to present a more detailed explanation for these processes, and the Chang'e-5 sample group behavior in particular. An extended discussion of the latter would thus significantly improve the paper.

Following points should be cleared up:

- The CE-5 sample group sticks out in covering a narrower exposure age, while both the Apollo 17 and the CE-6 group have multiple samples that fall into the range of the CE-5 samples in Figure 5. Are there systematic differences between sample collection of CE-5 and CE-6 that could potentially explain the smaller CE-5 variety? If not, do the authors have any other explanation for differences in the CE-5 sampling site that could explain a difference in rim formation?
- Did the composition of Chang'e 5 and 6 plagioclase grains differ significantly? Keller et al. (2021) specifically studied anorthite grains, how do the EDS composition results compare here?
- Stating that the simulations of the surface exposure by Xie et al. is consistent with ARTEMIS data within 10% could be misleading. The main result that the present work uses from the simulations is the surface precipitation flux and energy flux, specifically accounting for the effect of local magnetic anomalies. As an orbiter, these effects cannot be directly studied with the ARTEMIS spacecraft, especially considering the effects of local magnetic anomalies.
- It appears to be a minor detail for the study here, but the arguments in the response letter given for the usage of the radial velocity v_r^3 to calculate the energy flux are not fully convincing. Damage theories like the Kinchin-Pease model connect the number of vacancies created by ion impacts to the deposited energy. Simulation programs such as SRIM and SDTrimSP also predict the number of formed vacancies by impacting ions to be largely independent of angle of incidence, but instead the impact energy is important.

Version 2:

Reviewer comments:

Reviewer #2

(Remarks to the Author)

I am fine with the revised version of the paper. Only a few very minor clarifications of the text are requested before the final acceptance recommendation.

1. Line 31: "underlying mechanisms of"  "exact reasons for".
2. Line 39: "During solar activities"  "During solar activity events".
3. Line 234: Please remove "mechanistically".
4. Line 249: "grains, suggests that this region may have"  "grains, if confirmed, may suggest that this region has"

Reviewer #3

(Remarks to the Author)

Regarding the updated version of the manuscript, I appreciate the modifications and the response by the authors, providing some helpful additional background. Unfortunately, there are still some issues both in the old and the newly added text.

Especially arguments in the newly added sections on an increased solar wind flux at the Chang'e-5 sampling site are not convincing. The authors state surface slopes as a potential reason, but do not give any further details, which makes it hard to follow why this would be the case. Such an assumption would need a more quantitative calculation and evidence that relevant conditions are actually present at the Chang'e-5 sampling site to allow increased solar wind fluxes to the surface. Ultimately, the finding that the CE-5 samples are an outlier in the rim growth rate appears robust, but it remains an open question why, which the present study cannot answer at the moment.

The following are my specific remarks:

- Introduction: The newly added section at the beginning is selective to specific events of coronal mass ejections and solar-energetic particle event, while the solar wind is barely mentioned. As solar-wind alteration is the more important topic for this paper, the introduction should be adjusted to reflect this.
- Line 176: As a response to one of my previous comments, the authors added a paragraph about the agreement of the Hall-MHD simulations. However, a statement regarding the agreement with ARTEMIS on precipitation still remains. As stated before, precipitation cannot be directly observed from the ARTEMIS orbiters, which is why the precipitation maps presented in Figure 3 cannot be verified with this spacecraft dataset. The authors should remove this statement or clarify what they want to say here, as well as specify, which part of reference 47 they are citing.
- Discussion: The newly added section, which suggest higher water abundances could possibly be related to higher solar-wind precipitation, appears highly speculative. At two points the authors mention "the elevated SW flux at the CE-5 site" as if this was shown here, but there is no convincing evidence for this to be necessarily the case. Rather it could also be possible that surface effects, such as the mentioned lower temperature, affect both water abundance and rim growth in similar ways, for example through different annealing rates of damaged rims. An extended, more convincing discussion would be needed to support the conclusion that the CE-5 sample site is exposed to an on average higher SW flux.

Further Remarks:

- Abstract: "[...] reveals that CE-6 samples from the southern mid-latitude farside display rim growth rates comparable to or exceeding those of low-latitude Apollo samples. This supports the hypothesis of persistent irradiation asymmetry driven by Earth's magnetospheric shielding of the nearside."
This is inconsistent reasoning, if rates would be comparable then that would mean no asymmetry. This should be adapted.
- Line 64: It should say "matures".
- Line 65: It should be "regolith grains undergo a gardening process".
- Line 81/82: There is a word missing in this sentence.
- Line 85: "throughout the Moon", this wording should be changed
- Line 92+96: It should be "the farside".
- Line 110: "sample is space weathering-induced amorphous rim" There is a word missing here.
- Table S4: This information could be better presented in a Figure, and the μm label for H is wrong (it should be nm).

Version 3:

Reviewer comments:

Reviewer #3

(Remarks to the Author)

The revised version of the manuscript "Million-year Solar Wind Irradiation Recorded in Chang'E-5 and Chang'E-6 Samples" provides updated and extended introduction and discussion sections. I want to commend the authors on the additional work that they have put in this revision and the response to previous questions.

Generally, only few comments from my side remain:

- L94: I think the authors want to say "seemingly only minor variations"
- Line 291: "Therefore, the elevated SW irradiation at the CE-5 sampling site, relative to Apollo sites, [...]" As discussed in the previous version, I still see little evidence for there really being an increased SW precipitation of the Chang'E-5 site. The newly added section on porosity could explain this, but the differences of 10% might still be small. It would be best to adjust the wording here.
- Line 311: Given that the authors state there is no available precise slope data for either of the landing sites, this paragraph appears somewhat loosely connected to the rest of this section. I suggest the authors include, for example, the following reference on slope-related weathering: <https://doi.org/10.1002/2017GL075338>

Response to reviewers' comments for Million-year Solar Wind Irradiation Recorded in Chang'E-5 and Chang'E-6 Samples

AUTHORS: Renrui Liu^{1,2†}, Xiaoping Zhang^{1,2*}, Sizhe Zhao^{1,2†}, Yi Xu^{1,2*}, Yang Li^{3*}, Xiaojia Zeng^{1,3}, Pengwei Luo^{1,2†}, Chenkun Sun^{1,2}, Ronghua Pang³, Chen Li^{3,4}, Xiongyao Li³, Lianghai Xie⁵, Zhiguo Meng^{1,6}, Qiugang Zong^{1,2}, Chi Wang⁵

Affiliations:

¹State Key Laboratory of Lunar and Planetary Sciences, Macau University of Science and Technology, Macau, China.

²CNSA Macau Center for Space Exploration and Science, Macau, China.

³Center for Lunar and Planetary Sciences, Institute of Geochemistry, Chinese Academy of Sciences, Guiyang, China.

⁴School of Engineering, Yunnan University, Kunming, China.

⁵State Key Laboratory of Space Weather, National Space Science Center, Chinese Academy of Sciences, Beijing, China.

⁶College of Geoexploration Science and Technology, Jilin University, Changchun, China.

† These authors contributed equally to this work.

* Corresponding author. Email: xpzhangnju@gmail.com yixu@must.edu.mo
liyang@mail.gyig.ac.cn

Below are point-to-point responses to each of the reviewers' comments. Reviewer comments (black text), Our responses (green text), *Manuscript changes (red italic underlined text)*. The **yellow-highlighted sections** within the *red italic underlined text* specifically indicate the modifications made in response to the comments.

Reviewer #1:

This manuscript investigates the asymmetries in solar energetic particles and solar wind radiation of the lunar farside and nearside using the Chang'E-5 and Chang'E-6 returned samples. They find that the Earth's magnetotail can protect the Moon from the influence of solar wind particles at low energy to some extent, but cannot shield solar energetic particles. This study will be beneficial to gain a deeper understanding of the plasma environment of the Moon and may reveal the evolution of the Earth under the influence of solar activity. This topic is very important and the results look very interesting. I highly recommend this article be published in this journal. However, I have a major concern about some of the main results and statements presented in the manuscript. In my mind the following issues should be addressed before publication.

Reply: We sincerely appreciate your recognition of our work's significance, and the detailed feedback provided, which has been instrumental in refining our manuscript. In the revised version, we have made the following major revisions and additions to address your concerns:

1. Data Section:

- We added new data from CE5 and CE6 samples to improve the reliability of our conclusions.
- Based on the newly added data, we identified that the CE5-03 sample in the original manuscript exhibited a significant anomaly, characterized by extensive melt layer coverage, noticeable annealing features, and non-uniform solar flare track distributions. This anomaly stood out compared to the other data. To ensure the robustness of our analysis, we have removed this outlier in the revised manuscript.

2. Experimental Section:

We added additional quantitative transmission electron microscopy-energy dispersive X-ray spectroscopy (TEM-EDS) composition maps data to better distinguish between the vapor deposited rim and the solar wind (SW)-damaged rim. This distinction allowed us to ensure that the regions analyzed for SW-damaged rim were statistically accurate, thereby improving the reliability of the results.

3. Discussion Section about SW Flux and SW-Damaged rim Growth Rate:

- In the revised manuscript, we have compared the data from CE5, CE6, Apollo 11, Apollo 16, and Apollo 17. The data from CE6, Apollo 11, Apollo 16, and Apollo 17 exhibit a clear positive correlation between SW fluxes (number flux and energy flux) and SW-damaged rim growth rates, indicating that SW flux is the primary factor influencing the growth rate of SW-damaged rim. This further suggests that current observations and simulations are valid for analyzing these regions, and that the Earth's magnetosphere does change the SW irradiation in these regions.

- However, the CE5 region showed an anomalously high SW-damaged rim growth rate, which deviates from current models and observational results. This discrepancy suggests that the existing SW flux distribution models may not explain the conditions in this region. Other space weathering processes may contribute to this anomaly.

4. Discussion Section about Solar Energetic Particles (SEP):

To address the high uncertainty in this section and maintain focus on the SW analysis, we have removed the discussion of SEP energy spectrum in the revised manuscript. This modification also addresses feedback from other reviewers, ensuring the discussion remains robust and focused.

We have addressed your detailed questions point by point as follows.

Q1: In Introduction, the authors have provided a brief, but relatively thorough, review of the past observations of solar particle radiation. However, lunar surface space weathering processes also includes other particle radiation such as GCR, Earth wind, and micrometeorite impact. Could the author introduce more about these processes?

R1: In the revised manuscript, we have added the introduction of other factors related to lunar surface space weathering processes.

Page 2 Lines 64-70: *Lunar space weathering is driven primarily by charged particle irradiation (SW, SEPs, galactic cosmic rays [GCRs]), micrometeorite bombardment, and transient contributions from Earth's magnetospheric particles (Earth wind)¹⁷⁻¹⁹. Under the long-term combined effects of these space weathering processes, lunar regolith particles gradually "mature."²⁰ Through micrometeorite impacts, regolith grains undergo gardening, being repeatedly excavated to the surface and buried underground²¹. Lunar surface grains preserve distinct signatures of local solar wind and SEP irradiation histories.*

References

17. Mesick, K. E., Feldman, W. C., Coupland, D. D. S. & Stonehill, L. C. Benchmarking Geant4 for simulating galactic cosmic ray interactions within planetary bodies. Earth Space Sci. 5, 324–338 (2018).

18. Wang, H. Z. et al. Earth wind as a possible exogenous source of lunar surface hydration. Astrophys. J. Lett. 907, L32 (2021).

19. Quan Qi S. H. I. et al. Review of particle radiation environment of the earth-moon space and its impact on lunar surficial material generation. Chin. J. Geophys. 66, 2685–2702 (2023).

20. Morris, R. V. Surface Exposure Indices of Lunar Soils: A comparative FMR study. Lunar Planet. Sci. Conf. Proc. 1, 315–335 (1976).

21. Morris, R. V. In situ reworking (gardening) of the lunar surface: evidence from the Apollo cores. Lunar Planet. Sci. Conf. Proc. 2, 1801–1811 (1978).

Q2: Line 105-108: “In the CE-5 (CE5C1000YJFM006, CE5C0400YJFM00505 and CE5C0600YJFM00304) and CE-6 samples, we selected grains exhibiting surface exposure features such as micro-meteorite impact craters or melt splashes, indicating prior exposure on the lunar surface and thus recording SW and SEP irradiation damage.” Micro-meteorite impact can also affect the lunar soil grain. How do the authors distinguish the SW irradiation damage and the micro-meteorite damage of lunar soil grain in FIB section and STEM images?

R2: Micrometeorite impacts lead to the formation of impact-induced amorphous rims as well as vapor-deposited rims. For impact-induced amorphous rims, which are located directly below the

micrometeorite impact craters, we did not analyze the micrometeorite impact craters and their surrounding regions. Instead, the study focused on clean regions. For vapor-deposited rim, this kind of rim is enriched in iron or np-Fe_0 . Since plagioclase is a nominally non-iron bearing mineral, we can distinguish vapor-deposited rim from SW-damaged rim by TEM-EDS analysis of iron in uppermost surface. By using this method, we have clarified the characteristics of different types of rims, ensuring that the investigated subjects are SW-damaged rims. In the revised manuscript, we have included full-view images, elemental maps, and high-resolution STEM images of all analyzed particles in the Supplementary Information to ensure data transparency.

Page 3 Lines 109-125:

Fig. 1 The microscopic characteristics of representative studied grains. (a) Secondary electron (SE) image of the studied CE6-PL1 grain. The green rectangle indicates the FIB cross-section site. (b, d) STEM and HRTEM image of the studied grain surface, black dashed line represents the boundary between the amorphous rim and the matrix (c) STEM, quantitative TEM-EDS maps, and corresponding TEM-EDS profiles of space weathering rim. The white arrow and white dashed box indicate the vapor-deposited rim and solar wind (SW)-damaged rim, respectively. Note that FIB, STEM, HRTEM, and TEM-EDS are abbreviations for Focused Ion Beam, Scanning Transmission Electron Microscopy, High-Resolution Transmission Electron Microscopy, and Transmission Electron Microscopy-Energy Dispersive X-ray Spectroscopy, respectively.

In total, seven grains from CE-5 (CE5-01 to CE5-07) and eight from CE-6 (CE6-01 to CE6-08)

were selected for the focused ion beam (FIB) cross-section, respectively. As shown in Fig. 1b-d, the scanning transmission electron microscopy (STEM) and high-resolution transmission electron microscopy (HRTEM) images of the FIB cross-section indicate that the uppermost surface of the sample is space weathering-induced amorphous rim, while the underlying matrix region contains solar flare tracks. The formation of rims and solar flare tracks is closely related to the exposure of samples on the lunar surface. The types of rims are primarily classified as SW-damaged rims and vapor-deposited rims. In all samples, SW-damaged rims were observed. In the CE-5 samples, their thickness ranged from approximately 40-64 nm, while in the CE-6 samples, their thickness varied from around 19 to 109 nm (Figs. S1-15). Quantitative transmission electron microscopy-energy dispersive X-ray spectroscopy (TEM-EDS) composition maps indicate that SW-damaged rim composition is consistent with the matrix (Fig. 1c). However the impact events are episodic, we did not find vapor-deposited rims in all samples. In the six particles containing CE5-PL1, CE5-PL3, CE5-PL7, CE6-PL1, CE6-PL7, and CE6-PL8, iron-rich vapor-deposited rims were observed (Figs. S1, S3, S7, S8, S14, S15). Particle tracks within the matrix were observed in all grains. Grain CE6-PL1 is shown in Fig. 1, with the remaining grains presented in Figs. S1 to S15 of the Supplementary Information.

Q3: Line 112: The research utilizes FIB sectioning and STEM analysis for examining solar wind damage and SEP tracks. Could the authors provide more details on the limitations or potential sources of error associated with these techniques, particularly concerning the accuracy of track density measurements and their impact on the calculated exposure ages?"

R3: Uncertainties in track density primarily stem from statistical limitations in image resolution during experimental measurements. These uncertainties, combined with those in the track production rate, propagate through the exposure age calculation as detailed in Equation (4) of the Methods section in the revised manuscript. We have implemented rigorous measures to minimize systematic uncertainties arising from experiments. (1) We ensured that the thickness of the samples prepared for FIB cross-sections was approximately 100 nm. This parameter has also been validated as optimal by Keller et al. (2021)¹. (2) For rim thickness measurements, considering that the thickness of the rim is not completely uniform, we conducted systematic measurements of clean areas across the entire cross-section and calculated thickness ranges with error margins to ensure statistical robustness and fitting reliability. (3) Observation of SEP tracks: When characterizing SEP tracks using TEM, we utilized parameters that are completely consistent with those of Keller et al. (2021)¹, which have been validated as effective and used in several previous studies^{1,2}. Additionally, to verify our observations, we employed STEM alongside bright-field imaging for cross-validation

to assess whether any tracks were overlooked. (4) Statistical analysis of SEP tracks: All data were repeatedly counted by multiple researchers using varying brightness to determine track density error ranges, thereby ensuring data accuracy. (5) Calculating exposure age: During the calculation of exposure age, we simultaneously considered the errors in track growth rate and track density. Using the error propagation formula, we computed the uncertainty in the exposure age. This approach ensures the accuracy and reliable error estimation of the entire measurement.

References

1. Keller, L. P., Berger, E. L., Zhang, S. & Christoffersen, R. Solar energetic particle tracks in lunar samples: a transmission electron microscope calibration and implications for lunar space weathering. *Meteorit. Planet. Sci.* 56, 1685–1707 (2021).
2. McFadden J A, Thompson M S, Keller L P, et al. Determining surface exposure ages of regolith grains in lunar core sample 73002 by investigating solar particle irradiation damage. *Lunar Planet. Sci. Conf.* 1767 (2023).

Page 3 Lines 103-105: *To allow comparison with previous Apollo findings and minimize the influence of mineral-type variations, we restricted our analysis to plagioclase grains and employed the same experimental methods as in prior studies*^{5,22}.

References

5. *Poppe, A. R., Szabo, P. S., Imata, E. R., Keller, L. P. & Christoffersen, R. Solar energetic particle track-production rates at 1 au: comparing in situ particle fluxes with lunar sample-derived track densities. *Astrophys. J. Lett.* 958, L35 (2023).*
22. *Keller, L. P., Berger, E. L., Zhang, S. & Christoffersen, R. Solar energetic particle tracks in lunar samples: a transmission electron microscope calibration and implications for lunar space weathering. *Meteorit. Planet. Sci.* 56, 1685–1707 (2021).*

Q4: Line 117: Figure 2: Could the authors interpret why the CE5-03 sample shows significant abnormalities compared to other returned samples? And Figure 2b doesn't seem very necessary.

R4: With the addition of substantial new data, we confirmed this grain was an outlier (Fig. R4-1) and therefore removed the corresponding data point from Fig. 2 in the revised manuscript. Additionally, Fig. 2b from the original manuscript has been removed in the revised version. The rationale is as follows:

Fig. R4-1 Comparisons of solar wind (SW)-damaged rim thickness and exposure age in lunar grains from various sampling sites, using the expanded dataset. Notably, even after adding additional samples, CE5-03 remains an outlier.

Compared to other samples, the FIB cross-section of this sample is basically prepared within the pit (Fig. R4-2a). The TEM results indicate that the uppermost surface of this FIB cross-section is basically within the impact-induced amorphous rim, making it hard to distinguish from the SW-damaged rim (Fig. R4-2b). STEM observations revealed that both the density and length of particle tracks decreased markedly toward the central region, exhibiting characteristic lattice annealing features (Figs. R4-2c-e). In the revised manuscript, we refrained from selecting particles with extensive melt coverage and restricted FIB cross-section preparation to areas distant from sputtered melt deposits, thereby ensuring the analyzed regions remain unaffected by thermal events and annealing effects.

Fig. R4-2 Characterization of anomalous grain CE5-03 from the original manuscript: (a) Surface morphology of the anomalous grain. (b) FIB-prepared cross-section through the anomalous region. (c-e) Track density distributions across different regions of the FIB section. Panels b-e collectively show an inverse correlation between track density and proximity to the central pit, with significantly reduced track densities observed in area immediately adjacent to the pit.

Q5: "primarily" in sentence at Line 170-172: "According to simulations based ... ~ 3 MeV/nucleon." seems confusing. Only when the composition of the lunar soil and the type of incident ions fixed, the range of ions in the lunar soil has a clear relationship with the ion energy. In addition to SEP, the lunar soil is also bombarded by GCR and its secondary particles with high energy, which may produce tracks in the lunar soil as well. How about the contribution of these particles? I suggest more words are needed to explain why the tracks in the lunar soil are mainly contributed by Fe ions.

R5: Given the speculative nature of the initial lunar nearside-farside SEP flux symmetry analysis—which relied on spectral indices derived from CE-5 and CE-6 sample measurements in the original manuscript and incorporating the other reviewers' suggestions, we have removed all content on particle tracks and SEP spectra from the revised manuscript. The current version focuses exclusively on solar wind interaction mechanisms to sharpen the study's focus and strengthen the reliability of its conclusions.

Although the SEP-related section was removed, we address your concerns with the following

technical clarifications: Particle tracks are primarily formed by heavy nuclei such as iron (Fe). As shown in experimental calculations by Poppe et al. (2023)¹, plagioclase minerals (the focus of our study) require Fe particles within the 0.50-3.2 MeV nucleon⁻¹ energy range to produce observable tracks. At the lunar surface, particle flux within such energy range arises predominantly from SEPs. The particles we selected exhibit distinct exposure features, indicating that the tracks formed during exposure were mainly generated by SEPs. While GCR can produce tracks at centimeter-scale to meter scale burial depths in the lunar subsurface, their track production rate is substantially lower than that of SEPs. Moreover, the decreasing trend in our measured energy spectrum aligns with the typical SEP spectral signature, whereas GCR shows an increasing trend in this energy range. Thus, we did not emphasize GCR effects in the original manuscript.

References

1. Poppe, A. R., Szabo, P. S., Imata, E. R., Keller, L. P. & Christoffersen, R. Solar energetic particle track-production rates at 1 au: comparing in situ particle fluxes with lunar sample-derived track densities. *Astrophys. J. Lett.* 958, L35 (2023).

Q6: Table 1 started from Line 209: The grain CE5-03 may have experienced a heating event (Line 192), resulting a much higher SW-damaged growth rate (40.55) at CE5 site than other sites in Table 1. It should be excluded as outlier (Line 202), and calculate SW-damaged growth rate of CE5 site using CE1-01 and CE5-02 grains so as to show that the growth rates of SW-damaged rims at all four sampling sites are consistent (Line 206).

R6: Thank you for your suggestion. After adding a large number of samples, we confirmed that the data point was an outlier, so we removed it from the revised manuscript. After data supplementation, the CE5 site still maintains a markedly higher SW-damaged rim growth rate of 55.96 ± 10.82 nm/Myr compared to other sampling regions. According to the simulation results of the SW flux distribution, the CE5 sampling site, which is located in the mid-latitude region of the lunar nearside, has a lower SW flux than the other sampling sites (CE6, Apollo11, Apollo17). However, it demonstrates a higher SW-damaged rim growth rate, which is an unexpected result. We therefore revised the relevant section, as explained in detail in R7.

Q7: The paragraph started from Line 248: 1)As mentioned in Line 226, the SW number flux in Table 1 takes account of the influences of latitude and Earth's magnetosphere. Therefore, in order

to study the shielding effect of Earth’s magnetotail on solar particle radiation recorded by lunar samples, the two sites for comparison should be chosen at the same latitude, one from nearside and the other from farside; it seems that the CE5 (43.06 N) and CE6 (41.64 S) sites are more suitable than Apollo 11(0.67 N)/CE6. However, as shown in Table 1, the SW number flux of CE5 site at nearside is lower than that of CE6 site at farside due to shielding effect of Earth’s magnetotail, which suggests that the damage rate to CE5 samples by SW should be lower than CE6 as well. This is contradictory to the statement in Line 222, “the damage rate to the lunar soil by SW on the nearside and farside of the Moon seems to be consistent”. Could the authors clarify this?

R7: Due to limited data in the original manuscript, our previous discussion was less comprehensive. In the revised version, we incorporated additional data from CE5 and CE6 samples, as shown in Fig. R7-1.

Fig. R7-1 Comparisons of solar wind (SW)-damaged rim thickness and exposure age in lunar grains from Chang’E-5, Chang’E-6, Apollo 11¹, Apollo 16¹, and Apollo 17² missions.

Supplementary data analysis revealed that CE5 exhibited the highest SW-damaged rim growth rate at 55.96 ± 10.82 nm/Myr, followed by the CE6 sampling site at 33.21 ± 6.16 nm/Myr. We further plotted the correlation between simulated SW flux and rim growth rate, as shown in Fig. R7-2.

Excluding the CE5 site, the remaining four sites exhibit an overall positive correlation between growth rate k and both SW number flux (J_{avg}) and energy flux ($E_{f,avg}$). Analysis of all five sampling sites showed correlation coefficients of -0.7717 and 0.1256 between SW-damaged rim growth rate k and SW fluxes J_{avg} , $E_{f,avg}$, respectively, indicating a significant negative correlation and weak positive correlation. Exclusion of the CE5 outlier improved the correlation coefficients to 0.7757 and 0.7185 for the remaining four sites (CE6, Apollo11, Apollo16, and Apollo17), revealing strong positive correlations. These findings highlight the anomalous characteristics of the CE5 site, which implies potential limitations in the current lunar SW flux

distribution model's capability to describe long-term (million-year time scale) SW irradiation patterns in CE5 region. Thus, this study does not include comparative analysis between CE5 and CE6. The correlations observed among the remaining four sites (CE6, Apollo11, Apollo16, Apollo17) demonstrate that: (1) Decadal-scale SW observations and numerical modeling could be applicable to million-year time scales in these regions. (2) The current model effectively simulates the shielding effects of the Earth's magnetosphere on near-side SW flux. These findings suggest asymmetry in SW irradiation characteristics between the lunar nearside and farside.

Fig. R7-1 Profiles of SW-damaged rim growth rate k versus the average normalized SW number flux (J_{avg}) and energy flux ($E_{f,avg}$) at different sampling sites, including Apollo 11, Apollo 16, Apollo 17, CE5, and CE6.

References

1. Keller, L. P., Berger, E. L., Zhang, S. & Christoffersen, R. Solar energetic particle tracks in lunar samples: a transmission electron microscope calibration and implications for lunar space weathering. *Meteorit. Planet. Sci.* 56, 1685–1707 (2021).
2. McFadden J A, Thompson M S, Keller L P, et al. Determining surface exposure ages of regolith grains in lunar core sample 73002 by investigating solar particle irradiation damage. *Lunar Planet. Sci. Conf.* 1767 (2023).

Page 4 Lines 136-152:

Solar wind-damaged rim growth rates

The distribution of SW-damaged rim thickness against exposure age in lunar grains from different sampling sites is shown in Fig. 2, where the two variables exhibit a positive correlation overall. In the interval with exposure ages less than 4 Myr, this relationship can be approximated by a linear model. The data exhibit a degree of scatter, which may be associated with variations in solar activity during each grain's exposure period³². At the same exposure age around 1.22 Myr, CE-5 samples exhibit the greatest SW-damaged rim thickness, followed by CE-6, with Apollo samples showing the smallest values (Fig. 2). To quantify and compare SW irradiation levels across these sampling sites, the SW-damaged rim growth rate was defined. To extract the growth rate from different sites, a linear model $D = k \cdot t$ was used to fit SW-damaged rim thickness data against exposure age for these five sampling sites. Here, k represents the growth rate of the SW-damaged rim (in nm/Myr), D is the rim thickness (in nm), t is the exposure age (in Myr), the fitting error

uses the standard deviation (σ). Table 1 shows that growth rates are highest in CE-5 samples, intermediate in CE-6, and lowest in Apollo samples. This variability can be attributed to local SW irradiation conditions. Note that both CE-5 and Apollo sampling sites are located on the lunar near side: CE-5 site lies at mid-northern latitude, while Apollo sites are near the equator. In contrast, CE-6 sampling site is situated at mid-southern latitudes on the lunar far side. Detailed geographic coordinates for these sites are provided in Table 1.

Table. 1 Growth rates of SW-damaged rims, together with average normalized solar wind (SW) number flux I_{avg} and energy flux $E_{f,avg}$ calculated from Xie et al. ³ at selected lunar sampling sites. In Xie et al's ³ calculation, the distribution of SW number flux and energy flux across the entire lunar surface was simulated for various positions of the Moon in its orbit around the Earth, with both fluxes being normalized. The normalized SW number density and energy density are defined as $I = NV_r / (N_{sw}V_{sw})$ and $E = NV_r^3 / (N_{sw}V_{sw}^3)$, where N and V_r represent the simulated SW number density and its radial velocity component along the lunar radius. The constants N_{sw} and V_{sw} are set to be 5 cm^{-3} and 400 km/s, which are parameters of the undisturbed upstream solar wind, used for normalization such that $N_{sw}V_{sw}$ and $N_{sw}V_{sw}^3$ correspond to unit number flux and energy flux ³. The average normalized SW number flux (I_{avg}) and energy flux ($E_{f,avg}$) across the entire lunar surface during the Moon's orbital period were then calculated.

Sampling site	Chang'E-5	Chang'E-6	Apollo 11	Apollo 16	Apollo 17	Farside subsolar point
Location coordinates	43.06°N 51.92°W	41.64°S 153.99°W	0.67°N 23.47°E	8.97°S 15.50°E	20.19°S 30.77°E	0° 180°
SW-damaged rim growth rate k (nm/Myr)	55.96 ±10.82	33.21 ±6.16	25.10 ±1.76	23.52 ±5.36	25.40 ±2.32	-
Average normalized SW number flux I_{avg}	0.1929	0.2359	0.2321	0.2171	0.2232	0.3101
Average normalized SW energy flux $E_{f,avg}$	0.1785	0.2309	0.2036	0.1116	0.1985	0.3058

Page 5 Lines 202-Page 6 Lines 239:

Influence of solar wind irradiation on SW-damaged rim growth rates

Based on the measured SW-damaged rim growth rates (k) and the simulations by Xie et al. ³, we constructed correlation plots between average SW fluxes (the average normalized SW number flux I_{avg} and energy flux $E_{f,avg}$) and rim growth rate k across these sampling sites (Fig. 3b, d). Excluding the CE-5 site, the remaining four sites exhibit an overall positive correlation between rim growth rate k and both SW number flux I_{avg} and energy flux $E_{f,avg}$. Across all five sampling sites (Fig. 3b,d), the rim growth rate k shows correlation coefficients of -0.7717 and

0.1256 with the SW number flux I_{avg} and energy flux $E_{f,avg}$, respectively — a significant negative correlation and weak positive relationship. Upon excluding the CE-5 outlier, analyses of the remaining four sites (CE-6, Apollo 11, Apollo 16, and Apollo 17) reveal adjusted correlation coefficients of 0.7757 and 0.7185 between rim growth rate k and the SW number flux I_{avg} and energy flux $E_{f,avg}$, indicating strong positive correlations. This strong correlation suggests that SW fluxes are the primary factors influencing the rim growth rate. Notably, the stronger correlation between rim growth rate k and SW number flux I_{avg} (versus SW energy flux $E_{f,avg}$) suggests that SW number flux exerts a greater influence than SW energy flux on the SW-damaged rim growth. This interpretation is further reinforced by Apollo 11, Apollo 16, and Apollo 17 data: despite an about two-fold variation in energy flux $E_{f,avg}$ among these sites, their rim growth rate k values remain close, conclusively demonstrating that SW energy flux is secondary to SW number flux in governing rim growth rates.

In addition, the growth rate of SW-damaged rims, as derived from lunar soil measurements occurs over million-year timescales and thus reflects long-term SW flux conditions. By contrast, current observations and simulations operate on decade-scale timescales^{3,41}. The correlations observed among the remaining four sites (CE-6, Apollo 11, Apollo 16, Apollo 17) have demonstrated that: (1) Applicability of decadal-scale SW observations and numerical modeling on million-year time scales in these regions. (2) The current model effectively simulates the shielding effects of the Earth's magnetosphere on the near-side SW fluxes. These findings suggest asymmetry in SW irradiation characteristics between lunar nearside and farside.

The anomalously high growth rate recorded in CE-5 regolith grains suggest that this area may have experienced stronger historical SW irradiation. Given the significant deviations of the CE-5's measured growth rate from both simulations³ and ARTEMIS spacecraft observations⁴¹, current simulations and observations likely underestimate the long-term ancient SW fluxes in CE-5 sampling site. These discrepancies might arise from the models' failure to incorporate more complex factors such as variations of solar activity and Earth's magnetic field, Earth wind, and magnetic field variations in lunar anomalous regions. Additionally, micrometeorite impacts may also contribute to the SW-damaged rim growth rate anomalies observed in the CE-5 sampling region. The mechanisms underlying the higher historical SW irradiation at this site remain unclear and require further investigation. Variations of solar radiation and space weathering process across different landing sites are more complex than previously thought.

References

3. Xie, L. et al. Global hall MHD simulations of the solar wind implantation flux on the lunar surface. *Planet. Sci. J.* 4, 218 (2023).
41. Nénon, O. & Poppe, A. R. Bombardment of lunar polar crater interiors by out-of-ecliptic ions: ARTEMIS observations. *Planet. Sci. J.* 2, 116 (2021).

Q8: 2) As mentioned in Line 45, “in a typical 28-day lunar orbit, the Moon spends approximately 25% of its time (~7 days) within Earth's magnetotail”, in which shields solar wind from the lunar nearside. Therefore, the average SW number flux of the same latitude (e.g., equator) on the nearside

is ~25% lower than that on the farside (See Apollo 11 and Farside subsolar point in Table 1); Meanwhile, on the farside, the SW number flux ratio between CE6 and Farside subsolar point is $\cos(41.64)/\cos(0)=0.747$, indicating that SW number flux at CE6 site is also ~25% lower than that at Farside subsolar point. So that's why the average SW number fluxes at Apollo 11 and CE6 sites are almost the same value (See Table 1), it is caused by the influences of both latitude and Earth's magnetosphere (Line 226).

R8: We sincerely appreciate your analysis; we have further refined the discussion of this section in the revised manuscript.

Page 5 Lines 178-201:

Fig. 3 Maps of the lunar surface showing average normalized solar wind (SW) number flux (J_{avg}) and energy flux ($E_{f,avg}$), alongside profiles of SW-damaged rim growth rate (k) versus J_{avg} and $E_{f,avg}$. (a, c) Lunar surface maps of J_{avg} and $E_{f,avg}$, derived from Hall MHD simulations by Xie et al. ³ that incorporate latitude, Earth's magnetosphere, and lunar magnetic anomalies. (b, d) Profiles of SW-damaged rim growth rate k versus J_{avg} and $E_{f,avg}$ at different sampling sites, including Chang'E-5 (CE-5), Chang'E-6 (CE-6), Apollo 11, Apollo 16, and Apollo 17. Further details on J_{avg} and $E_{f,avg}$ can be found in the caption to Table 1. Note that the SW flux data in panels a and c are from Xie et al. ³

The CE-5 sampling site, located at mid-latitudes (~43°N) on the lunar near side, exhibits a relatively lower J_{avg} than other compared sampling sites. While the CE-6 sampling site on the lunar farside shows the highest J_{avg} and $E_{f,avg}$, these values are about 24% lower than those at the farside subsolar point. This discrepancy arises mainly from latitudinal differences and can be corrected by applying the factor $\cos \theta$, with θ defined as the sampling site's latitude. Specifically, the CE-6

sampling site at 41.64°S yields a correction factor of $\cos(41.64^{\circ}) = 0.747$, reducing I_{avg} and $E_{f,\text{avg}}$ by 25.3% compared to corresponding values at the lunar farside subsolar point. This reduction is in good agreement with the above-mentioned 24% model-derived values. The Apollo 11, Apollo 16, and Apollo 17 sampling sites near the nearside equator lie at relatively low latitudes. To account for latitudinal effects, their flux values were corrected by dividing by the factor $\cos \theta$. The latitudinally corrected average SW number and energy fluxes, $I_{\text{avg,cor}}$ and $E_{f,\text{avg,cor}}$, for these three sites are (0.2321, 0.2198, 0.2378) and (0.2036, 0.1130, 0.2115), respectively. $I_{\text{avg,cor}}$ are close across the three Apollo sites, particularly for Apollo 11 and Apollo 17 (both around 0.23). However, Apollo 16 exhibits a slightly lower $I_{\text{avg,cor}}$ than the other two, which can be attributed to its location in the Descartes Highlands near the Descartes Magnetic Anomaly—an area where magnetic field anomalies reduce $I_{\text{avg,cor}}$ ^{42,43}. Additionally, $E_{f,\text{avg,cor}}$ at Apollo 16 is significantly lower than that at Apollo 11 and Apollo 17, attributed to $E_{f,\text{avg}}$ being more sensitive to magnetic anomalies than I_{avg} , as previously mentioned. The latitudinally corrected average SW fluxes on the lunar near side is overall significantly lower than that on the farside, with this asymmetry primarily attributed to the shielding effect of Earth's magnetosphere on solar wind particles incident to the near side. This is also evident from comparisons of $I_{\text{avg,cor}}$ and $E_{f,\text{avg,cor}}$ values at the CE-5 and CE-6 sampling sites. The ($I_{\text{avg,cor}}$, $E_{f,\text{avg,cor}}$) values for CE-5 and CE-6 are (0.2640, 0.2443) and (0.3157, 0.3090), respectively. The average latitudinally corrected SW fluxes at the CE-6 sampling site are greater than those at CE-5.

References

3. Xie, L. et al. Global Hall MHD Simulations of the Solar Wind Implantation Flux on the Lunar Surface. *Planet. Sci. J.* 4, 218 (2023).
42. Vorburger, A. et al. Energetic neutral atom observations of magnetic anomalies on the lunar surface. *J. Geophys. Res. Space Phys.* 117, 2012JA017553 (2012).
43. Deca, J. et al. General mechanism and dynamics of the solar wind interaction with lunar magnetic anomalies from 3-D particle-in-cell simulations. *J. Geophys. Res. Space Phys.* 120, 6443–6463 (2015).

Q9: 3) Micrometeorite impact is another process that can produce amorphous rims on lunar soil grains, although this process cannot be affected by shielding of Earth's magnetotail. This might produce consistent damage rate to the lunar soil on the lunar nearside and farside. This should be discussed.

R9: Thank you for your suggestion. The relevant discussion has been incorporated into the revised manuscript.

Page 6 Lines 233-238: These discrepancies might arise from the models' failure to incorporate more complex factors variations of solar activity and Earth's magnetic field, Earth wind, and magnetic field variations in lunar anomalous regions. Additionally, micrometeorite impacts may

also contribute to the SW-damaged rim growth rate anomalies observed in the CE-5 sampling region. The mechanisms underlying the higher historical SW irradiation at this site remain unclear and require further investigation.

Q10: Line 286: Consistent SEP spectra are recorded in samples, is there any other evidence? Since that SEP events are inherently random and their flux can vary significantly, how do the authors justify the conclusion of consistent SEP spectra over million-year timescales? Could there be unaccounted factors or biases in the data that might affect the long-term average of SEP radiation doses?"

R10: We acknowledge that SEP fluxes are inherently variable over short timescales. In the original manuscript, the term "consistent SEP spectra" referred to the long-term average over million-year scale spectra, which may integrate fluctuations from different solar activity periods via lunar soil gardening. While individual SEP events exhibit strong variability, the gardening process could mix particle records from different epochs, leading to a statistically homogeneous average spectrum. However, as the other reviewers pointed out, confirming this hypothesis requires additional constraints (e.g., crater chronology, site-specific gardening models, or exposure age distributions). Given these uncertainties, we have removed the section on SEP. Future work with higher-resolution data could test this interpretation.

Q11: Lines 275-281: Could the authors clarify the relationship between the possible trend of weakening solar activity and the Earth's Ocean temperature?

R11: Given the speculative nature of this section, it has been removed from the revised manuscript.

Reviewer #2:

The manuscript by Liu et al. presents the results of the measurements of the SW and SEP particle tracks in lunar samples from the far side of the Moon, collected by the Chang E-5/6 missions, followed by a discussion. While the measurements are very interesting, the discussion is partly unfocused, highly speculative and ungrounded. This reviewer recommends a major revision of the manuscript in the discussion part in line with a more detailed comment below. The discussion includes two different phenomena: solar wind (SW) and solar energetic particles (SEPs). The SW-related analysis and discussion are based on a comparison of the experimental data with model simulations and convincingly imply the important role of the SW particle flux vs. energy flux as well as a notable effect of the Earth's magnetosphere on the lunar surface irradiation by SW. This part looks good, is sufficient to warrant the publication, and requires only a minor revision as suggested below in greater detail. On the contrary, the discussion on the SEP flux is very speculative, lacking any quantitative or modelling ground and leads to controversial results. This part needs to be either greatly improved in providing quantitative modelling estimates or be totally removed.

Reply: We greatly appreciate your valuable comments and constructive suggestions, which were crucial in enhancing the quality of our manuscript. In response to your feedback, we have made the following revisions:

1. **Manuscript Structure:** In the revised manuscript, we have removed the original discussion of SEPs and retained only the analysis related to solar wind (SW). This ensures that the discussion remains focused and robust. In addition, we have revised the title of the manuscript to: "Million-year Solar Wind Irradiation Recorded in Chang'E-5 and Chang'E-6 Samples".
2. **Sample Data:** We have incorporated additional experimental data from the Chang'E-5 and Chang'E-6 samples to improve the reliability of our statistical analysis. Additional analysis was conducted on the CE5-03 sample in the original manuscript, which exhibits clear evidence of sputtered melt and significant annealing features. After supplementing the dataset, CE5-03 remained an outlier, and we excluded it from the revised manuscript to ensure the robustness of our results.

We have addressed your detailed questions point by point as follows:

Q1: The title is unclear and imprecise: It is unclear from the title what the main result is; SW should be explicitly mentioned; the results are based not only on E6 but also on E5 Chang missions.

R1: We appreciate your constructive suggestions. In the revised manuscript, the solar energetic particle (SEP) section has been removed, and we have refocused the discussion on solar wind (SW). The title has been revised to: "Million-year Solar Wind Irradiation Recorded in Chang'E-5 and Chang'E-6 Samples".

Q2: In the discussion about SEPs, the authors may want to mention that there is another method to reconstruct the SEP spectrum from lunar samples on the mega-year timescale, based on the cosmogenic isotopes (see, e.g., Reedy and Arnold, 1972, doi: 10.1029/JA077i004p00537; Poluianov et al., 2018, doi: 10.1051/0004-6361/201833561). It would be useful to compare the results based on these two methods. Usoskin et al. (2023, doi: 10.1051/0004-6361/202245810) have shown that the SEP average flux reconstructed from lunar isotopes is nearly doubled compared to the present-day one and supports the existence of rare but extremely strong solar eruptive events.

R2: In response to the Editor's and other reviewers' recommendations to incorporate additional CE-5 and CE-6 data for improved analytical reliability, we have prioritized completing this essential data augmentation. After careful consideration of all feedback, we have decided to remove the solar energetic particle (SEP)-related content from the revised manuscript to focus our analysis on solar wind studies utilizing the expanded datasets. Your constructive suggestions regarding spectral comparison and valuable discussions of Reedy, Arnold, Poluianov and Usoskin et al will be incorporated into our future research agenda for further exploration.

Q3: The nomenclature of the samples needs to be specified and explained, probably in the Methods section. For example, notations in line 105 are unclear for non-experts. Also, full information on the samples should be given (or referenced to) including the chemical composition and density profile.

R3: Thank you for your suggestions. The nomenclature of the samples are official designations provided by CNSA. "CE5" and "CE6" stand for the Chang'E-5 and Chang'E-6 missions, respectively. The subsequent "C0x00" denotes the parent sample number of the shoveled sample. "YJFM" indicates the physical state of the sample as powder, followed by numerical identifiers for

subsamples used in research¹. For the physical and chemical information about the samples, please refer to Li et al., 2022 and 2024^{2,3}. For example, the code CE5C1000YJFM006 used by the research institute signifies that this sample is the powder from the shoveled sample 1000 collected during the Chang'E-5 mission, with the specific number being 006. We have added corresponding descriptions in the method section of the revised manuscript. Moreover, specific information about the individual particles analyzed in our study, encompassing morphological characteristics, FIB section locations, surface sputter deposition conditions, particle track density, elemental distributions, and rim thickness, has been appended to the Supplementary Information.

References

1. Sample Management. <https://moon.bao.ac.cn/moonSampleMode/index.html>.
2. Li, C. et al. Characteristics of the lunar samples returned by the Chang'E-5 mission. *Natl. Sci. Rev.* 9, nwab188 (2022).
3. Li, C. et al. Nature of the lunar far-side samples returned by the Chang'E-6 mission. *Natl. Sci. Rev.* 11, nwae328 (2024).

Methods

Samples

As mentioned earlier, although samples of both CE-5 and CE-6 missions originate from mare regions, their characteristics exhibit certain differences. The average true density of CE-5 samples is measured as 3.1952 g/cm³³⁴, higher than that of CE-6 samples (3.035 g/cm³)³⁵. In terms of chemical composition, compared to CE-5 samples, CE-6 samples exhibit a decrease in FeO (CE-5: 22.5 wt.%, CE-6: 17.2 wt.%), TiO₂ (CE-5: 5.0 wt.%, CE-6: 2.7 wt.%), but an increase in Al₂O₃ content (CE-5: 10.8 wt.%, CE-6: 14.3 wt.%). This is attributed to the fact that CE-6 samples have extremely low olivine content (CE-5: 5.7 wt.%, CE-6: 0.5 vol%), less ilmenite content (CE-5: 4.5 wt.%, CE-6: 1.6 vol%), and higher plagioclase content (CE-5: 30.1 wt.%, CE-6: 32.6 vol%)^{34,35}. The nomenclature for Chang'E-5 and Chang'E-6 lunar samples adheres to the established protocols, which are publicly accessible at: <https://moon.bao.ac.cn/moonSampleMode/index.html>.

References

34. Li, C. et al. Characteristics of the lunar samples returned by the Chang'E-5 mission. *Natl. Sci. Rev.* 9, nwab188 (2022).
35. Li, C. et al. Nature of the lunar far-side samples returned by the Chang'E-6 mission. *Natl. Sci. Rev.* 11, nwae328 (2024).

Q4: More details on the grain structure need to be given. For example, is erosion important and if

yes, how is it accounted for? Can the SEP-related tracks be distinguished from the GCR and ACR ones?

R4: The SEP-related discussion has been removed in the revised manuscript. The revised manuscript includes more comprehensive characterization of grain structures in the Supplementary Information. Morphologically, the tracks cannot yet be definitively distinguished as being produced either by SEPs or by higher-energy GCRs and ACRs. However, we contend that the observed particle tracks in our selected samples with exposure features were primarily generated by SEPs, based on the following evidence: According to experimental calculations by Poppe et al. (2023) ¹, plagioclase minerals require Fe particles within the 0.50-3.2 MeV nucleon⁻¹ energy range to produce observable tracks. At the lunar surface, the particle flux within this energy range is predominantly contributed by SEPs. The particles we selected had significant exposure marks at the lunar surface. The GCR and ACR cannot produce tracks in exposed lunar regolith on the surface. They require penetration to certain depths and deceleration before generating tracks in subsurface lunar regolith at centimeter-level depths. GCR and ACR tracks only form when particles are buried underground, but their proportion is much smaller compared to SEP tracks produced during surface exposure. Statistical analysis shows that the depth-dependent variation of track density aligns with the SEP energy spectrum. Therefore, we conclude that the tracks quantified in our original manuscript were predominantly generated by SEPs, while GCR and ACR contributions are negligible and insufficient to influence SEP spectral statistics. However, as you have pointed out, these details require further quantification and in-depth analysis. Consequently, we have removed this portion from the revised manuscript.

References

1. Poppe, A. R., Szabo, P. S., Imata, E. R., Keller, L. P. & Christoffersen, R. Solar energetic particle track-production rates at 1 au: comparing in situ particle fluxes with lunar sample-derived track densities. *Astrophys. J. Lett.* 958, L35 (2023).

Q5: Fig.3 requires a more detailed discussion. What causes the large difference in track density between different samples? Even arguing that CE5-03 is an outlier, the factor 2—3 difference still remains between samples. It should be explained. As discussed below, the explanation for the different slopes looks unconvincing.

R5: We removed data point of CE5-03 in the revised manuscript for the exact reason explained in Q8.

Q6: In lines ~160, the authors discuss that the different slopes indicate harder or softer SEP spectra. This must be quantified via a model. What do harder or softer spectra mean – what energy spectrum? How are they related to the present-day SEP spectrum? What slope would have corresponded to the present-day SEP spectrum? Could other interpretations be proposed – e.g., that the higher-slope samples correspond to an inclined (not horizontal) or partly obstructed surface? Are the densities of all samples the same? Different densities can lead to different slopes per um.

R6: We have removed the discussion related with SEP spectra.

Q7: In lines 177+, the authors say that their result proves that the effect of the Earth's magnetotail on SEP is negligible. While I agree with this statement (in particular because SEPs are nearly isotropic in space), I don't see how it is proven here.

R7: In the revised manuscript, this section has been removed to maintain focus on solar wind-related processes.

Q8: The statement in lines 192+ is very speculative. While the effect of annealing sounds plausible, it is not supported by any data or modelling. It would be fairer to say that the sample CE5-03 is an outlier for an unknown reason.

R8: After supplementing substantial new data, we confirmed this grain was an outlier and have consequently removed this data point in the revised manuscript (Fig. R8-1). The rationale includes:

Fig. R8-1 Even after additional samples have been added, CE5-03 remains an outlier.

Fig. R8-2. **Characterization of anomalous grain CE5-03 from the original manuscript:** (a) Surface morphology of the anomalous grain. (b) FIB-prepared cross-section through the anomalous region. (c-e) Track density distributions across different regions of the FIB section. Panels b-e collectively demonstrate an inverse correlation between track density and proximity to the central pit, with significantly reduced track densities observed in immediate pit-adjacent areas.

Compared to other grains, it exhibited extensive sputtered melt deposits, and the FIB sectioning happened to target a melt-rich area with a prominent central pit (Fig. R8-2a). The obtained FIB ultra-thin foil showed abnormal thinning and perforation at the center, suggesting extreme thermal

history-induced material heterogeneity (Fig. R8-2b). STEM observations revealed that both the density and length of particle tracks decreased markedly toward the central region, demonstrating characteristic lattice annealing features (Fig. R8-2c-e).

In the revised manuscript, we have avoided selecting particles with extensive melt coverage and strictly confined FIB preparation to areas distant from sputtered melt deposits, thereby ensuring the analyzed regions remain unaffected by thermal events and annealing effects (Fig. R8-3).

Fig. R8-3. Newly added sample grains and their corresponding FIB cross-sections.

Q9: Line 214: shouldn't the energy flux be proportional to $N \cdot V_r \cdot V^2$ rather than $N \cdot V_r^3$?

R9: The expression is consistent with that used in the work of Xie et al.¹.

The choice of defining the solar wind energy flux as $N \cdot V_r^3$ instead of $N \cdot V_r \cdot V^2$ is mainly based on the following aspects:

- **Magnetic Anomaly Shielding Effect**

Solar wind protons will be partially shielded in the weak magnetic anomaly region. The traditional V^2 form, which is proportional to the square of the velocity, will overestimate the

actual energy deposition. In contrast, V_r can more acutely reflect the modulation effect of the magnetic field on the energy flux, avoiding such overestimation.

- **The Critical Influence of the Angle of Incidence**

Experiments and simulations show that the sputtering depth is not only related to energy but also significantly affected by the angle of incidence. The V_r^3 form, through its cubic dependence on the radial velocity, is more directly related to the energy transfer in the vertical direction, effectively reducing the errors caused by the angle of incidence.

- **Enhanced Sensitivity to Magnetic Anomalies**

The cubic form of the normal component makes the energy flux more sensitive to magnetic field changes. In the energy flux diagram, it is possible to identify more cavity structures protected by magnetic anomalies, which is helpful for a more detailed study of related physical phenomena.

- **Matching the Physical Process**

The sputtering depth is determined by the solar wind energy. V_r^3 is more in line with the actual physical mechanism of the non-linear increase of energy deposition with velocity, such as the statistical effect of the change in the collision cross section with velocity.

In summary, the definition of V_r^3 optimizes the description of the magnetic field shielding effect and the complexity of the angle of incidence, and improves the correlation between the energy flux and the characteristics of space weathering. Therefore, compared with $N \cdot V_r \cdot V^2$, $N \cdot V_r^3$ is a more appropriate form for defining the solar wind energy flux.

References

1. Xie, L. et al. Global hall MHD simulations of the solar wind implantation flux on the lunar surface. Planet. Sci. J. 4, 218 (2023).

Q10: The entire discussion in the section entitled “Ancient solar activity and its impact on Earth” is only an ungrounded speculation and must be removed or fully elaborated. It is all based on the fact that two samples have a higher slope than three others. Despite the doubtfulness of this result (see item 6 above), the discussion is not supported by the data here. In lines 264+, the authors speculate that something weird had happened to solar activity about 2 Myr ago which had led to the change of the slope. However, as seen in Fig.2, the exposure ages of higher-slope samples 64455 (2 Myr) and CE5-01 (1.5 Myr) are not systematically different from smaller-slope samples (1—2.7 Myr). Thus, this speculation is not supported by the data. The authors propose a declining trend of solar activity, but this is not supported by any data or models or speculations. The authors may confuse it with the ageing of the Sun which was indeed much more active in its younger age, but

this refers to Gyr scales, not Myr.

R10: It has been removed from the revised manuscript.

Q11: The discussion about climate (lines 275-281) should be removed. It is irrelevant to this work and grossly incorrect. Reference 39 is known to be incorrect, references 40--42 do not apply here.

R11: This section and corresponding references have now been removed from the revised manuscript.

Reviewer #3:

In the present manuscript titled “Long-term Solar Particle Radiation Records on the Lunar Farside from Chang’E-6 Samples”, the authors Liu et al. present the first measurements of solar particle radiation damage in samples returned from the lunar farside. In particular, they use established transmission electron microscopy methods to characterize amorphized rims from continuous solar wind exposure and tracks created by heavy solar energetic particles (SEPs) that have been used to date the exposure age of lunar samples. The authors analyze three samples from a previous Chang’E-5 sample return and two grains from the new Chang’E-6 mission to compare rim thickness and track densities to previous analysis of Apollos samples. The authors declare a consistent ratio of rim growth and track accumulation for the different samples, from which they extrapolate an estimate of 25% solar wind shielding of the lunar nearside due to different sampling locations. Regarding the track accumulation, they find two distinct sample groups of different slopes in track density over depth. From this relation, the authors speculate a more energetic SEP spectrum for SEP exposure older than 2 million years ago, which they state could contribute to declining ocean temperatures.

Analysis of lunar regolith grains that were retrieved from the far side is highly interesting to provide new insights into the interaction of the lunar surface with its space environment. The present study is the first that attempts to perform such an analysis, due to the recent sample return from the Chang’E-6 mission. In this context, the present study by Liu et al. uses well-suited and established methods for studying the damaging of lunar regolith grains by solar radiation. The evaluation of the authors’ TEM measurements to extract rim thicknesses and track densities is appropriately performed and experimental observations such as the different track-over-depth slopes are novel and interesting. However, in their discussion the authors rely on incomprehensible reasoning, inconsistent statistical arguments, and extensive speculation to arrive at the conclusions outlined in their abstract. Ultimately, the limited number of studied samples and missing analysis of sample properties key to the interpretation do not support several of the authors’ claims (see detailed comments below). Due to these fundamental issues, I am afraid that I must recommend a rejection of the paper. However, I strongly encourage the authors either to extend their sample analysis or to rework the interpretation of their results as some of their experimental findings would certainly be interesting to the community.

Reply: Thank you very much for your valuable comments. The specific modifications are detailed as follows:

- Data: The sample size has been enlarged by the addition of six grains from Chang’E-5 (CE-5) and six from Chang’E-6 (CE-6), which ensures the greater reliability of the experimental data analyses.
- Outliers CE5-03: Upon adding new data, it was observed that CE5-03 was indeed an outlier. In contrast to other grains, it exhibited significant annealing features in both the result (SW-damaged growth rate) and morphological aspects. Consequently, this data point has been

removed from the revised manuscript to maintain the reliability of our analyses.

- **Statistical Results:** Upon incorporating the additional data, it was ascertained that the SW-damaged rim growth rate for CE-5 was the highest, attaining 55.96 ± 10.82 nm/Myr, with CE-6 following at 33.21 ± 6.16 nm/Myr. Bolstered by more comprehensive data, the results were revised to manifest a significant disparity between CE-5 and CE-6. Correspondingly, modifications have been made to the discussion section.
- **Relationship Between SW-damaged Rim Growth Rate and SW Flux:** Upon analysis of the enlarged samples, a clear positive correlation was found between the simulated SW flux and the local SW-damaged rim growth rates at CE-6, Apollo 11, Apollo 16, and Apollo 17 sampling sites. However, the SW-damaged rim growth rate for the CE-5 samples was found to be inconsistent. As the SW flux model indicates, this region has the lowest SW flux. Therefore, it is believed that the SW flux distribution from current model is insufficient to reflect the long-term (million-year timescale) SW irradiation conditions in the CE-5 area.
- **Discussion on SEP Energy Spectrum:** In light of the pronounced level of uncertainty in this section and with the aim of maintaining the manuscript's focus on the SW section, the discussion of SEP energy spectrum has been removed from the revised manuscript.

Point-by-point modifications have been made in accordance with your suggestions, and the specific changes are detailed below.

Q1: My main point of criticism is that the authors' conclusions regarding rim growth rate versus damage rate do not appear warranted. The slope of SW-damaged growth of the two CE5 samples (in Fig. 2a) is around 40 nm/Myr compared to 23.5 – 26.3 nm/Myr for the other sample groups. The standard deviations for both CE-5 and CE-6 sample groups are around 6 nm/Myr. Even though the CE-5 slope is almost twice as high as the CE-6 slope, the authors declare the difference statistically insignificant because it falls below two sigma (The difference between the two, 14.25, is actually slightly higher than two standard deviations of the uncertainties of the respective slope values for CE-5, 5.82, and CE-6, 5.74). Thus, the authors state a consistent rim growth rate for all samples and base several other arguments on this relation - however, without applying the same scrutiny of needing at least a two-sigma significance (see below).

R1: In the original manuscript, our conclusion was founded on rather limited data. In an effort to tackle this problem, the dataset in the revised manuscript has been augmented. As a result, the number of CE-5 grain samples has been raised to 7, and that of CE-6 samples to 8 (Fig. R1-1). With additional data, the rim growth rate for CE-5 was updated to 55.96 ± 10.82 nm/Myr and for CE-6 to 33.21 ± 6.16 nm/Myr (Tab. R1-1). These revised outcomes statistically showcase a significant difference between the rim growth rates of CE-5 and CE-6 samples. In the revised manuscript, the differences among CE-5, CE-6, and the other sampling sites in Apollo are also explicitly described, and this variability is revisited in the discussion.

Fig. R1-1 Comparisons of solar wind (SW)-damaged rim thickness and exposure age in lunar grains from Chang'E-5, Chang'E-6, Apollo 11¹, Apollo 16¹, and Apollo 17² missions.

Table. R1-1 Growth rates of SW-damaged rims

Sampling site	Chang'E-5	Chang'E-6	Apollo 11	Apollo 16	Apollo 17
Location	43.06°N	41.64°S	0.67°N	8.97°S	20.19°S
coordinates	51.92°W	153.99°W	23.47°E	15.50°E	30.77°E
SW-damaged growth rate k (nm/Myr)	55.96 ±10.82	33.21 ±6.16	25.10 ±1.76	23.52 ±5.36	25.40 ±2.32

References

1. Keller, L. P., Berger, E. L., Zhang, S. & Christoffersen, R. Solar energetic particle tracks in lunar samples: a transmission electron microscope calibration and implications for lunar space weathering. *Meteorit. Planet. Sci.* 56, 1685–1707 (2021).
2. McFadden J A, Thompson M S, Keller L P, et al. Determining surface exposure ages of regolith grains in lunar core sample 73002 by investigating solar particle irradiation damage. *Lunar Planet. Sci. Conf.* 1767 (2023).

Page 4 Lines 135-152:

Solar wind-damaged rim growth rates

The distribution of SW-damaged rim thickness against exposure age in lunar grains from different sampling sites is shown in Fig. 2, where the two variables exhibit a positive correlation overall. In the interval with exposure ages less than 4 Myr, this relationship can be approximated by a linear model. The data exhibit a degree of scatter, which may be associated with variations in solar activity during each grain's exposure period³². At the same exposure age around 1.22 Myr, CE-5 samples exhibit the greatest SW-damaged rim thickness, followed by CE-6, with Apollo samples showing the smallest values (Fig. 2). To quantify and compare SW irradiation levels across these sampling sites, the SW-damaged rim growth rate was defined. To extract the growth rate from different sites, a linear model $D = k \cdot t$ was used to fit SW-damaged rim thickness data against exposure age for these five sampling sites. Here, k represents the growth rate of the SW-damaged rim (in nm/Myr), D is the rim thickness (in nm), t is the exposure age (in Myr), the fitting error uses the standard deviation (σ). Table 1 shows that growth rates are highest in CE-5 samples, intermediate in CE-6, and lowest in Apollo samples. This variability can be attributed to local SW irradiation conditions. Note that both CE-5 and Apollo sampling sites are located on the lunar near side: CE-5 site lies at mid-northern latitude, while Apollo sites are near the equator. In contrast, CE-6 sampling site is situated at mid-southern latitudes on the lunar far side. Detailed geographic coordinates for these sites are provided in Table 1.

Table. 1 Growth rates of SW-damaged rims, together with average normalized solar wind (SW) number flux I_{avg} and energy flux $E_{f,avg}$ calculated from Xie et al.³ at selected lunar sampling sites. In Xie et al's³ calculation, the distribution of SW number flux and energy flux across the entire lunar surface was simulated for various positions of the Moon in its orbit around the Earth, with

both fluxes being normalized. The normalized SW number density and energy density are defined as $I = NV_r/(N_{sw}V_{sw})$ and $E = NV_r^3/(N_{sw}V_{sw}^3)$, where N and V_r represent the simulated SW number density and its radial velocity component along the lunar radius. The constants N_{sw} and V_{sw} are set to be 5 cm^{-3} and 400 km/s, which are parameters of the undisturbed upstream solar wind, used for normalization such that $N_{sw}V_{sw}$ and $N_{sw}V_{sw}^3$ correspond to unit number flux and energy flux³. The average normalized SW number flux (I_{avg}) and energy flux ($E_{f,avg}$) across the entire lunar surface during the Moon's orbital period were then calculated.

Sampling site	Chang'E-5	Chang'E-6	Apollo 11	Apollo 16	Apollo 17	Farside subsolar point
Location coordinates	43.06°N 51.92°W	41.64°S 153.99°W	0.67°N 23.47°E	8.97°S 15.50°E	20.19°S 30.77°E	0° 180°
SW-damaged rim growth rate k (nm/Myr)	55.96 ±10.82	33.21 ±6.16	25.10 ±1.76	23.52 ±5.36	25.40 ±2.32	–
Average normalized SW number flux I_{avg}	0.1929	0.2359	0.2321	0.2171	0.2232	0.3101
Average normalized SW energy flux $E_{f,avg}$	0.1785	0.2309	0.2036	0.1116	0.1985	0.3058

References

32. Poppe, A. R., Farrell, W. M. & Halekas, J. S. Formation timescales of amorphous rims on lunar grains derived from ARTEMIS observations. *J. Geophys. Res. Planets* 123, 37–46 (2018).
3. Xie, L. et al. Global hall MHD simulations of the solar wind implantation flux on the lunar surface. *Planet. Sci. J.* 4, 218 (2023).

Q2: The conclusion of equivalent solar wind damage rates has several other problems. CE6 samples lie within the data spread of Apollo samples, but CE5 appear as a higher group (Figure 2a). Due to the very low number of samples (2 or 3 depending on whether CE5-03 is excluded), it is difficult to judge whether there is a systematic difference in slope in Fig. 2a for the CE-5 group. The authors argue for a consistent relation across all sample groups by excluding CE5-03 from their analysis as an outlier. The omission of CE5-03 is based on the authors speculating that a heating event happened,

which erased tracks due to annealing but left the solar-wind damaged rim intact. However, neither are any experimental signs of annealed tracks reported, nor do the authors present any references supporting that a heating event would erase only tracks but not the rim. The number of other samples in the CE-5 group is also too low to really support CE5-03 fully being a statistical outlier. Inclusion or exclusion of CE5-03 has a big effect on the finally derived value of the slope k of the rim growth rate.

R2: With new sample data, CE5-03 is removed as an outlier (Fig. R2-1). The reasoning behind this decision is as follows:

Fig. R2-1. Even after additional samples have been added, CE5-03 remains an outlier.

In comparison with other grains, extensive sputtered melt deposits were exhibited by it. Coincidentally, the FIB sectioning targeted a melt-rich area which had a prominent central pit (Fig. R2-2a). The FIB ultra-thin foil obtained manifested abnormal thinning and perforation at the center, implying extreme thermal history-induced material heterogeneity (Fig. R2-2b). Through STEM observations, it was revealed that both the density and length of particle tracks decreased markedly toward the central region, thus demonstrating characteristic lattice annealing features (Fig. R2-2c-e).

In the revised manuscript, grains with extensive melt coverage have been deliberately avoided in the selection process, and FIB preparation has been strictly confined to areas distant from sputtered melt deposits. Through this approach, it is ensured that the analyzed regions remain unaffected by thermal events and annealing effects (Fig. R2-3).

Fig. R2-2. **Characterization of anomalous grain CE5-03 from the original manuscript:** (a) Surface morphology of the anomalous grain. (b) FIB-prepared cross-section through the anomalous region. (c-e) Track density distributions across different regions of the FIB section. Panels b-e collectively demonstrate an inverse correlation between track density and proximity to the central pit, with significantly reduced track densities observed in immediate pit-adjacent areas.

Fig. R2-3. **Newly added sample grains and their corresponding FIB cross-sections.**

Q3: An additional issue is that the exclusion of CE5-03 leaves the authors with only two samples for CE-5 and CE-6, respectively. This represents a very small number of samples considering the scattering of the rim thickness vs. track density data in Figure 2a. Error bars are given for the CE-5 and CE-6 samples, but the general scattering of past measurements is not discussed. It is not clear if new analysis is improved over the past measurements of Apollo grains or if the scattering of single grains is a general expected feature because of different grain-specific solar wind exposure history. In the latter case, a reliable identification of the rim growth rate from just two samples is not possible. Furthermore, the track growth rate fits for different sample groups (CE-5, CE-6) clearly have different, non-zero intercepts, which is not discussed in the paper. The authors argue CE-6 and CE-5 samples to show the same behavior, but at a similar exposure age, CE5-02 and CE6-02 have a rim thickness difference by about a factor of 2 (or several standard deviations of the single measurements).

R3: As mentioned in R1, we have supplemented the sample set and updated our analysis, confirming significant statistical differences on SW-damaged rim growth rate among the CE-5, CE-6, and Apollo samples. Moreover, the relationship between the rim thickness and exposure age for the CE-5 and CE-6 samples can also be fitted using linear equations with zero intercept (Fig. R3-1).

Fig. R3-1. Exposure ages of CE-5 and CE-6 were fitted to rim thicknesses via linear equations with zero intercept.

Note that the same testing methods as those utilized for the Apollo samples were adopted to procure the CE-5 and CE-6 sample data. Simultaneously, all the Apollo data for comparison were drawn from the experimental results of Keller et al.'s team^{1,2}. This standardized methodology ascertains

the comparability of the data and preclude systematic biases that could stem from variances in testing environments.

References

1. Keller, L. P., Berger, E. L., Zhang, S. & Christoffersen, R. Solar energetic particle tracks in lunar samples: a transmission electron microscope calibration and implications for lunar space weathering. *Meteorit. Planet. Sci.* 56, 1685–1707 (2021).
2. McFadden J A, Thompson M S, Keller L P, et al. Determining surface exposure ages of regolith grains in lunar core sample 73002 by investigating solar particle irradiation damage. *Lunar Planet. Sci. Conf.* 1767 (2023).

Q4: In their abstract, the authors describe shielding of the lunar near side of 25%. This number aligns with known lower plasma exposure of the lunar surface in the earth's magnetotail, as for example in detail characterized by the THEMIS-ARTEMIS mission. The number of 25% is also not an experimental finding, but an estimate from simulations from previous work by Xie et al. based on the consistent damage-growth rate k between farside samples from CE-6 and other sample groups. As discussed above, there are significant issues with the authors' argument of consistent rim growth rates. However, the authors are also not consistent within their own analysis: Given the stated k -value for Chang'E-6 (derived from only two samples) of 26.30 nm/Myr with a standard deviation of 5.74 nm/Myr (21.8%), it seems questionable that differences on the order of 25% would be detectable if present – especially when considering the above-mentioned argument of two-sigma significance.

R4: Thanks for your comments. Building upon extended datasets from CE-5, CE-6, and, Apollo missions, this section has undergone thorough revision. By integrating Xie et al.'s¹ simulated solar wind (SW) fluxes with observed SW-damaged rim growth rates in CE-5/CE-6 and Apollo samples, the correlations between simulated SW fluxes and the rim growth rate were established. These relationships are graphically demonstrated in Fig. R4-1.

Fig. R4-1 Profiles of SW-damaged rim growth rate k versus the average normalized SW number flux (J_{avg}) and energy flux ($E_{f,avg}$) at different sampling sites, including Apollo 11, Apollo 16, Apollo 17, CE-5, and CE-6.

Excluding the CE-5 site, the remaining four sites exhibit an overall positive correlation between growth rate k and both SW number flux (J_{avg}) and energy flux ($E_{f,avg}$). Analysis of all five sampling sites showed correlation coefficients of -0.7717 and 0.1256 between SW-damaged rim growth rate k and SW fluxes J_{avg} , $E_{f,avg}$, respectively, indicating a significant negative correlation and weak positive correlation. Exclusion of the CE-5 outlier improved the correlation coefficients to 0.7757 and 0.7185 for the remaining four sites (CE-6, Apollo11, Apollo16, and Apollo17), revealing strong positive correlations. These findings highlight the anomalous characteristics of the CE-5 site, which implies potential limitations in our current lunar SW flux distribution model's capability to describe long-term (million-year time scale) SW irradiation patterns in CE-5 region. Thus, this study does not include comparative analysis between CE-5 and CE-6. The correlations observed among the remaining four sites (CE-6, Apollo11, Apollo16, Apollo17) demonstrate that: (1) Decadal-scale SW observations and numerical modeling are applicable to million-year time scales in these regions. (2) The current model effectively simulates the shielding effects of the Earth's magnetosphere on near-side solar wind flux. These findings suggest asymmetry in SW irradiation characteristics between the lunar nearside and farside.

References

1. Xie, L. *et al.* Global Hall MHD Simulations of the Solar Wind Implantation Flux on the Lunar Surface. *Planet. Sci. J.* 4, 218 (2023).

Page 5 Lines 202-Page 6 Line 235:

Influence of solar wind irradiation on SW-damaged rim growth rates

Fig. 3 Maps of the lunar surface showing average normalized solar wind (SW) number flux (J_{avg}) and energy flux ($E_{f,avg}$), alongside profiles of SW-damaged rim growth rate (k) versus J_{avg} and $E_{f,avg}$. (a, c) Lunar surface maps of J_{avg} and $E_{f,avg}$, derived from Hall MHD simulations by Xie et al.³ that incorporate latitude, Earth's magnetosphere, and lunar magnetic anomalies. (b, d) Profiles of SW-damaged rim growth rate k versus J_{avg} and $E_{f,avg}$ at different sampling sites, including Apollo 11, Apollo 16, Apollo 17, Chang'E-5 (CE-5), and Chang'E-6 (CE-6). Further details on J_{avg} and $E_{f,avg}$ can be found in the caption to Table 1. Note that the SW flux data in panels a and c are from Xie et al. 3

Based on the measured SW-damaged rim growth rates (k) and the simulations by Xie et al.³, we constructed correlation plots between average SW fluxes (the average normalized SW number flux J_{avg} and energy flux $E_{f,avg}$) and rim growth rate k across these sampling sites (Fig. 3b, d). Excluding the CE-5 site, the remaining four sites exhibit an overall positive correlation between rim growth rate k and both SW number flux J_{avg} and energy flux $E_{f,avg}$. Across all five sampling sites (Fig. 3b,d), the rim growth rate k shows correlation coefficients of -0.7717 and 0.1256 with the SW number flux J_{avg} and energy flux $E_{f,avg}$, respectively — a significant negative correlation and weak positive relationship. Upon excluding the CE-5 outlier, analyses of the remaining four sites (CE-6, Apollo 11, Apollo 16, and Apollo 17) reveal adjusted correlation coefficients of 0.7757 and 0.7185 between rim growth rate k and the SW number flux J_{avg} and energy flux $E_{f,avg}$, indicating strong positive correlations. This strong correlation suggests that SW fluxes are the primary factors influencing the rim growth rate. Notably, the stronger correlation between rim growth rate k and SW number flux J_{avg} (versus SW energy flux $E_{f,avg}$) suggests that SW number flux exerts a greater influence than SW energy flux on the SW-damaged rim growth. This interpretation is further reinforced by Apollo 11, Apollo 16, and Apollo 17 data: despite an about two-fold variation in energy flux $E_{f,avg}$ among these sites, their rim growth rate k values remain close, conclusively demonstrating that SW energy flux is secondary to SW number flux in governing rim growth kinetics.

In addition, the growth rate of SW-damaged rims, as derived from lunar soil measurements occurs over million-year timescales and thus reflects long-term SW flux conditions. By contrast, current observations and simulations operate on decade-scale timescales^{3,41}. The correlations observed among the remaining four sites (CE-6, Apollo11, Apollo16, Apollo17) have demonstrated that: (1) Applicability of decadal-scale SW observations and numerical modeling on million-year time scales in these regions. (2) The current model effectively simulates the shielding effects of the Earth's magnetosphere on the near-side SW fluxes. These findings suggest asymmetry in SW irradiation characteristics between lunar nearside and farside.

The anomalously high growth rate recorded in CE-5 regolith grains suggest that this area may have experienced stronger historical SW irradiation. Given the significant deviations of the CE-5's measured growth rate from both simulations³ and ARTEMIS spacecraft observations⁴¹, current simulations and observations likely underestimate the long-term ancient SW fluxes in CE-5 sampling site. These discrepancies might arise from the models' failure to incorporate more complex factors such as variations of solar activity and Earth's magnetic field, Earth wind, and magnetic field variations in lunar anomalous regions.

References

- 3. Xie, L. et al. Global hall MHD simulations of the solar wind implantation flux on the lunar surface. Planet. Sci. J. 4, 218 (2023).*
- 41. Nénon, Q. & Poppe, A. R. Bombardment of lunar polar crater interiors by out-of-ecliptic ions: ARTEMIS observations. Planet. Sci. J. 2, 116 (2021).*

Q5: The different slopes of track density over depth presented in Figure 3 are quite interesting and to my knowledge, this has not been reported before. The authors connect these slope differences to a different hardness of the SEP spectrum, potentially related to higher solar activity longer in the past than 2 Myr ago. An assumption on a different SEP exposure is not unreasonable, but the authors are not able to provide any evidence for the grains from Group 1 being exposed to SEPs at different times than those from Group 2. Conclusions on different solar activity at different times (and especially when a solar activity decrease would have occurred) cannot be made with the results presented in this study. Any connection to Earth's ocean temperatures in this context is completely speculative.

R5: Thanks for your comments. In light of the pronounced level of uncertainty in this section and with the aim of maintaining the manuscript's focus on the SW section, the discussion of SEP energy spectrum has been removed from the revised manuscript. This alteration also serves as a response to comments from other reviewers, guaranteeing that the discussion remains both robust and focused.

Q6 Minor comments:- Lines 51 & 177: The authors should cite recent reports from the ARTEMIS spacecraft that have shown SEP access to the Moon even deep in the magnetotail (<https://doi.org/10.1029/2023GL103990>, <https://doi.org/10.1029/2024GL110228>).

R6: Thanks for your reminder. The reference has been added in the revised manuscript.

Page 2 Lines 48-56:

In a typical 28-day lunar orbit, the Moon spends approximately 25% of its time (~7 days) within Earth's magnetotail⁷⁻¹⁰. During lunar daytime on the nearside, the central region spends roughly half its time within Earth's magnetotail. In contrast, the central region of the farside, when in daytime, remains continuously exposed to direct solar radiation². Earth's magnetic field and the lunar body can effectively shield low-energy (<100 keV) SW particles on the lunar nearside; however, this shielding effect diminishes considerably for higher-energy SW particles (>150 keV) and SEPs (>1 MeV)^{2,10,11}. Observations from the ARTEMIS spacecraft and other probes have revealed that SEPs can penetrate to the Moon deep within Earth's magnetotail^{13,14}.

References

- 2. Xu, X. et al. The energetic particle environment of the lunar nearside: SEP influence. *Astrophys. J.* 849, 151 (2017).*
- 7. Winglee, R. M. & Harnett, E. M. Radiation mitigation at the moon by the terrestrial magnetosphere. *Geophys. Res. Lett.* 34, (2007).*
- 8. Kallenrode, M.-B. *Space Physics: An introduction to plasmas and particles in the heliosphere and magnetospheres. (Springer Science & Business Media, 2013).**
- 9. Liuzzo, L., Poppe, A. R., Halekas, J. S., Simon, S. & Cao, X. Investigating the moon's interaction with the terrestrial magnetotail lobe plasma. *Geophys. Res. Lett.* 48, e2021GL093566 (2021).*
- 10. Liuzzo, L., Poppe, A. R. & Halekas, J. S. A statistical study of the moon's magnetotail plasma environment. *J. Geophys. Res. Space Phys.* 127, e2022JA030260 (2022).*
- 11. Huang, C.-L., Spence, H. E. & Kress, B. T. Assessing access of galactic cosmic rays at moon's orbit. *Geophys. Res. Lett.* 36, L09109 (2009).*
- 12. Case, A. W. et al. GCR access to the moon as measured by the CRaTER instrument on LRO. *Geophys. Res. Lett.* 37, L19101 (2010).*
- 13. Liuzzo, L., Poppe, A. R., Lee, C. O. & Angelopoulos, V. Solar energetic electron access to the moon within the terrestrial magnetotail and shadowing by the lunar surface. *Geophys. Res. Lett.* 51, e2024GL110228 (2024).*
- 14. Liuzzo, L., Poppe, A. R., Lee, C. O., Xu, S. & Angelopoulos, V. Unrestricted solar energetic particle access to the moon while within the terrestrial magnetotail. *Geophys. Res. Lett.* 50, e2023GL103990 (2023).*

Q7 Minor comments:- Line 105: In comparison to the description of the Chang'e-6 samples, context on Chang'E-5 mission and sampling location are missing. It should furthermore be described in more detail that the Chang'E-5 samples also represent newly analyzed samples specifically for this paper.

R7: We truly appreciate your reminder. In the revised manuscript, the relevant contents have been added as per your suggestion.

Page 3 Lines 90-100:

On December 1, 2020, the Chang'E-5 (CE-5) mission successfully returned 1.731 kg of lunar soil from a mid-latitude region on the Moon's nearside, the northeastern Oceanus Procellarum (51.916°W, 43.058°N)³⁴. On June 25, 2024, the Chang'E-6 (CE-6) mission successfully returned the first samples from the South Pole-Aitken (SPA) basin (41.625°S, 153.978°W) at lunar farside³⁵⁻³⁷, providing a unique opportunity to investigate the long-term solar particle irradiation environment at lunar farside and its differences from the nearside.

In this paper, we examine the SW-damaged rims and SEP tracks within the CE-5 and CE-6 samples, explore the long-term differences in irradiation environment between the nearside and farside of the Moon over millions of years, and study their impacts on the evolution of lunar soil grains.

METHODS

Samples

As mentioned earlier, although samples from both CE-5 and CE-6 missions originate from mare regions, their characteristics exhibit certain differences. The average true density of CE-5 samples is measured as 3.1952 g/cm³³⁴, whereas that of CE-6 samples is 3.035 g/cm³³⁵. In terms of chemical composition, compared to CE-5 samples, CE-6 samples exhibit a decrease in FeO (CE-5: 22.5 wt.%, CE-6: 17.2 wt.%), TiO₂ (CE-5: 5.0 wt.%, CE-6: 2.7 wt.%), but an increase in Al₂O₃ content (CE-5: 10.8 wt.%, CE-6: 14.3 wt.%). This is attributed to the fact that CE-6 samples have extremely low olivine content (CE-5: 5.7 wt.%, CE-6: 0.5 vol%), less ilmenite content (CE-5: 4.5 wt.%, CE-6: 1.6 vol%), and higher plagioclase content (CE-5: 30.1 wt.%, CE-6: 32.6 vol%)^{34,35}. The nomenclature for Chang'E-5 and Chang'E-6 lunar samples adheres to the established protocols, which are publicly accessible at: <https://moon.bao.ac.cn/moonSampleMode/index.html>.

References

34.Li, C. et al. Characteristics of the lunar samples returned by the Chang'E-5 mission. Natl. Sci. Rev. 9, nwab188 (2022).

35. Li, C. et al. Nature of the lunar far-side samples returned by the Chang'E-6 mission. Natl. Sci. Rev. 11, nwa328 (2024).

36. Yue, Z. et al. Geological context of the Chang'e-6 landing area and implications for sample analysis. The Innovation 5, (2024).

37. Zeng, X. et al. Landing site of the Chang'e-6 lunar farside sample return mission from the Apollo basin. Nat. Astron. 7, 1188–1197 (2023).

Q8 Minor comments:- Lines 224-225: The wording needs to be changed to explain more clearly what the authors want to say.

R8: Following the augmentation of data samples from CE-5 and CE-6, the corresponding results and discussions have been updated.

Q9 Minor comments:- Lines 231-234: The authors should reference studies from the Chandrayaan-1 spacecraft using H reflected from the lunar surface, which provide evidence for the extent of shielding at some magnetic anomalies (<https://doi.org/10.1029/2012JA017553>).

R9: In the revised manuscript, the reference has been added.

Page 4 Lines 169-Page 5 Lines 174:

Additionally, $E_{f,avg}$ is more strongly influenced by magnetic anomaly regions than I_{avg} , as lunar magnetic anomalies only partially shielded SW protons, particularly in weakly magnetized areas⁴². This partial shielding results in limited reductions in number flux within crustal magnetic fields due to SW penetration. However, deceleration and deflection during penetration lead to a decrease in energy flux along the normal direction, making energy flux more sensitive to the presence of magnetic anomalies³.

References

3. Xie, L. et al. Global hall MHD simulations of the solar wind implantation flux on the lunar surface. Planet. Sci. J. 4, 218 (2023).

42. Vorburger, A. et al. Energetic neutral atom observations of magnetic anomalies on the lunar surface. J. Geophys. Res. Space Phys. 117, 2012JA017553 (2012).

Q10 Minor comments:- Lines 235-237: Some context on the simulation work by Xie et al. is missing here. Which aspects of the model have been verified with measurements? Is there any way to estimate an uncertainty for the used number fluxes and energy fluxes? Especially regarding the previously mentioned issues with the uncertainties of the measurement results and the related conclusions, this would be helpful.

R10: The relevant content has been added in the revised manuscript. For the event on December 31, 2019, the difference of absolute values between the simulation results and the ARTEMIS satellite orbit observations is within 10% (Xie et al. ¹). Hence, we expect the relative uncertainties between near side and far side is smaller than 10%.

References

1. Xie, L. et al. Global hall MHD simulations of the solar wind implantation flux on the lunar surface. *Planet. Sci. J.* 4, 218 (2023).

Page 4 Lines 154-167:

Lunar surface SW flux is strongly influenced by latitude, Earth's magnetosphere, and lunar magnetic anomalies. Xie et al. ³ employed Hall MHD simulations to investigate SW irradiation conditions across the entire Moon, accounting for the abovementioned factors. Based on observational data from the ARTEMIS spacecraft and the Chang'E-4 (CE-4) rover during 01:00–08:00 UT on December 31, 2019, the model successfully simulated the solar wind shielding process of the lunar mini-magnetosphere (LMM). The absolute differences between the simulation results and ARTEMIS satellite observations were within 10%, confirming the model's validity ^{3,40}. In comparison with in situ measurements from spacecraft, simulations reveal that the spatial dependence of SW flux across the lunar surface—including magnetic anomaly regions, latitude effects and near/far side differences—shows relatively good agreement with data from the ARTEMIS spacecraft ⁴¹. Since the model is primarily constructed using recent short-term observations and does not incorporate the long-term evolution of solar activity, magnetic field intensity, etc., its applicability to million-year-scale solar wind irradiation on the lunar surface may be limited.

References

3. Xie, L. et al. Global hall MHD simulations of the solar wind implantation flux on the lunar surface. *Planet. Sci. J.* 4, 218 (2023).
40. Xie, L. et al. Multipoint observation of the solar wind interaction with strong lunar magnetic anomalies by ARTEMIS spacecraft and Chang'E-4 rover. *Astrophys. J. Lett.* 937, L5 (2022).

41.Nénon, Q. & Poppe, A. R. Bombardment of lunar polar crater interiors by out-of-ecliptic ions: ARTEMIS observations. Planet. Sci. J. 2, 116 (2021).

Q11 Minor comments:- Lines 240-247: As mentioned before, the ratio of rim growth slope to J_{avg} is almost twice as high for the CE5 samples. J_{avg} is actually the lowest for the CE5 sampling site.

R11: Based on the expanded data, the discussion in this section has been modified. The updated data corroborate that the SW-damaged rim growth rate within the CE-5 region is, in fact, the highest, which deviates from the current estimates derived from Xie et al.'s³ simulations. This implies that the SW flux models may fail to describe the SW irradiation conditions in the CE-5 region over extended timescales (millions of years). For further details, please refer to R4.

References

3. Xie, L. *et al.* Global Hall MHD Simulations of the Solar Wind Implantation Flux on the Lunar Surface. *Planet. Sci. J.* 4, 218 (2023).

Reviewer #2 (Remarks to the Author):

The paper has been improved by the authors in the process of revision and is now closer to acceptable quality. However, a further revision is still needed as detailed below.

Reply: Thank you for your constructive comments. In accordance with your suggestions, we have revised or removed the speculative statements in the revised manuscript. The point-by-point responses to your comments are as follows.

Q1: The paragraph about the correlation (lines 203+) should be removed as misleading. First, the CE5 point is identified as an outlier (l. 210) and should not be used in the correlation analysis. Second, the remaining 4-point correlation is, while yielding a high formal value of the Pearson's linear correlation coefficient, insignificant, viz. the p-values for the points shown in Fig.3b and 3d are 0.09 and 0.08 (formally insignificant). Moreover, these correlations are mostly based on the single CE-6 point, which lies higher than the Apollo-based points and has large error bars. If the CE-6 point is excluded, the remaining Apollo points don't exhibit any significant trend. Thus, the claimed relation is not well-established.

R1: This section has been modified as outlined in **R2**.

Q2: The last paragraph (l. 229+) also needs to be removed. The CE-5 point is identified as an outlier, and thus, no conclusions can be drawn based on its difference from other points. The conclusion that the present solar wind models are incorrect in that the model computations correctly predict all other data points but the CE-5 one, which is an outlier. Accordingly, the last two sentences of the Abstract should be removed, too.

R2: We have modified the relevant sections as follows:

Abstract

The long-term effects of Earth's magnetosphere on solar wind (SW) irradiation asymmetry between the lunar nearside and farside, and their implications for space weathering processes, remain poorly characterized. Here, we have measured irradiation exposure ages and SW induced amorphous rim thicknesses of individual grains from Chang'E-5 (CE-5) and Chang'E-6 (CE-6) lunar soils, from which rim growth rates are derived. Comparative analysis of SW irradiation

records from CE-5, CE-6, and Apollo (11, 16, 17) samples, combined with contemporary lunar surface SW flux simulations, reveals that CE-6 samples from the southern mid-latitude farside display rim growth rates comparable to or exceeding those of low-latitude Apollo samples. This supports the hypothesis of persistent irradiation asymmetry driven by Earth's magnetospheric shielding of the nearside. However, CE-5 samples from the northern mid-latitude nearside exhibit unexpectedly high rim growth rates. The underlying cause of this anomaly remains unclear and warrants further investigation.

Page 6 Line 220 - Page 7 Line 275

Influence of solar wind irradiation on SW-damaged rim growth rates

Based on the measured SW-damaged rim growth rates (k) and the simulations, we constructed correlation plots between average SW fluxes (the average normalized SW number flux J_{avg} and energy flux $E_{f,avg}$) and rim growth rate k across these sampling sites (Fig. 3b, d). Excluding the CE-5 site, the remaining four sites appear to exhibit a positive correlation between k and both J_{avg} and $E_{f,avg}$, with Pearson's correlation coefficients of 0.7757 and 0.7158 ($n = 4$), respectively. However, the corresponding p -values (0.2243 and 0.2815) indicate the correlations do not reach statistical significance at the conventional threshold ($p < 0.05$). This is primarily attributed to two factors: limited sample size due to the small number of sampling sites, and high data-point clustering caused by the geographic proximity of Apollo sampling sites, which leads to similar rim growth rates. Notably, the CE-6 sample from the mid-latitude farside exhibits a relatively higher k compared to the low-latitude nearside Apollo samples. This observation supports the hypothesis that shielding by Earth's magnetosphere on the nearside leads to a persistent asymmetry in SW irradiation between the lunar hemispheres.

Mechanistically, elevated proton-dominated SW flux enhances particle bombardment on mineral surfaces. During direct collisions, the transferred kinetic energy displaces lattice atoms, initiating collision cascades that propagate through the crystalline framework. These cascades predominantly produce vacancy-interstitial pairs, which accumulate when the rate of defect generation exceeds the material's intrinsic capacity for defect annealing—namely, the thermal mobility and recombination of defects under ambient conditions. With continued irradiation, the increasing defect density drives progressively lattice disorder: isolated atomic displacements evolve into interconnected defect networks, ultimately facilitating cooperative atomic rearrangements that transform the crystalline structure into an amorphous phase. Our Stopping and Range of Ions in Matter (SRIM)⁵⁰ simulations show that higher-energy ions penetrate more deeply into the target material and exhibit elevated vacancy generation rate (Table. S3). When SW flux intensifies—particularly through simultaneous increase in both particle number and energy—the combined effects typically accelerate the formation rate of amorphous layers on mineral surfaces and extend the amorphization depth.

Table S3. Impact of injected ion parameters on damage depth and vacancy generation rate. SRIM simulations reveal that as the incident energy of injected ions increases, both the damage depth (penetration depth into the material) and the vacancy generation rate (number of vacancies per ion) rise.

Injection Energy (keV)	Incident Angle (Degree)	Damage Depth (nm)	Vacancy Generation Rate (Vacancies/Ion)
0.3	0.0	12.0 nm	0.3
1.0	0.0	30.3 nm	1.8
3.0	0.0	70.9 nm	4.2

Additionally, the anomalously high growth rate of the SW-damaged rim observed in CE-5 regolith grains suggests that this region may have experienced stronger SW irradiation than previously expected. Xu et al.⁵¹ reported a significantly higher abundance of SW-derived water in CE-5 samples compared to Apollo samples, attributing this enrichment to the lower surface temperatures at the CE-5 site, which suppress SW-hydrogen outgassing. While this thermal effect likely plays a role, evidence from multiple studies suggests that the amorphous layer in minerals acts as the primary reservoir for SW-derived water⁵¹⁻⁵⁴. Bradley et al.⁵³ employed transmission electron microscopy (TEM) and valence electron energy-loss spectroscopy (VEELS) to examine the amorphous rims of interstellar dust particles (IDPs), revealing that SW protons interact with oxygen in silicate minerals to form hydroxyl (-OH) and/or molecular water (H₂O), which are subsequently stored within vesicular structures in the amorphous rims. Their results further demonstrated that SW irradiation induces structural disorder, potentially facilitating both the formation and retention of water. In a separate study, Zeng et al.⁵⁴ conducted ion implantation experiments using deuterium ions as analogs for SW protons to irradiate olivine, pyroxene, and plagioclase. They found that over 73% of the implanted deuterium was retained within a fully amorphous rim ~70 nm thick, whereas only ~25% extended to depths of ~190 nm, where amorphization was incomplete. Similarly, Zhou et al.⁵² investigated the depth distribution of SW-derived water in CE-5 mineral samples and found that it is predominantly hosted within amorphous rims, with only a minor fraction preserved in the underlying crystalline matrix. As mentioned earlier, enhanced SW flux promotes the formation of amorphous layers that serve as primary reservoirs for implanted water. Therefore, the elevated SW flux at the CE-5 sampling site, relative to Apollo sites, likely represents an additional contributing factor to the higher SW-derived water content observed in CE-5 materials. As for the elevated SW flux at the CE-5 site, one potential explanation is the influence of local topography—exposure on a sloped surface may have increased the effective incident particle flux. However, the exact cause remains to be determined and warrants further investigation. Taken together, these findings indicate that variations in SW irradiation environments and space weathering processes across the lunar surface may be more complex than previously recognized.

References

- 50. Ziegler, J. F. et al. SRIM – The Stopping and Range of Ions in Matter (2010). Nucl. Instrum. Methods Phys. Res. Sect. B Beam Interact. Mater. At. 268, 1818–1823 (2010).*
- 51. Xu, Y. et al. High Abundance of Solar Wind-derived Water in Lunar Soils from the Middle Latitude. Proc. Natl. Acad. Sci. 119, e2214395119 (2022).*
- 52. Zhou, C. et al. Chang'E-5 Samples Reveal High Water Content in Lunar Minerals. Nat. Commun. 13, 5336 (2022).*
- 53. Bradley, J. P. et al. Detection of Solar Wind-produced Water in Irradiated Rims on Silicate*

Minerals. Proc. Natl. Acad. Sci. 111, 1732–1735 (2014).

54. *Zeng, X. et al. Depth Profiling of Implanted D+ in Silicates: Contribution of Solar Wind Protons to Water in the Moon and Terrestrial Planets. Astron. Astrophys. 691, A275 (2024).*

Q3: The authors overuse self-citations. For example, the first 5 references should be related to the process of solar wind energetic particle acceleration, but instead cite publications about measurements at the lunar surface. General references to solar energetic particles and solar wind are missing. The following ones are recommended: Desai & Giacalone (2016, doi: 10.1007/s41116-016-0002-5) and Vidotto (2018, doi: 10.1007/s41116-021-00029-w).

R3: The relevant references have been added as per your suggestion.

Page2 Lines 39 - 41: During solar activities, particles are accelerated primarily by mechanisms associated with solar flares and coronal mass ejection (CME)-driven shock waves and are subsequently transported through the interplanetary magnetic field to the lunar surface ¹⁻⁸.

References

1. *Xu, Z. et al. First Solar Energetic Particles Measured on the Lunar Far-side. Astrophys. J. Lett. 902, L30 (2020).*
2. *Xu, X. et al. The Energetic Particle Environment of the Lunar Nearside: SEP Influence. Astrophys. J. 849, 151 (2017).*
3. *Xie, L. et al. Global Hall MHD Simulations of the Solar Wind Implantation Flux on the Lunar Surface. Planet. Sci. J. 4, 218 (2023).*
4. *Luo, P. et al. First Measurements of Low-energy Cosmic Rays on the Surface of the Lunar Farside from Chang'E-4 Mission. Sci. Adv. 8, eabk1760 (2022).*
5. *Poppe, A. R. et al. Solar Energetic Particle Track-production Rates at 1 au: Comparing in Situ Particle Fluxes with Lunar Sample-derived Track Densities. Astrophys. J. Lett. 958, L35 (2023).*
6. *Desai, M. & Giacalone, J. Large Gradual Solar Energetic Particle Events. Living Rev. Sol. Phys. 13, 3 (2016).*
7. *Vidotto, A. A. The Evolution of the Solar Wind. Living Rev. Sol. Phys. 18, 3 (2021).*
8. *Ó Fionnagáin, D. & Vidotto, A. A. The Solar Wind in Time: a Change in the Behaviour of Older Winds? Mon. Not. R. Astron. Soc. 476, 2465–2475 (2018).*

Q4: Line 114-115: “flare tracks”  “particle tracks”.

R4: The phrase “flare tracks” has been revised to “particle tracks” in the revised manuscript as suggested.

Q5: Line 140+: Solar activity is not expected to vary on the mega-year time scale. It varies on the synoptic through secular timescales (solar variability) and also on the giga-year scale (solar evolution), but not on mega-year scale. There are numerous pieces of evidence that solar activity is roughly stable over millions of years, including lunar samples. This needs to be re-formulated.

R5: The section has been modified as follows:

Page 4 Lines 133 - 145: The distribution of SW-damaged rim thickness against exposure age in lunar grains from different sampling sites is shown in Fig. 2, where the two variables exhibit a positive correlation overall. In the interval with exposure ages less than 4 Myr, this relationship can be approximated by a linear model. The data exhibit a degree of scatter. Notably, the CE-5 data points are relatively clustered, probably because the samples were collected from a single basalt unit with a relatively young formation age (~2.0 Gyr)⁴³. In contrast, the CE-6 (~2.8 Gyr)⁴⁴ samples and those from the Apollo missions⁴⁵—including Apollo 11 (3.6–3.9 Gyr), Apollo 16 (3.4–3.8 Gyr), and Apollo 17 (3.7–3.8 Gyr)—originate from regions with older formation ages, more complex geological contexts (e.g., mare-highland boundary), and a higher likelihood of incorporating exotic components. Moreover, due to the Apollo missions' samples being collected through extravehicular activities (EVAs) and considering the significant distances between different stations, this could also lead to the collection of samples with varying exposure histories. These combined factors likely account for the broader dispersion observed in the data.

References

43. Che, X. et al. Age and Composition of Young Basalts on the Moon, Measured from Samples Returned by Chang'e-5. *Science* 374, 887–890 (2021).
44. Zhang, Q. W. L. et al. Lunar Farside Volcanism 2.8 Billion Years Ago from Chang'e-6 Basalts. *Nature* (2024) doi:10.1038/s41586-024-08382-0.
45. Hiesinger, H. & Head, J. W. New Views of Lunar Geoscience: An Introduction and Overview. in *New Views of the Moon* 1–82 (2006).

Q6: Line 159: “the absolute difference ... within 10%”. If it is given in %, it must be the relative difference.

R6: Thank you for pointing this out. As noted by another reviewer, direct comparison between SW flux simulation data at the lunar surface and ARTEMIS observations is currently not feasible due to the influence of local magnetic anomalies. The relevant part has been modified as follows:

Page 4 Line 160 - Page 5 Line 179: Lunar surface SW flux is strongly influenced by latitude, Earth's

magnetosphere, and lunar magnetic anomalies. Xie et al.³ employed Hall MHD simulations to investigate SW irradiation conditions across the entire Moon, accounting for the abovementioned factors. Using data from the ARTEMIS spacecraft and the Chang'E-4 (CE-4) rover during 01:00–08:00 UT on December 31, 2019, the model successfully reproduced the interaction between the SW and strong lunar magnetic anomalies. Key features observed in-situ were accurately captured, including distinct boundary structures and wake dynamics consistent with plasma and magnetic field profiles along the ARTEMIS P2 trajectory. The model also reproduced shock-associated density enhancements and magnetic field rotations in close agreement with ARTEMIS measurements. Furthermore, it simulated the formation of a downstream plasma cavity, consistent with observed plasma depletion linked to magnetotail dynamics. Notably, the simulations replicated the observed reduction in SW kinetic energy and flux, with deceleration rates and shielding efficiencies consistent with CE-4's energetic neutral atom (ENA) flux data, confirming partial surface shielding by the magnetic anomaly. These results validate the model's capability to resolve the complex dynamics of SW-lunar magnetic anomaly interactions^{3,46}. The lunar surface maps depicting the distributions of average normalized SW number flux (I_{avg}) and energy flux ($E_{f,avg}$) are presented in Fig. 3a, c. In comparison with in-situ measurements from spacecraft, simulations reveal that the spatial dependence of SW flux across the lunar surface—including magnetic anomaly regions, latitude effects and near/far side differences—shows relatively good agreement with data from the ARTEMIS spacecraft⁴⁷.

References

- 3. Xie, L. et al. Global Hall MHD Simulations of the Solar Wind Implantation Flux on the Lunar Surface. Planet. Sci. J. 4, 218 (2023).*
- 46. Xie, L. et al. Multipoint Observation of the Solar Wind Interaction with Strong Lunar Magnetic Anomalies by ARTEMIS Spacecraft and Chang'E-4 Rover. ApJL 937, L5 (2022).*
- 47. Nénon, Q. & Poppe, A. R. Bombardment of Lunar Polar Crater Interiors by Out-of-ecliptic Ions: ARTEMIS Observations. Planet. Sci. J. 2, 116 (2021).*

Reviewer #3 (Remarks to the Author):

In their revised manuscript, Liu et al. present a largely reworked paper on electron microscopy analysis of Chang'e 5 and Chang'e 6 lunar samples. Compared to the previous draft, detailed studies of the solar energetic particle (SEP) spectra are removed as well as any connection to Earth climate studies. Instead, the paper now focuses solely on the formation of weathering features in the lunar regolith across different sampling sites. In order to improve the significance of their findings, the authors have significantly increased the number of studied samples for both Chang'e 5 and Chang'e 6, which is highly commendable. Weathering features are characterized by comparing SEP track densities with the formation of amorphized rims from solar wind (SW) exposure. When compared to Apollo samples, the Chang'e 6 farside samples show a similar weathering behavior, while the Chang'e 5 samples tend to have faster rim formation. This appears to be different with the behavior in other samples, especially as simulations predict lower SW exposure than at other some of the other locations.

Compared to the previous version of the manuscript, the methodological aspects of the study are improved. As similarly stated in my previous review, the selection of sample analysis methods is highly appropriate and through the increased number of samples the findings are more robust. The observation that Chang'e 5 rims have a tendency of being formed more quickly is interesting to report to the community and this finding appears robust. Both for amorphous rim and SEP track formation, previous work has presented models that explain these processes qualitatively but not fully quantitative yet. As a result, we do not fully understand the details of the formation processes and why some samples form larger rims than others. The submitted manuscript is also not able to present a more detailed explanation for these processes, and the Chang'e-5 sample group behavior in particular. An extended discussion of the latter would thus significantly improve the paper.

Reply: Thank you for your constructive comments. We have revised the manuscript in response to your comments and provide a detailed point-by-point reply below.

Q1: The CE-5 sample group sticks out in covering a narrower exposure age, while both the Apollo 17 and the CE-6 group have multiple samples that fall into the range of the CE-5 samples in Figure 5. Are there systematic differences between sample collection of CE-5 and CE-6 that could potentially explain the smaller CE-5 variety? If not, do the authors have any other explanation for differences in the CE-5 sampling site that could explain a difference in rim formation?

R1: The dataset exhibits some scatter. CE-5 points are notably more clustered, likely due to the relatively young age (~2.0 Gyr) of the local mare unit. In contrast, CE-6 (~2.8 Gyr) and the Apollo

sites—Apollo 11 (3.6–3.9 Gyr), Apollo 16 (3.4–3.8 Gyr), and Apollo 17 (3.7–3.8 Gyr)—are associated with older, more geologically complex terrains, resulting in a broader range of exposure ages. Apollo 17 shows the greatest dispersion, likely reflecting the geological heterogeneity at the mare–highland boundary where it was collected.

Moreover, we have further included a discussion of the high growth rate of CE-5 SW-damage rim. The CE-5 samples exhibit significantly higher contents of SW-derived water than Apollo samples from lower latitudes¹ (Fig. R1). This enrichment has been attributed to the lower surface temperatures at the CE-5 site, which suppress SW-hydrogen outgassing. However, multiple studies suggest that the amorphous layer in minerals serves as the primary reservoir for SW-derived water. When SW flux intensifies—particularly through concurrent increases in both particle number and energy flux—the formation of amorphous layers on mineral surfaces is typically accelerated, and the depth of amorphization increases. Therefore, the elevated SW flux likely constitutes an additional contributing factor to the enhanced water content observed in CE-5 materials. One possible explanation for this intensified SW flux is the influence of local topography: exposure on a sloped surface may have increased the effective incident particle flux. In light of these considerations, we do not consider the elevated flux at the CE-5 sampling site to be caused by systematic errors and have substantially revised the discussion regarding the anomalously high growth rate of the SW-damaged rim in CE-5 samples. The revised content is presented below.

[REDACTED]

Fig. R1. [Redacted] (*Proc. Natl. Acad. Sci. U.S.A.* 119 (51) e2214395119).

References

1. Xu, Y. et al. High Abundance of Solar Wind-derived Water in Lunar Soils from the Middle Latitude. *Proc. Natl. Acad. Sci.* 119, e2214395119 (2022).

Page 4 Lines 133 - 158: *The distribution of SW-damaged rim thickness against exposure age in lunar grains from different sampling sites is shown in Fig. 2, where the two variables exhibit a positive correlation overall. In the interval with exposure ages less than 4 Myr, this relationship can be approximated by a linear model. The data exhibit a degree of scatter. Notably, the CE-5 data points are relatively clustered, probably because the samples were collected from a single basalt unit with a relatively young formation age (~2.0 Gyr)⁴³. In contrast, the CE-6 (~2.8 Gyr)⁴⁴ samples and those from the Apollo missions⁴⁵—including Apollo 11 (3.6–3.9 Gyr), Apollo 16 (3.4–3.8 Gyr), and Apollo 17 (3.7–3.8 Gyr)—originate from regions with older formation ages, more complex geological contexts (e.g., mare-highland boundary), and a higher likelihood of incorporating exotic components. Moreover, due to the Apollo missions' samples being collected through extravehicular activities (EVAs) and considering the significant distances between different stations, this could also lead to the collection of samples with varying exposure histories. These combined factors likely account for the broader dispersion observed in the data. At the same exposure age around 1.22 Myr, CE-5 samples exhibit the greatest SW-damaged rim thickness, followed by CE-6, with Apollo samples showing the smallest values (Fig. 2). To quantify and compare SW irradiation levels across these sampling sites, the SW-damaged rim growth rate was defined. To extract the growth rate from different sites, a linear model $D = k \cdot t$ was used to fit SW-damaged rim thickness data against exposure age for these five sampling sites. Here, k represents the growth rate of the SW-damaged rim (in nm/Myr), D is the rim thickness (in nm), t is the exposure age (in Myr), the fitting error uses the standard deviation (σ). Table 1 shows that growth rates are highest in CE-5 samples, intermediate in CE-6, and lowest in Apollo samples. This variability can be attributed to local SW irradiation conditions. Note that both CE-5 and Apollo sampling sites are located on the lunar near side: CE-5 site lies at mid-northern latitude, while Apollo sites are near the equator. In contrast, CE-6 sampling site is situated at mid-southern latitudes on the lunar far side. Detailed geographic coordinates for these sites are provided in Table 1.*

Page 6 Line 220 - Page 7 Line 275

Influence of solar wind irradiation on SW-damaged rim growth rates

Based on the measured SW-damaged rim growth rates (k) and the simulations, we constructed correlation plots between average SW fluxes (the average normalized SW number flux J_{avg} and energy flux $E_{f,avg}$) and rim growth rate k across these sampling sites (Fig. 3b, d). Excluding the CE-5 site, the remaining four sites appear to exhibit a positive correlation between k and both J_{avg} and $E_{f,avg}$, with Pearson's correlation coefficients of 0.7757 and 0.7158 ($n = 4$), respectively. However, the corresponding p -values (0.2243 and 0.2815) indicate the correlations do not reach statistical significance at the conventional threshold ($p < 0.05$). This is primarily attributed to two factors: limited sample size due to the small number of sampling sites, and high data-point clustering caused by the geographic proximity of Apollo sampling sites, which leads to similar rim growth rates. Notably, the CE-6 sample from the mid-latitude farside exhibits a relatively higher k compared to the low-latitude nearside Apollo samples. This observation supports the hypothesis that shielding by Earth's magnetosphere on the nearside leads to a persistent asymmetry in SW irradiation between the lunar hemispheres.

Mechanistically, elevated proton-dominated SW flux enhances particle bombardment on mineral surfaces. During direct collisions, the transferred kinetic energy displaces lattice atoms,

initiating collision cascades that propagate through the crystalline framework. These cascades predominantly produce vacancy-interstitial pairs, which accumulate when the rate of defect generation exceeds the material's intrinsic capacity for defect annealing—namely, the thermal mobility and recombination of defects under ambient conditions. With continued irradiation, the increasing defect density drives progressively lattice disorder: isolated atomic displacements evolve into interconnected defect networks, ultimately facilitating cooperative atomic rearrangements that transform the crystalline structure into an amorphous phase. Our Stopping and Range of Ions in Matter (SRIM)⁵⁰ simulations show that higher-energy ions penetrate more deeply into the target material and exhibit elevated vacancy generation rate (Table. S3). When SW flux intensifies—particularly through simultaneous increase in both particle number and energy—the combined effects typically accelerate the formation rate of amorphous layers on mineral surfaces and extend the amorphization depth.

Table S3. Impact of injected ion parameters on damage depth and vacancy generation rate. SRIM simulations reveal that as the incident energy of injected ions increases, both the damage depth (penetration depth into the material) and the vacancy generation rate (number of vacancies per ion) rise.

Injection Energy (keV)	Incident Angle (Degree)	Damage Depth (nm)	Vacancy Generation Rate (Vacancies/Ion)
0.3	0.0	12.0	0.3
1.0	0.0	30.3	1.8
3.0	0.0	70.9	4.2

Additionally, the anomalously high growth rate of the SW-damaged rim observed in CE-5 regolith grains suggests that this region may have experienced stronger SW irradiation than previously expected. Xu et al.⁵¹ reported a significantly higher abundance of SW-derived water in CE-5 samples compared to Apollo samples, attributing this enrichment to the lower surface temperatures at the CE-5 site, which suppress SW-hydrogen outgassing. While this thermal effect likely plays a role, evidence from multiple studies suggests that the amorphous layer in minerals acts as the primary reservoir for SW-derived water⁵¹⁻⁵⁴. Bradley et al.⁵³ employed transmission electron microscopy (TEM) and valence electron energy-loss spectroscopy (VEELS) to examine the amorphous rims of interstellar dust particles (IDPs), revealing that SW protons interact with oxygen in silicate minerals to form hydroxyl (-OH) and/or molecular water (H₂O), which are subsequently stored within vesicular structures in the amorphous rims. Their results further demonstrated that SW irradiation induces structural disorder, potentially facilitating both the formation and retention of water. In a separate study, Zeng et al.⁵⁴ conducted ion implantation experiments using deuterium ions as analogs for SW protons to irradiate olivine, pyroxene, and plagioclase. They found that over 73% of the implanted deuterium was retained within a fully amorphous rim ~70 nm thick, whereas only ~25% extended to depths of ~190 nm, where amorphization was incomplete. Similarly, Zhou et al.⁵² investigated the depth distribution of SW-derived water in CE-5 mineral samples and found that it is predominantly hosted within amorphous rims, with only a minor fraction preserved in the

underlying crystalline matrix. As mentioned earlier, enhanced SW flux promotes the formation of amorphous layers that serve as primary reservoirs for implanted water. Therefore, the elevated SW flux at the CE-5 sampling site, relative to Apollo sites, likely represents an additional contributing factor to the higher SW-derived water content observed in CE-5 materials. As for the elevated SW flux at the CE-5 site, one potential explanation is the influence of local topography—exposure on a sloped surface may have increased the effective incident particle flux. However, the exact cause remains to be determined and warrants further investigation. Taken together, these findings indicate that variations in SW irradiation environments and space weathering processes across the lunar surface may be more complex than previously recognized.

References

43. Che, X. et al. *Age and Composition of Young Basalts on the Moon, Measured from Samples Returned by Chang'e-5*. *Science* 374, 887–890 (2021).
44. Zhang, Q. W. L. et al. *Lunar Farside Volcanism 2.8 Billion Years Ago from Chang'e-6 Basalts*. *Nature* (2024) doi:10.1038/s41586-024-08382-0.
45. Hiesinger, H. & Head, J. W. *New Views of Lunar Geoscience: An Introduction and Overview*. in *New Views of the Moon* 1–82 (2006).
50. Ziegler, J. F. et al. *SRIM – The Stopping and Range of Ions in Matter* (2010). *Nucl. Instrum. Methods Phys. Res. Sect. B Beam Interact. Mater. At.* 268, 1818–1823 (2010).
51. Xu, Y. et al. *High Abundance of Solar Wind-derived Water in Lunar Soils from the Middle Latitude*. *Proc. Natl. Acad. Sci.* 119, e2214395119 (2022).
52. Zhou, C. et al. *Chang'E-5 Samples Reveal High Water Content in Lunar Minerals*. *Nat. Commun.* 13, 5336 (2022).
53. Bradley, J. P. et al. *Detection of Solar Wind-produced Water in Irradiated Rims on Silicate Minerals*. *Proc. Natl. Acad. Sci.* 111, 1732–1735 (2014).
54. Zeng, X. et al. *Depth Profiling of Implanted D+ in Silicates: Contribution of Solar Wind Protons to Water in the Moon and Terrestrial Planets*. *Astron. Astrophys.* 691, A275 (2024).

Q2: Did the composition of Chang'e 5 and 6 plagioclase grains differ significantly? Keller et al. (2021) specifically studied anorthite grains, how do the EDS composition results compare here?

R2: The vast majority of feldspar on the Moon belongs to the plagioclase group. In the samples from Chang'E-5 (CE-5) and Chang'E-6 (CE-6), EDS results indicate that the feldspar is predominantly composed of the two calcium-rich varieties: bytownite ($\text{Na}_{0.3}\text{Ca}_{0.7}\text{Al}_{1.3}\text{Si}_{2.7}\text{O}_8$) and anorthite ($\text{CaAl}_2\text{Si}_2\text{O}_8$). The differences in feldspar composition within the CE-5 and CE-6 samples are minimal, primarily reflected in the mineral proportions (CE-5: bytownite 90%, anorthite 10%; CE-6: bytownite 50%, anorthite 50%) and average compositions (CE-5: An range 76.1-97.6, bytownite average composition $\text{An}_{84.5}\text{Ab}_{14.6}\text{Or}_{0.9}$, anorthite average composition $\text{An}_{92.5}\text{Ab}_{7.3}\text{Or}_{0.2}$; CE6: An range 82.9-99.7, bytownite average composition $\text{An}_{86.6}\text{Ab}_{12.9}\text{Or}_{0.6}$, anorthite average

composition $An_{94.3}Ab_{5.4}Or_{0.3}$). In contrast, the Apollo samples exhibit calcium enrichment, with An ranging from 80.5 to 95.7. Additionally, SRIM simulations of H and Fe ion implantation in anorthite and bytownite (Table S4) show that both ions produce nearly identical damage depths in the two minerals. Overall, the feldspar in the CE-5 and CE-6 samples is highly similar to that in the Apollo samples, demonstrating significant comparability.

METHODS

Samples

As mentioned earlier, although samples of both CE-5 and CE-6 missions originate from mare regions, their characteristics exhibit certain differences. The average true density of CE-5 samples is measured as 3.1952 g/cm^3 ³⁷, higher than that of CE-6 samples (3.035 g/cm^3)³⁸. In terms of chemical composition, compared to CE-5 samples, CE-6 samples exhibit a decrease in FeO (CE-5: 22.5 wt.%, CE-6: 17.2 wt.%), TiO_2 (CE-5: 5.0 wt.%, CE-6: 2.7 wt.%), but an increase in Al_2O_3 content (CE-5: 10.8 wt.%, CE-6: 14.3 wt.%). This is attributed to the fact that CE-6 samples have extremely low olivine content (CE-5: 5.7 wt.%, CE-6: 0.5 vol%), less ilmenite content (CE-5: 4.5 wt.%, CE-6: 1.6 vol%), and higher plagioclase content (CE-5: 30.1 wt.%, CE-6: 32.6 vol%)^{37,38}. The nomenclature for Chang'E-5 and Chang'E-6 lunar samples adheres to the established protocols, which are publicly accessible at: <https://moon.bao.ac.cn/moonSampleMode/index.html>.

In the samples from CE-5 and CE-6, EDS results (Table. S2) indicate that the feldspar is predominantly composed of the two calcium-rich varieties: bytownite ($Na_{0.3}Ca_{0.7}Al_{1.3}Si_{2.7}O_8$) and anorthite ($CaAl_2Si_2O_8$). The differences in feldspar composition within the CE-5 and CE-6 samples are minimal, primarily reflected in the mineral proportions (CE-5: bytownite 90%, anorthite 10%; CE-6: bytownite 50%, anorthite 50%) and average compositions (CE-5: An range 76.1-97.6, bytownite average composition $An_{84.5}Ab_{14.6}Or_{0.9}$, anorthite average composition $An_{92.5}Ab_{7.3}Or_{0.2}$; CE6: An range 82.9-99.7, bytownite average composition $An_{86.6}Ab_{12.9}Or_{0.6}$, anorthite average composition $An_{94.3}Ab_{5.4}Or_{0.3}$). In contrast, the Apollo samples exhibit calcium enrichment, with An ranging from 80.5 to 95.7. Additionally, SRIM⁵⁰ simulations of H and Fe ion implantation in anorthite and bytownite (Table S4) show that both ions produce nearly identical damage depths in the two minerals. Overall, the feldspar in the CE-5 and CE-6 samples is highly similar to that in the Apollo samples, demonstrating significant comparability.

Table S2. EDS analysis of particle An and mineral types.

Grain Number	An	Mineral
CE5-PL1	90	Anorthite
CE5-PL2	93	Anorthite
CE5-PL3	91	Anorthite
CE5-PL4	88	Bytownite
CE5-PL5	87	Bytownite

CE5-PL6	83	Bytownite
CE5-PL7	82	Bytownite
CE6-PL1	92	Anorthite
CE6-PL2	91	Anorthite
CE6-PL3	91	Anorthite
CE6-PL4	86	Bytownite
CE6-PL5	89	Bytownite
CE6-PL6	89	Bytownite
CE6-PL7	81	Bytownite
CE6-PL8	86	Bytownite

Table S4. SRIM-simulated damage depths of Fe and H ions implanted into bytownite ($\text{Na}_{0.3}\text{Ca}_{0.7}\text{Al}_{1.3}\text{Si}_{2.7}\text{O}_8$, density 2.72 g/cm³) and anorthite ($\text{CaAl}_2\text{Si}_2\text{O}_8$, density 2.75 g/cm³). For a given ion and implantation energy, the damage depths are nearly identical between the two minerals.

Injected ion: Fe			Injected ion: H		
Damage depth (μm)			Damage depth (μm)		
Ion Energy (MeV/nuc)	Bytownite	Anorthite	Ion Energy (keV)	Bytownite	Anorthite
0.107143	3.2838	3.2876	0.100	5.4	5.5
0.116071	3.4173	3.4212	0.110	5.8	5.9
0.125000	3.5504	3.5543	0.120	6.2	6.3
0.142857	3.7965	3.8005	0.130	6.5	6.6
0.160714	4.0314	4.0355	0.140	6.9	6.9
0.178571	4.2355	4.2396	0.150	7.3	7.3
0.196429	4.4390	4.4432	0.160	7.6	7.6
0.214286	4.6320	4.6363	0.170	8.0	8.0
0.232143	4.8047	4.8090	0.180	8.3	8.3
0.250000	4.9872	4.9914	0.200	9.0	8.9
0.267857	5.1494	5.1537	0.225	9.7	9.7
0.285714	5.3114	5.3157	0.250	10.5	10.5
0.303571	5.4733	5.4776	0.275	11.3	11.3
0.321429	5.6350	5.6393	0.300	12.1	12.1
0.357143	5.9396	5.9439	0.325	12.8	12.8
0.401786	6.3055	6.3098	0.350	13.6	13.5
0.446429	6.6609	6.6651	0.375	14.2	14.2
0.491071	7.0058	7.0100	0.400	15.0	15.0
0.535714	7.3504	7.3445	0.450	16.4	16.4
0.580357	7.6747	7.6788	0.500	17.7	17.8
0.625000	8.0088	8.0028	0.550	19.1	19.2
0.669643	8.3327	8.3267	0.600	20.4	20.5

0.714286	8.6464	8.6404
0.803571	9.2888	9.2726
0.892857	9.9104	9.8939
0.982143	10.5313	10.5147
1.071429	11.1517	11.1249
1.160714	11.7617	11.7347
1.250000	12.3613	12.3342
1.428571	13.5955	13.5479
1.607143	14.8072	14.7590
1.785714	16.0369	15.9782
1.964286	17.2551	17.1860
2.142857	18.4820	18.4025
2.321429	19.7079	19.6181
2.500000	20.9532	20.8531
2.678571	22.2081	22.0977
2.857143	23.4725	23.3618
3.035714	24.7465	24.6256
3.214286	26.0403	25.9191
3.571429	28.7195	28.5674
4.017857	32.1676	31.9943
4.464286	35.7100	35.5100
4.910714	39.3400	39.1100
5.357143	43.0700	42.8300
5.803571	46.9100	46.6400
6.250000	50.8600	50.5600
6.696429	54.9000	54.5800
7.142857	59.0600	58.7100
8.035714	67.9100	67.4800
8.928571	77.1600	76.6600
9.821429	86.8500	86.2800
10.000000	88.7800	88.2000

0.650	21.8	21.8
0.700	23.0	23.1
0.800	25.6	25.6
0.900	28.1	28.1
1.000	30.5	30.5
1.100	32.9	32.9
1.200	35.3	35.3
1.300	37.6	37.5
1.400	39.9	39.8
1.500	42.0	42.0
1.600	44.3	44.1
1.700	46.4	46.4
1.800	48.5	48.5
2.000	52.7	52.6
2.250	57.8	57.6
2.500	62.7	62.6
2.750	67.6	67.5
3.000	72.3	72.2
3.250	77.0	76.9
3.500	81.6	81.4
3.750	86.0	85.9
4.000	90.4	90.3
4.500	98.9	98.9
5.000	107.2	107.1
5.500	115.2	115.1
6.000	123.0	122.9
6.500	130.6	130.6
7.000	137.9	138.0
8.000	152.1	152.1
9.000	165.7	165.7
10.000	178.8	178.7

References

37. Li, C. et al. Characteristics of the Lunar Samples Returned by the Chang'E-5 Mission. Natl. Sci. Rev. 9, nwab188 (2022).
38. Li, C. et al. Nature of the Lunar Far-side Samples Returned by the Chang'E-6 Mission. Natl. Sci. Rev. 11, nwae328 (2024).
50. Ziegler, J. F., Ziegler, M. D. & Biersack, J. P. SRIM – The Stopping and Range of Ions in Matter (2010). Nucl. Instrum. Methods Phys. Res. Sect. B Beam Interact. Mater. At. 268, 1818–1823 (2010).

Q3: Stating that the simulations of the surface exposure by Xie et al. is consistent with ARTEMIS data within 10% could be misleading. The main result that the present work uses from the simulations is the surface precipitation flux and energy flux, specifically accounting for the effect of local magnetic anomalies. As an orbiter, these effects cannot be directly studied with the ARTEMIS spacecraft, especially considering the effects of local magnetic anomalies.

R3: Thank you for the helpful comment. We have revised the relevant section accordingly, as follows:

Page 4 Lines 160- Page 5 Line 179: Lunar surface SW flux is strongly influenced by latitude, Earth's magnetosphere, and lunar magnetic anomalies. Xie et al.³ employed Hall MHD simulations to investigate SW irradiation conditions across the entire Moon, accounting for the abovementioned factors. Using data from the ARTEMIS spacecraft and the Chang'E-4 (CE-4) rover during 01:00–08:00 UT on December 31, 2019, the model successfully reproduced the interaction between the SW and strong lunar magnetic anomalies. Key features observed in situ were accurately captured, including distinct boundary structures and wake dynamics consistent with plasma and magnetic field profiles along the ARTEMIS P2 trajectory. The model also reproduced shock-associated density enhancements and magnetic field rotations in close agreement with ARTEMIS measurements. Furthermore, it simulated the formation of a downstream plasma cavity, consistent with observed plasma depletion linked to magnetotail dynamics. Notably, the simulations replicated the observed reduction in SW kinetic energy and flux, with deceleration rates and shielding efficiencies consistent with CE-4's energetic neutral atom (ENA) flux data, confirming partial surface shielding by the magnetic anomaly. These results validate the model's capability to resolve the complex dynamics of SW-lunar magnetic anomaly interactions^{3,46}. The lunar surface maps depicting the distributions of average normalized SW number flux (J_{avg}) and energy flux ($E_{f,avg}$) are presented in Fig. 3a, c. In comparison with in-situ measurements from spacecraft, simulations reveal that the spatial dependence of SW flux across the lunar surface—including magnetic anomaly regions, latitude effects and near/far side differences—shows relatively good agreement with data from the ARTEMIS spacecraft⁴⁷.

References

- 3. Xie, L. et al. Global Hall MHD Simulations of the Solar Wind Implantation Flux on the Lunar Surface. Planet. Sci. J. 4, 218 (2023).*
- 46. Xie, L. et al. Multipoint Observation of the Solar Wind Interaction with Strong Lunar Magnetic Anomalies by ARTEMIS Spacecraft and Chang'E-4 Rover. ApJL 937, L5 (2022).*
- 47. Nénon, Q. & Poppe, A. R. Bombardment of Lunar Polar Crater Interiors by Out-of-ecliptic Ions: ARTEMIS Observations. Planet. Sci. J. 2, 116 (2021).*

Q4: It appears to be a minor detail for the study here, but the arguments in the response letter given for the usage of the radial velocity v_r^3 to calculate the energy flux are not fully convincing. Damage theories like the Kinchin-Pease model connect the number of vacancies created by ion impacts to the deposited energy. Simulation programs such as SRIM and SDTrimSP also predict the number of formed vacancies by impacting ions to be largely independent of angle of incidence, but instead the impact energy is important.

R4: Thanks for your comments. Upon consideration, we contend that using the traditional $V_r V^2$ for calculating solar wind (SW) energy flux is more appropriate. Following careful verification, we confirm that the normalized SW energy flux values presented in the manuscript were derived from $V_r V^2$, rather than the V_r^3 definition reported in Xie et al.¹. To visualize the differences between these two formulations, Table R1 provides normalized SW energy flux calculations for each sampling site and the lunar farside subsolar point using both definitions. Comparison of the distributions of normalized SW energy flux derived from $V_r V^2$ and V_r^3 is shown in Fig. R2.

Table R1. Growth rates of solar wind (SW)-damaged rims and average normalized SW energy flux ($E_{f,avg}$) derived from $V_r V^2$ and V_r^3 at selected lunar sampling sites.

Sampling site	Chang'E-5	Chang'E-6	Apollo 11	Apollo 16	Apollo 17	Farside subsolar point
Location coordinates	43.06°N 51.92°W	41.64°S 153.99°W	0.67°N 23.47°E	8.97°S 15.50°E	20.19°S 30.77°E	0° 180°
SW-damaged rim growth rate k (nm/Myr)	55.96 ±10.82	33.21 ±6.16	25.10 ±1.76	23.52 ±5.36	25.40 ±2.32	–
Average normalized SW energy flux $E_{f,avg}$ (Calculated using $V_r V^2$)	0.1785	0.2309	0.2036	0.1116	0.1985	0.3058
Average normalized SW energy flux $E_{f,avg}$ (Calculated using V_r^3)	0.0659	0.0873	0.1242	0.0635	0.1099	0.2108

Fig. R2. Comparison of normalized SW energy flux distributions derived from $V_r V^2$ and V_r^3 . The left and right panels show results calculated using $V_r V^2$ and V_r^3 , respectively.

References

1. Xie, L. et al. *Global Hall MHD Simulations of the Solar Wind Implantation Flux on the Lunar Surface. Planet. Sci. J. 4, 218 (2023).*

Reviewer #2 (Remarks to the Author):

I am fine with the revised version of the paper. Only a few very minor clarifications of the text are requested before the final acceptance recommendation.

Reply: We would like to express our sincere gratitude for your thoughtful and constructive review of our manuscript. Your insightful comments have been invaluable in improving the quality and clarity of our work.

In response to suggestions from another reviewer, we have revised the *Introduction* section to place greater emphasis on solar wind–related alteration while de-emphasizing the role of solar energetic particles. Additionally, we have expanded the *Discussion* section to provide a more comprehensive analysis of the potential factors contributing to the elevated SW-damaged rim growth rates observed at the Chang'E-5 site.

Please refer to the revised manuscript for detailed changes. Our point-by-point responses to your comments are provided below.

Q1: Line 31: “underlying mechanisms of”  “exact reasons for”.

R1: Thank you for your suggestion. As recommended, we have revised “underlying mechanisms of” to “exact reasons for”. We appreciate your thoughtful input.

Page 1 Lines 32-33:

“The exact reasons for this anomaly remain unclear and warrant further investigation.”

Q2: “During solar activities”  “During solar activity events”.

R2: Thank you for your suggestion regarding the phrase “During solar activities”. In response to another reviewer’s comment that the original *introduction* section overemphasized solar energetic particle (SEP) events while underrepresenting the solar wind (SW), we have revised the *Introduction* section to better reflect the study’s focus on SW–driven processes. As part of this revision, the relevant sentence has been removed. We sincerely appreciate your careful reading and thoughtful feedback.

Q3: Line 234: Please remove “mechanistically”.

R3: Thank you for pointing this out. We agree that the term “mechanistically” was not essential in this context and have removed it in the revised manuscript. Additionally, the previous version did not comprehensively describe the formation mechanism of solar wind (SW)–damaged amorphous rims; this has been corrected in the current revision.

Pages 6-7 Line 248-270:

“Elevated proton-dominated SW irradiation enhances particle bombardment on mineral surfaces. Upon penetration, low-energy protons (~1 keV) primarily lose energy through electronic stopping—namely, ionization and excitation of target electrons⁶⁸. The deposited energy can break chemical bonds and generate localized electron-hole pairs. Non-radiative recombination of these excitations or energy transfer via localized electronic excitations may induce atomic displacements, particularly in low-binding-energy lattice environments. Although such displacements are typically isolated, continuous irradiation leads to the accumulation of point defects such as vacancies and interstitials. These defects may further agglomerate into clusters or form interconnected networks, especially when the rate of damage accumulation exceeds the material’s intrinsic annealing capacity, which is governed by defect mobility and recombination efficiency under ambient conditions. In parallel, a smaller but non-negligible fraction of the proton’s energy is transferred directly to lattice atoms through nuclear stopping. These direct collisions initiate displacement cascades, producing vacancy–interstitial pairs that further contribute to lattice damage. While each cascade event is localized, their cumulative effect enhances the density and spatial connectivity of defects. With continued irradiation, the synergistic interplay between electronically and collision-induced defects drives the progressive disordering of the crystalline lattice, eventually facilitating cooperative atomic rearrangements and the formation of an amorphous phase in the near-surface region. Our Stopping and Range of Ions in Matter (SRIM)⁶⁹ simulations show that higher-energy ions penetrate more deeply into the target material and exhibit elevated vacancy generation rate (Table. S3). When SW irradiation intensifies—particularly through simultaneous increase in both particle number and energy—the combined effects typically accelerate the formation rate of amorphous layers on mineral surfaces and extend the amorphization depth.”

Q4: Line 249: “grains, suggests that this region may have”  “grains, if confirmed, may suggest that this region has”.

R4: Thank you for your suggestion. We have revised the sentence to “grains, if confirmed, may suggest that this region has”, as recommended.

Page 7 Lines 271-273:

“Additionally, the anomalously high growth rate of the SW-damaged rim observed in CE-5 regolith grains, if confirmed, may suggest that this region has experienced stronger SW irradiation than previously expected.”

Thanks again.

Reviewer #3 (Remarks to the Author):

Regarding the updated version of the manuscript, I appreciate the modifications and the response by the authors, providing some helpful additional background. Unfortunately, there are still some issues both in the old and the newly added text.

Especially arguments in the newly added sections on an increased solar wind flux at the Chang'e-5 sampling site are not convincing. The authors state surface slopes as a potential reason, but do not give any further details, which makes it hard to follow why this would be the case. Such an assumption would need a more quantitative calculation and evidence that relevant conditions are actually present at the Chang'e-5 sampling site to allow increased solar wind fluxes to the surface. Ultimately, the finding that the CE-5 samples are an outlier in the rim growth rate appears robust, but it remains an open question why, which the present study cannot answer at the moment.

Reply: Thank you for your constructive feedback. We have revised the manuscript accordingly and provide a detailed point-by-point response below.

The following are my specific remarks:

Q1: Introduction: The newly added section at the beginning is selective to specific events of coronal mass ejections and solar-energetic particle event, while the solar wind is barely mentioned. As solar-wind alteration is the more important topic for this paper, the introduction should be adjusted to reflect this.

R1: Thank you for your valuable comment. In response, we have substantially revised the *Introduction* section to place greater emphasis on solar wind (SW)-driven alteration, which is the central focus of this study. Content related to solar energetic particle (SEP) events has been significantly reduced or removed to maintain a clear focus. We hope the updated *Introduction* section more accurately reflects the motivation and scope of the work.

Pages 2-3 Lines 40-110:

“Solar wind (SW) ions, consisting mainly of low-energy H and He ions with typical energies around 1 keV/amu¹⁻⁴, can induce structural damage in lunar mineral grains, leading to the formation of SW-damaged amorphous rims⁵⁻⁸. This kind of rims record cumulative SW-induced radiation damage and serve as reliable proxies for long-term SW exposure^{7,9}. The growth rates of such rims provide critical constraints for evaluating the contribution of SW to the overall space weathering processes on the lunar surface^{7,10}. In contrast, solar energetic particles (SEPs)¹¹⁻¹⁴—particularly high-energy heavier ions like Fe—can penetrate more deeply into mineral grains and produce damage tracks^{7,15}. These tracks serve as chronological marker that allow the exposure ages of individual grains to be estimated^{7,16,17}. Although SEPs are not the main focus of this study, the grain

exposure ages they help determine are essential for quantifying the long-term growth rates of SW-damaged rims.

While individual SEP events can exhibit strong anisotropy depending on heliospheric magnetic connectivity, the cumulative SEP flux is generally considered near-isotropic across the lunar surface over geological timescale^{18–20}. In comparison, the flux of lower-energy SW ions can be significantly modulated by various shielding mechanisms, including Earth's magnetosphere (especially the magnetotail), the bow shock at the SW–magnetosphere interface, direct occlusion by the Moon itself, and localized magnetic anomalies^{21–28}. In a typical 28-day lunar orbit, the Moon spends approximately 25% of its time (~7 days) within Earth's magnetosphere, including up to ~4 days in the magnetotail, where the incident SW flux is largely suppressed due to magnetic shielding^{24,29}. During nearside daytime, the central region may spend roughly half its daytime within the magnetotail, experiencing partial shielding from incoming SW ions. In contrast, the sunlit farside remains continuously exposed to direct SW irradiation, without the geomagnetic protection³⁰. Earth's magnetosphere can efficiently shield solar particles with energies up to ~4 MeV when the Moon is within it during SEP events. Outside this protective environment, under direct SW exposure, the lunar body provides only limited occlusion of particles up to ~100 keV. For particles exceeding ~150 keV, the shadowing effect becomes negligible, as fluxes within the lunar wake approximate those in the ambient SW³⁰. Moreover, although Earth's bow shock offers some protection, it can also inject additional low-energy particles (below a few hundred keV) into lunar orbit, exerting a significantly greater influence on the nearside relative to the farside^{26,30}. Additionally, local magnetic anomalies on the Moon may form mini-magnetospheres that partially shield the surface by deflecting incoming SW particles, thereby reducing their flux and mitigating space weathering in certain regions^{21–23,25}. This spatially variable shielding landscape gives rise to potential long-term asymmetries in SW irradiation between the lunar nearside and farside, which may in turn be recorded as distinct space weathering signatures in the surface regolith.

Lunar space weathering is primarily driven by the irradiation of charged particles—including the SW, SEPs, galactic cosmic rays (GCRs)^{31–35}—as well as micrometeorite impacts^{36–40} and episodic contributions from Earth's magnetospheric particles (often referred to as “Earth wind”), particularly when the Moon traverses the geomagnetic tail^{20,41–47}. Over geological timescales, these processes collectively modify the physical and chemical properties of the lunar regolith, leading to progressive surface maturation^{39,44}. Micrometeorite impacts continuously “garden” the regolith, cyclically excavating and burying grains⁴⁸. Lunar surface grains retain distinct structural and chemical signatures of irradiation by SW and SEPs. Low-energy (~1 keV/amu) SW H/He ions induce radiation damage primarily within the outermost ~100 nm of grains, forming a characteristic SW-damaged rim⁷. In contrast, high-energy SEPs (e.g., Fe ions ranging from <1 MeV/nucleon to ~100 MeV/nucleon) penetrate to depths of a few millimeters, generating lattice damage and leaving latent particle tracks throughout the regolith^{7,43,49–51}. Surface grains exposed for extended periods

accumulate substantial radiation damage⁴³, reflecting persistent regional variations in SW irradiation^{42,43,52}. These complementary records enable investigations into the spatial variability of long-term SW irradiation across the lunar surface.

To date, lunar soil samples returned by the Apollo missions have been employed to investigate the solar particle radiation differences across various regions⁴¹⁻⁴³. Keller et al.⁷ investigated the relationship between SW-damaged rims and exposure ages in lunar soils from various regions, finding no significant differences in the growth rates of SW-damaged rims among samples. These findings suggest that seemingly minor variations in SW radiation exist across different regions^{43,53}. However, these studies are limited to the low latitudes on the nearside of the Moon, leaving the condition on the farside and higher latitudes largely unknown. To enhance our understanding of the long-term changes in the SW irradiation environment across different regions of the Moon and its effects on lunar soil evolution over a million-year timescale, an analysis of the SW-damaged rims and SEP track records in lunar soils from the lunar farside and at high-latitudes regions is essential for gaining further insights. On December 1, 2020, the Chang'E-5 (CE-5) mission successfully returned 1.731 kg of lunar soil from a mid-latitude region on the Moon's nearside, the northeastern Oceanus Procellarum (51.916°W, 43.058°N)⁵⁴. On June 25, 2024, the Chang'E-6 (CE-6) mission successfully returned the first samples from the South Pole-Aitken (SPA) basin (41.625°S, 153.978°W) at the lunar farside⁵⁵⁻⁵⁷, providing a unique opportunity to investigate the long-term solar particle irradiation environment at the lunar farside and its differences from the nearside.

In this work, we examine the SW-damaged rims and SEP tracks within the CE-5 and CE-6 samples, explore the long-term differences in SW irradiation environment between the nearside and the farside of the Moon over millions of years, and study their impacts on the evolution of lunar soil grains.”

Q2: Line 176: As a response to one of my previous comments, the authors added a paragraph about the agreement of the Hall-MHD simulations. However, a statement regarding the agreement with ARTEMIS on precipitation still remains. As stated before, precipitation cannot be directly observed from the ARTEMIS orbiters, which is why the precipitation maps presented in Figure 3 cannot be verified with this spacecraft dataset. The authors should remove this statement or clarify what they want to say here, as well as specify, which part of reference 47 they are citing.

R2: Thank you for pointing this out again, and we apologize for not addressing the issue adequately in the previous revision. As you rightly noted, precipitation cannot be directly observed by the ARTEMIS orbiters. To avoid misinterpretation, we have removed the corresponding statement from

the revised manuscript. We appreciate your careful review and helpful clarification.

Page 5 Lines 173-196:

*“Lunar surface SW flux is strongly influenced by latitude, Earth’s magnetosphere, and lunar magnetic anomalies. Xie et al. ²² employed Hall MHD simulations to investigate SW irradiation conditions across the entire Moon, accounting for the abovementioned factors. Using data from the ARTEMIS spacecraft ⁶⁶ and the Chang’E-4 (CE-4) rover during 01:00–08:00 UT on December 31, 2019, the model successfully reproduced the interaction between the SW and strong lunar magnetic anomalies. Key features observed in situ were accurately captured, including distinct boundary structures and wake dynamics consistent with plasma and magnetic field profiles along the ARTEMIS P2 trajectory. The model also reproduced shock-associated density enhancements and magnetic field rotations in close agreement with ARTEMIS measurements. Furthermore, it simulated the formation of a downstream plasma cavity, consistent with observed plasma depletion linked to magnetotail dynamics. Notably, the simulations replicated the observed reduction in SW kinetic energy and flux, with deceleration rates and shielding efficiencies consistent with CE-4’s energetic neutral atom (ENA) flux data, confirming partial surface shielding by the magnetic anomaly. These results validate the model’s capability to resolve the complex dynamics of SW-lunar magnetic anomaly interactions ^{22,23}. The lunar surface maps depicting the distributions of average normalized SW number flux (J_{avg}) and energy flux ($E_{(f,avg)}$) are presented in Fig. 3a, c. **Note** that the normalized SW energy flux is defined as $E = NV_r V^2 / (N_{sw} V_{sw}^3)$ in this work, where N , V , and V_r denote the simulated SW number density, total velocity, and radial velocity component along the lunar radius, respectively, while N_{sw} and V_{sw} represent parameters of the undisturbed upstream SW. This definition contrasts with that employed in Xie et al. ²². Please refer to Table 1 for more details. Since the model is primarily constructed using recent short-term observations and does not incorporate the long-term evolution of solar activity, magnetic field intensity, etc., its applicability to million-year-scale SW irradiation on the lunar surface may be limited.”*

Q3: Discussion: The newly added section, which suggest higher water abundances could possibly be related to higher solar-wind precipitation, appears highly speculative. At two points the authors mention “the elevated SW flux at the CE-5 site” as if this was shown here, but there is no convincing evidence for this to be necessarily the case. Rather it could also be possible that surface effects, such as the mentioned lower temperature, affect both water abundance and rim growth in similar ways, for example through different annealing rates of damaged rims. An extended, more convincing discussion would be needed to support the conclusion that the CE-5 sample site is exposed to an on average higher SW flux.

R3: Thank you for your thoughtful and constructive comments regarding the interpretation of the anomalously high growth rate of solar wind (SW)-damaged amorphous rims observed in Chang'E-5 (CE-5) grains, and for pointing out the speculative nature of attributing this enhancement solely to elevated SW flux due to local topographic geometry. In the revised manuscript, we have substantially restructured and expanded the relevant discussion to adopt a more cautious and balanced tone. We now explicitly acknowledge that the hypothesis of enhanced SW flux at the CE-5 site remains speculative and is not directly demonstrated by current observations. To address your concerns, we have revised the text to emphasize that multiple environmental and structural factors—each with varying degrees of plausibility—may contribute to the observed rim growth, and that their relative contributions are presently difficult to constrain.

Specifically, the revised discussion now includes:

- Enhanced regolith porosity, which can increase the spatial density of SW–grain interactions and promote amorphization, as suggested by the simulations of Szabo et al. (2022)⁷³ and supported by the in situ porosity measurements from Hu et al. (2025)⁷⁴;
- Local surface slope effects, which can increase the effective incident SW flux on inclined surfaces by reducing the solar zenith angle and enlarging the projected area facing the incoming particles;
- Suppressed thermal annealing at lower surface temperatures, which may enhance the preservation and accumulation of irradiation-induced damage;

We now present these mechanisms as potentially synergistic, rather than attributing the observed effects to SW flux alone. The revised summary explicitly states that, while porosity appears to be one plausible driver, the potential roles of slope, thermal influences, and other yet-unidentified factors cannot be ruled out. We have also removed or revised wording that may have implied a definitive enhancement of SW flux at the CE-5 site.

We appreciate your critical perspective, which helped us strengthen the rigor and clarity of our interpretations. We hope the revised discussion now more appropriately reflects the uncertainties involved and provides a more robust basis for future investigation.

Pages 7-9 Lines 271-348:

“Additionally, the anomalously high growth rate of the SW-damaged rim observed in CE-5 regolith grains, if confirmed, may suggest that this region has experienced stronger SW irradiation than previously expected. Xu et al.⁷⁰ reported a significantly higher abundance of SW-derived water in CE-5 samples compared to Apollo samples, attributing this enrichment to the lower surface temperatures at the CE-5 site, which suppress SW-hydrogen outgassing. While this thermal effect likely plays a role, evidence from multiple studies suggests that the amorphous layer in minerals acts as the primary reservoir for SW-derived water⁷⁰⁻⁷³. Bradley et al.⁷¹ used transmission electron microscopy (TEM) and valence electron energy-loss spectroscopy (VEELS) to examine the

amorphous rims of interstellar dust particles (IDPs), revealing that SW protons interact with oxygen in silicate minerals to form hydroxyl (-OH) and/or molecular water (H₂O), which are subsequently stored within vesicular structures in the amorphous rims. Their findings further demonstrated that SW irradiation induces structural disorder, potentially facilitating both the formation and retention of water. In a separate study, Zeng et al.⁷² conducted ion implantation experiments using deuterium ions as analogs for SW protons to irradiate olivine, pyroxene, and plagioclase. They found that over 73% of the implanted deuterium was retained within a fully amorphous rim ~70 nm thick, whereas only ~25% extended to depths of ~190 nm, where amorphization was incomplete. Similarly, Zhou et al.⁷³ investigated the depth distribution of SW-derived water in CE-5 mineral samples and found that it is predominantly hosted within amorphous rims, with only a minor fraction preserved in the underlying crystalline matrix. As discussed above, enhanced SW irradiation can facilitate the formation of amorphous layers that act as primary reservoirs for implanted water. Therefore, the elevated SW irradiation at the CE-5 sampling site, relative to Apollo sites, likely represents an additional contributing factor to the higher SW-derived water content observed in CE-5 materials. Such environmental differences—particularly elevated SW irradiation and lower surface temperatures at the CE-5 site—may not only contribute to the higher SW-derived water content, but also offer a potential explanation for the enhanced development of SW-damaged rims observed in CE-5 grains, as further discussed below.

One plausible explanation for the enhanced effective SW irradiation at the CE-5 site involves regolith porosity, which likely modulates SW–grain interactions at the microscale. Simulations by Szabo et al. (2022)⁷⁴ demonstrated that highly porous regolith enhances ion scattering and increases the spatial distribution of ion–grain interactions. Such structures may allow SW ions to penetrate deeper into the near-surface region and interact with a larger number of mineral grains, effectively increasing the local exposure of grain surfaces to incident particles. This elevated interaction density enhances the accumulation of irradiation-induced damage at grain boundaries, thereby promoting faster formation and thickening of SW-damaged amorphous rims over time. These simulation results are in line with the in situ findings of Hu et al. (2025)⁷⁵, who reported that the porosity at the CE-5 site is higher than that at CE-6, despite sample-based analyses suggesting otherwise—a discrepancy that likely results from mechanical disturbance during sample return. This elevated porosity may have contributed to the enhanced development of SW-damaged rims at the CE-5 site. It could also help explain the observed differences between CE-5 and Apollo sites.

Another contributing factor to the enhanced effective SW irradiation at the CE-5 site is local surface slope. A steeper slope can effectively reduce the solar zenith angle and enlarge the projected area facing the incoming SW, thereby increasing the effective flux of incident particles. While terrain-related enhancement does not alter the incident SW flux itself, it can increase the effective flux received by sloped surfaces, thereby elevating energy deposition onto individual grains and accelerating the formation of amorphous rims. This mechanism may partly account for both the

increased growth rate of SW-damaged rims and the higher SW-derived water content observed in CE-5 samples relative to those from the Apollo missions. It may also contribute to the higher rim growth rate at CE-5 compared to CE-6. Notably, due to severe erosion of the lunar surface at the CE-5 and CE-6 sampling sites by the landers' engine plumes⁷⁶⁻⁸⁰, precise local topographic slope data prior to plume-induced erosion are currently unavailable.

In addition, local thermal environments may influence the efficiency of amorphous rim formation by modulating the accumulation of SW irradiation-induced lattice damage. Although CE-5 and CE-6 are located at similar latitudes and share nearly identical surface temperature conditions—average (~226 K), maximum (~361 K), and minimum (~89 K)—Apollo sites tend to experience slightly higher maximum (~384–396 K) and minimum (~93–97 K) temperatures⁸¹. While the thermal sensitivity of amorphous rim formation remains poorly constrained, elevated temperatures could partially anneal irradiation-induced defects before they accumulate to the threshold needed for amorphization. Experimental studies demonstrate the critical role of temperature in mediating the balance between irradiation-induced lattice damage and thermal annealing⁸². At keV ion energies, temperature significantly affects defect dynamics. In Ge crystals irradiated with 25–600 keV ions, amorphization proceeds more readily at cryogenic than at room temperature, underscoring the role of thermally activated defect recombination in mitigating lattice disorder⁸³. At higher ion energies (MeV scale), similar thermally driven recovery processes are observed. For example, in Fa_{12} olivine, the threshold for amorphization under heavy-ion irradiation increases markedly near ~500 K, indicative of the onset of dynamic annealing processes⁸⁴. The surface temperatures at the CE-5 and CE-6 sampling sites are generally lower than those at the Apollo sites, which may be more conducive to the growth of SW-damaged amorphous rims. However, when considered alongside the influence of regolith porosity, the combined effect of temperature and porosity remains poorly constrained. In addition, the potential long-term impact of thermal cycling on the lunar surface is not well understood. Both aspects warrant further investigation.

In summary, while regolith porosity appears to be one plausible driver of the anomalously rapid growth of SW-damaged amorphous rims at the CE-5 site, potential contributions from the local surface slope and local thermal environment cannot be entirely ruled out. Other contributing factors may also exist but yet to be identified. Despite these efforts, current interpretations remain qualitative and largely inferential. Quantitative investigations are required to disentangle these factors and to clarify the physical drivers underlying the observed enhancement.”

References

74. Szabo, P. et al. Deducing lunar regolith porosity from energetic neutral atom emission. *Geophys. Res. Lett.* 49, e2022GL101232 (2022)

75. Hu, T. et al. In-suit Bulk Density of Lunar Regolith Obtained by Morphology Reconstruction with Delicate Excavation in CE-6/CE-5 Sampling Area. in (Universität Münster in Münster, Germany, 2025). doi:10.5281/zenodo.15675324.
76. Zhang, H. et al. The Investigation of Plume-Regolith Interaction and Dust Dispersal during Chang'E-5 Descent Stage. *Aerospace* 9, (2022).
77. You, J. et al. Unveiling the mechanics of lunar regolith erosion through analysis of CE-4 and CE-5 landing images and fluid simulation. *Acta Astronaut.* 208, 343–354 (2023).
78. Wang, C. et al. Exploring the effects of the rocket exhaust of the Chang'E-5 lander on the lunar regolith using LROC NAC and landing camera images. *Icarus* 403, 115649 (2023).
79. You, J. et al. Analysis of plume-lunar surface interaction and soil erosion during the Chang'E-4 landing process. *Acta Astronaut.* 185, 337–351 (2021).
80. Kim, J., Kim, J. & Lee, B. J. A survey of research on lunar dust dispersal due to rocket plume impingement. *Acta Astronaut.* 227, 67–95 (2025).
81. Williams, J.-P., Paige, D. A., Greenhagen, B. T. & Sefton-Nash, E. The global surface temperatures of the Moon as measured by the Diviner Lunar Radiometer Experiment. *Icarus* 283, 300–325 (2017).
82. Harries, D. & Langenhorst, F. The mineralogy and space weathering of a regolith grain from 25143 Itokawa and the possibility of annealed solar wind damage. *Earth Planets Space* 66, 163 (2014).
83. Impellizzeri, G., Mirabella, S. & Grimaldi, M. Ion implantation damage and crystalline-amorphous transition in *Ge. Appl. Phys. A* 103, 323–328 (2011).
84. Wang, L., Gong, W., Wang, S. & Ewing, R. C. Comparison of ion-beam irradiation effects in X₂YO₄ 561 compounds. *J. Am. Ceram. Soc.* 82, 3321–3329 (1999).

Further Remarks:

Q4: Abstract: “[...] reveals that CE-6 samples from the southern mid-latitude farside display rim growth rates comparable to or exceeding those of low-latitude Apollo samples. This supports the hypothesis of persistent irradiation asymmetry driven by Earth’s magnetospheric shielding of the nearside.”

This is inconsistent reasoning, if rates would be comparable then that would mean no asymmetry. This should be adapted.

R4: Thank you for your insightful comment. As you rightly noted, stating that the rim growth rates at the CE-6 site are “comparable to” those at Apollo sites does not convincingly support a hemispheric asymmetry in solar wind (SW) irradiation. In the revised manuscript, we have clarified that the rim growth rates observed in CE-6 samples exceed those from the low-latitude nearside Apollo samples. This updated statement more appropriately aligns with the hypothesis that Earth’s magnetospheric shielding reduces SW exposure on the nearside.

We also acknowledge that a higher modeled SW flux at the CE-6 site alone is not sufficient to explain the elevated rim growth rates, which are likely influenced by multiple environmental and structural factors. These include:

- **Local topography**, which can enhance the effective SW flux by increasing surface exposure through favorable slope orientation.
- **Regolith porosity**, which modulates ion implantation efficiency by affecting the microscale distribution of SW–grain interactions;
- **Surface temperature**, a key external control on defect annealing, which regulates the efficiency of damage recovery via thermal atomic motion;

To reflect these complexities, we have revised the relevant sections of both the abstract and main text to adopt a more cautious tone and avoid overstating the role of magnetospheric shielding. Please refer to the revised **Abstract** and our **response to Q3** for further details.

Page 1 Lines 20-33:

ABSTRACT:

*“The long-term effects of Earth’s magnetosphere on solar wind (SW) irradiation asymmetry between the lunar nearside and farside, and their implications for space weathering processes, remain poorly characterized. Here, we **measure** exposure ages and SW-induced amorphous rim thicknesses of individual grains from **the** Chang’E-5 (CE-5) and Chang’E-6 (CE-6) lunar soils **to derive** rim growth rates. Comparative analysis of SW irradiation records from CE-5, CE-6, and Apollo (11, 16, 17) **samples reveals** that CE-6 **grains** from the southern mid-latitude farside **exhibit higher** rim growth rates **than those from the** low-latitude **nearside** Apollo **sites**. This **trend aligns with simulated lunar surface SW fluxes and is consistent with the hypothesis that reduced SW exposure on the nearside, due to Earth’s magnetospheric shielding, may contribute to a persistent hemispheric asymmetry in SW irradiation**. However, CE-5 samples from the northern mid-latitude nearside **yield unexpectedly high rim growth rates, suggesting the potential involvement of additional local factors**. The **exact reasons for** this anomaly remain unclear and need further investigation.”*

Q5: Line 64: It should say “matures”.

R5: We appreciate your careful reading and thank you for pointing out this issue. In revising the *Introduction* section to better emphasize solar wind–related effects, we have rephrased the relevant sentence.

Pages 2-3 Lines 75-89:

“Lunar space weathering is primarily driven by the irradiation of charged particles—including the SW, SEPs, galactic cosmic rays (GCRs)^{31–35}—as well as micrometeorite impacts^{36–40} and episodic contributions from Earth’s magnetospheric particles (often referred to as “Earth wind”), particularly when the Moon traverses the geomagnetic tail^{20,41–47}. Over geological timescales, these processes collectively modify the physical and chemical properties of the lunar regolith, leading to progressive surface maturation^{39,44}. Micrometeorite impacts continuously “garden” the regolith, cyclically excavating and burying grains⁴⁸. Lunar surface grains retain distinct structural and chemical signatures of irradiation by SW and SEPs. Low-energy (~1 keV/amu) SW H/He ions induce radiation damage primarily within the outermost ~100 nm of grains, forming a characteristic SW-damaged rim⁷. In contrast, high-energy SEPs (e.g., Fe ions ranging from <1 MeV/nucleon to ~100 MeV/nucleon) penetrate to depths of a few millimeters, generating lattice damage and leaving latent particle tracks throughout the regolith^{7,43,49–51}. Surface grains exposed for extended periods accumulate substantial radiation damage⁴³, reflecting persistent regional variations in SW irradiation^{42,43,52}. These complementary records enable investigations into the spatial variability of long-term SW irradiation across the lunar surface.”

Q6: Line 65: It should be “regolith grains undergo a gardening process”.

R6: Thank you for pointing this out. Please refer to our **response to Q5** for details.

Q7: Line 81/82: There is a word missing in this sentence.

R7: Thank you for pointing this out. Upon careful review, we identified that the sentence lacked a main verb. We have revised it to improve clarity and grammatical correctness. The updated version now reads as follows:

Page 3 Lines 90-106:

*“To date, lunar soil samples returned by the Apollo missions have been employed to investigate the solar particle radiation differences across various regions⁴¹⁻⁴³. Keller et al.⁷ investigated the relationship between SW-damaged rims and exposure ages in lunar soils from various regions, finding no significant differences in the growth rates of SW-damaged rims among samples. **These findings suggest that seemingly minor variations in SW radiation exist across different regions**^{43,53}. However, these studies are limited to the low latitudes on **the** nearside of the Moon, leaving the condition on the farside and higher latitudes largely unknown. To enhance our understanding of the long-term changes in the SW irradiation environment **across different regions of the Moon** and its effects on lunar soil evolution over a million-year timescale, an analysis of the SW-damaged rims and **SEP** track records in lunar soils from the lunar farside and at high-latitudes regions is essential for gaining further insights. On December 1, 2020, the Chang’E-5 (CE-5) mission successfully returned 1.731 kg of lunar soil from a mid-latitude region on the Moon’s nearside, the northeastern Oceanus Procellarum (51.916°W, 43.058°N)⁵⁴. On June 25, 2024, the Chang’E-6 (CE-6) mission successfully returned the first samples from the South Pole-Aitken (SPA) basin (41.625°S, 153.978°W) at **the** lunar farside⁵⁵⁻⁵⁷, providing a unique opportunity to investigate the long-term solar particle irradiation environment at **the** lunar farside and its differences from the nearside.”*

Q8: Line 85: “throughout the Moon”, this wording should be changed

R8: Thank you for your helpful suggestion. In response, we have revised the phrase “throughout the Moon” to “across different regions of the Moon” to better reflect the spatial variability of solar wind irradiation across the lunar surface. The revised sentence now reads:

Page 3 Lines 90-106:

*“To date, lunar soil samples returned by the Apollo missions have been employed to investigate the solar particle radiation differences across various regions⁴¹⁻⁴³. Keller et al.⁷ investigated the relationship between SW-damaged rims and exposure ages in lunar soils from various regions, finding no significant differences in the growth rates of SW-damaged rims among samples. **These findings suggest that seemingly minor variations in SW radiation exist across different regions**^{43,53}. However, these studies are limited to the low latitudes on **the** nearside of the Moon, leaving the condition on the farside and higher latitudes largely unknown. **To enhance our understanding of the long-term changes in the SW irradiation environment across different regions of the Moon***

and its effects on lunar soil evolution over a million-year timescale, an analysis of the SW-damaged rims and SEP track records in lunar soils from the lunar farside and at high-latitudes regions is essential for gaining further insights. On December 1, 2020, the Chang'E-5 (CE-5) mission successfully returned 1.731 kg of lunar soil from a mid-latitude region on the Moon's nearside, the northeastern Oceanus Procellarum (51.916°W, 43.058°N) ⁵⁴. On June 25, 2024, the Chang'E-6 (CE-6) mission successfully returned the first samples from the South Pole-Aitken (SPA) basin (41.625°S, 153.978°W) at the lunar farside ⁵⁵⁻⁵⁷, providing a unique opportunity to investigate the long-term solar particle irradiation environment at the lunar farside and its differences from the nearside."

Q9: Line 92+96: It should be "the farside".

R9: Thank you for your careful reading. We agree with your suggestion and have added the definite article "the" before "farside" where appropriate to ensure grammatical correctness. The revised sentences now read:

Page 3 Lines 90-110:

"To date, lunar soil samples returned by the Apollo missions have been employed to investigate the solar particle radiation differences across various regions ⁴¹⁻⁴³. Keller et al. ⁷ investigated the relationship between SW-damaged rims and exposure ages in lunar soils from various regions, finding no significant differences in the growth rates of SW-damaged rims among samples. These findings suggest that seemingly minor variations in SW radiation exist across different regions ^{43,53}. However, these studies are limited to the low latitudes on the nearside of the Moon, leaving the condition on the farside and higher latitudes largely unknown. To enhance our understanding of the long-term changes in the SW irradiation environment across different regions of the Moon and its effects on lunar soil evolution over a million-year timescale, an analysis of the SW-damaged rims and SEP track records in lunar soils from the lunar farside and at high-latitudes regions is essential for gaining further insights. On December 1, 2020, the Chang'E-5 (CE-5) mission successfully returned 1.731 kg of lunar soil from a mid-latitude region on the Moon's nearside, the northeastern Oceanus Procellarum (51.916°W, 43.058°N) ⁵⁴. On June 25, 2024, the Chang'E-6 (CE-6) mission successfully returned the first samples from the South Pole-Aitken (SPA) basin (41.625°S, 153.978°W) at the lunar farside ⁵⁵⁻⁵⁷, providing a unique opportunity to investigate the long-term solar particle irradiation environment at the lunar farside and its differences from the nearside. In this work, we examine the SW-damaged rims and SEP tracks within the CE-5 and CE-6 samples, explore the long-term differences in SW irradiation environment between the nearside

and the farside of the Moon over millions of years, and study their impacts on the evolution of lunar soil grains.”

Q10: Line 110: “sample is space weathering-induced amorphous rim” There is a word missing here.

R10: Thank you for pointing this out. We have revised the sentence to improve clarity and grammatical accuracy. The updated version now reads:

Pages 3-4 Lines 113-135:

“To allow comparison with previous Apollo findings and minimize the influence of mineral-type variations, we restricted our analysis to plagioclase grains and employed the same experimental methods as in prior studies^{7,58}. In the CE-5 (CE5C1000YJFM006, CE5C0400YJFM00505 and CE5C0600YJFM00304) and CE-6 (CE6C0300YJFM001) shoveled samples, we selected grains exhibiting surface exposure features such as micro-meteorite impact craters or melt splashes, indicating prior exposure on the lunar surface and thus recording SW and SEP irradiation damage (Fig. 1a)^{59,60}. In total, seven grains from CE-5 (CE5-01 to CE5-07) and eight from CE-6 (CE6-01 to CE6-08) were selected for the focused ion beam (FIB) cross-section, respectively. As shown in Fig. 1b-d, the scanning transmission electron microscopy (STEM) and high-resolution transmission electron microscopy (HRTEM) images of the FIB cross-section indicate that the uppermost surface of the sample consists of a space weathering-induced amorphous rim, while the underlying matrix region contains solar particle tracks. The formation of rims and solar particle tracks is closely related to the exposure of samples on the lunar surface. The types of rims are primarily classified as SW-damaged rims and vapor-deposited rims. In all samples, SW-damaged rims were observed. In the CE-5 samples, their thickness ranged from approximately 40-64 nm, while in the CE-6 samples, their thickness varied from around 19 to 109 nm (Figs. S1-15). Quantitative transmission electron microscopy-energy dispersive X-ray spectroscopy (TEM-EDS) composition maps indicate that SW-damaged rim composition is consistent with the matrix (Fig. 1c). However the impact events are episodic, we did not find vapor-deposited rims in all samples. In the six particles containing CE5-PL1, CE5-PL3, CE5-PL7, CE6-PL1, CE6-PL7, and CE6-PL8, iron-rich vapor-deposited rims were observed (Figs. S1, S3, S7, S8, S14, S15). Particle tracks within the matrix were observed in all grains. Grain CE6-PL1 is shown in Fig. 1, with the remaining grains presented in Figs. S1 to S15 of the Supplementary Information.”

Q11: Table S4: This information could be better presented in a Figure, and the μm label for H is wrong (it should be nm).

R11: Thank you for your helpful suggestion. In response, we have converted Table S4 into a figure to more effectively illustrate the SRIM-simulated damage depths of Fe and H ions in bytownite and anorthite. The new figure clearly shows that, for each ion and implantation energy, the damage depths are nearly identical between the two minerals. In addition, we have corrected the unit label for H ions damage depth from “ μm ” to “nm” as you pointed out. The updated figure and caption are now included in the revised Supplementary Information (see **Fig. S16**).

Fig. S16. SRIM-simulated damage depths for (a) Fe and (b) H ions implanted into bytownite ($\text{Na}_{0.3}\text{Ca}_{0.7}\text{Al}_{1.3}\text{Si}_{2.7}\text{O}_8$, density = 2.72 g/cm^3) and anorthite ($\text{CaAl}_2\text{Si}_2\text{O}_8$, density = 2.75 g/cm^3). For each ion species and implantation energy, the resulting damage depths are nearly identical between the two minerals, indicating negligible dependence on target composition.

Reviewer #3 (Remarks to the Author):

The revised version of the manuscript "Million-year Solar Wind Irradiation Recorded in Chang'E-5 and Chang'E-6 Samples" provides updated and extended introduction and discussion sections. I want to commend the authors on the additional work that they have put in this revision and the response to previous questions.

Reply: We sincerely appreciate your thoughtful and constructive review of our manuscript. Detailed revisions have been incorporated into the updated manuscript, and our point-by-point responses to your comments are provided below.

Generally, only few comments from my side remain:

Q1: - L94: I think the authors want to say "seemingly only minor variations"

R1: We thank you for the suggestion and have revised the sentence accordingly, changing "seemingly minor variations" to "seemingly only minor variations."

Original text:

"These findings suggest that seemingly minor variations in SW radiation exist across different regions^{43,53}."

Revised text:

Page 3 Lines 94-95:

*"These findings suggest that seemingly **only** minor variations in SW radiation exist across different regions^{43,53}."*

Q2: - Line 291: "Therefore, the elevated SW irradiation at the CE-5 sampling site, relative to Apollo sites, [...]" As discussed in the previous version, I still see little evidence for there really being an increased SW precipitation of the Chang'E-5 site. The newly added section on porosity could explain this, but the differences of 10% might still be small. It would be best to adjust the wording here.

R2: Thank you for your constructive comment. We agree that the available evidence does not support a definitive claim of increased SW flux relative to Apollo sites. To avoid over-interpretation, we have revised the text to reflect potential modulation of the local irradiation environment by regolith porosity, surface slope, and thermal conditions, rather than a genuine increase in incident flux. We hope that these clarifications adequately address your concerns and demonstrate that the revised text more accurately reflects the current evidence.

Original text:

“Therefore, the elevated SW irradiation at the CE-5 sampling site, relative to Apollo sites, likely represents an additional contributing factor to the higher SW-derived water content observed in CE-5 materials. Such environmental differences—particularly elevated SW irradiation and lower surface temperatures at the CE-5 site—may not only contribute to the higher SW-derived water content, but also offer a potential explanation for the enhanced development of SW-damaged rims observed in CE-5 grains, as further discussed below.

One plausible explanation for the enhanced effective SW irradiation at the CE-5 site involves regolith porosity, which likely modulates SW–grain interactions at the microscale.”

Revised text:

Page 7 Line 293-299:

“Therefore, the enhanced growth rate of the SW-damaged rim at the CE-5 sampling site, relative to Apollo sites, may represent an additional contributing factor to the higher SW-derived water content observed in CE-5 materials.”

We note that an enhanced growth rate of the SW-damaged rim at the CE-5 sampling site does not necessarily imply an increase in the incident SW flux in space as local factors may contribute to enhanced effective SW irradiation at the sampling site. One plausible explanation involves regolith porosity, which likely modulates SW-grain interactions at the microscale.”

Q3: - Line 311: Given that the authors state there is no available precise slope data for either of the landing sites, this paragraph appears somewhat loosely connected to the rest of this section. I suggest the authors include, for example, the following reference on slope-related weathering: <https://doi.org/10.1002/2017GL075338>

R3: Thank you for your helpful suggestion. The recommended study shows that solar wind–induced space weathering varies with crater wall orientations, supporting the slope-related effects we discuss. We have added this reference and revised the text accordingly.

Original text:

“Another contributing factor to the enhanced effective SW irradiation at the CE-5 site could be local surface slope. A steeper slope can effectively reduce the solar zenith angle and enlarge the projected area facing the incoming SW, thereby increasing the effective flux of incident particles.”

Revised text:

Pages 8 Line 312-316:

“Another contributing factor to the enhanced effective SW irradiation at the CE-5 site could be local surface slope. Sim et al. (2017)⁷⁶ showed that SW-induced space weathering varies with crater wall orientations, consistent with slope-related effects. A steeper slope can effectively reduce the solar zenith angle and enlarge the projected area facing the incoming SW, thereby increasing the effective flux of incident particles.”

Thanks again.